# Comparing components for seismic risk modelling using data from the 2019 Le Teil (France) earthquake

Konstantinos Trevlopoulos[1], Pierre Gehl[1], Caterina Negulescu[1], Helen Crowley[2], Laurentiu Danciu[3]

[1]BRGM, F-45060 Orléans, France
[2]EUCENTRE, Pavia, 27100, Italy
[3]Swiss Seismological Service, ETH Zurich, Zurich, 8092, Switzerland

*Correspondence to*: Konstantinos Trevlopoulos (k.trevlopoulos@brgm.fr)

**Abstract.** Probabilistic seismic hazard and risk models are essential to improving our awareness of seismic risk, to its management, and to increasing our resilience against earthquake disasters. These models consist of a series of components,
which may be evaluated and validated individually, although evaluating and validating these types of models as a whole is challenging due to the lack of recognised procedures. Estimations made with other models, as well as observations of damages from past earthquakes lend themselves to evaluating the components used to estimate the severity of damage to buildings. Here, we are using a dataset based on emergency post-seismic assessment made after the Le Teil 2019 earthquake, third-party estimations of macroseismic intensity for this seismic event, shake-maps, and scenario damage calculations to compare
estimations under different modelling assumptions. First we select a rupture model using estimations of ground motion intensity measures and macroseismic intensity. Subsequently, we use scenario damage calculations based on different exposure models, including the aggregated exposure model in the 2020 European Seismic Risk Model (ESRM20), as well as different site models. Moreover, a building-by-building exposure model is used in scenario calculactions, which models individually the buildings in the dataset. Lastly, we compare the results of a semi-empirical approach to the estimations made with the
scenario calculations. The post-seismic assessments are converted to EMS-98 damage grades and then used to estimate the damages for the entirety of the building stock in Le Teil. In general, the scenario calculations estimate lower probabilities for damage grades 3-4 than the estimations made using the emergency post-seismic assessments. An exposure and fragility model assembled herein leads to probabilities for damage grades 3-5 with small differences from the probabilities based on the ESRM20 exposure and fragility model, while the semi-empirical approach leads to lower probabilities. The comparisons in
this paper also help us learn lessons on how to improve future testing, An improvement would be the use of damage observations collected directly on the EMS-98 scale or on the damage scale in the ESRM20. Advances in testing may also be made by employing methods that inform about the damage at the scale of a city, such as remote sensing, or data-driven learning methods fed by a large number of low-cost seismological instruments spread over the building stock.

# 1    Introduction

Earthquakes are among the disasters with most severe consequences, which include loss of human life, disruption of critical infrastructures, insured and uninsured direct economic losses, as well as socio-technical impacts in multi-risk safety contexts. Assessments based on probabilistic seismic hazard and risk analysis (PSHA, PSRA) are key elements of efforts to improve awareness of seismic risk, response, and resilience to earthquakes. As far as seismic hazard and risk in Europe is concerned, the 2020 European Seismic Hazard and Risk Models (ESHM20, ESRM20 - Crowley et al., 2021a; Danciu et al., 2021) are the state of the art models, which were created by the European Facilities for Earthquake Hazard and Risk consortium. The predictive accuracy of the multi-component ESHM20 and ESRM20 models, as that of all seismic hazard and risk models, and as that of all statistical and probabilistic models, needs to be evaluated, despite the fact that the individual components consisting them have already undergone evaluation.

In the nuclear industry, testing and evaluation of PSHA models and their components have been formalized in the form of Senior Seismic Hazard Analysis Committee (SSHAC) Hazard Studies (Ake et al., 2018). SSHAC projects aim to produce technically defensible distributions and probabilities of exceedance of ground motion intensity measures. Bommer et al. (2013) tested ground motion models and their logic tree by comparing their implementations by three independent teams of modellers. As far as the evaluation of PSHA logic trees is concerned, Marzocchi et al. (2015) argue that the hazard should be considered to be an ensemble of models, which do not need to be mutually exclusive and collectively exhaustive. Rood et al. (2020) used observations of geomechanical failures, i.e., rock toppling, to estimate upper limits of ground motion intensity measures and constrain hazard estimations for long return periods. Their procedure always leads to a reduction of the seismic hazard estimation, which depends on the model for the seismic fragility, i.e., the model estimating the probability of geomechanical failure conditioned on a ground motion intensity measure. Moreover, they proposed a procedure for dropping branches of the PSHA logic tree and reweighting the remaining. Gerstenberger et al. (2020) note that tests of national or regional hazard models are only meaningful at the level of the site, and that resorting to conversions of macroseismic intensity to ground motion intensity, when ground motion records are lacking, may introduce errors. Nevertheless, Mak and Schorlemmer (2016) did use such a conversion after testing the conversion equation itself.

In this study, to evaluate components used in seismic risk modelling, we use observations of damage in buildings in the municipality of Le Teil, France, caused by the 2019 Le Teil earthquake. Section 2 focuses on the interpretation of post-earthquake assessment damage data acquired for a small sample of buildings in terms of a 3-level scale (i.e., a scale using the green, yellow, and red colour tags), to EMS-98 damage grades. In Section 3, we detail the various assumptions and modelling choices with respect to the components of the damage calculation chain that we investigate, namely various source rupture models, building exposure models and ground motion models (GMMs) along with their site amplification models.

Subsequently, in Section 4, we do a series of comparisons based on ground motion intensity, macroseismic intensity, and damage distribution. For the comparison based on ground motion intensity, we generate samples for a set of ground motion

intensity measures (IMs) estimated by scenario computations or shake-map methods (Wald et al., 2022), for rupture parameters reported by different sources. Shake-maps are employed due to their capability to take into account any available ground motion records or macroseismic observations in the interpolation of the estimated shaking. Subsequently, we convert the IMs to macroseismic intensities using different ground-motion intensity conversion equations (GMICEs). A third-party macroseismic intensity estimation for the municipality of Le Teil, provided by detailed on-site investigations (Schlupp et al.,

2022), is then used to select the rupture parameters that lead to the most compatible macroseismic intentensities, and which are used in the scenario damage calculations.

Finally, in Section 5, we perform three types of comparisons based on probabilities of EMS-98 damage grades: (i) comparisons based on a building-by-building model, (ii) comparisons based on aggregated exposure models, and (iii) a comparison using

different risk analysis tools (Armagedom (Sedan et al. 2013), and the OpenQuake Engine (Pagani et al., 2014; Silva et al., 2014)). In the first two types, we use alternative $V_{S30}$ (the time-averaged shear-wave velocity up to a depth of 30 m) models to compare their effects on the estimated damages. The $V_{S30}$ models used are the ESRM20 topography-based model, and a geology-based model specific to France (Roullé & Monfort, 2016). In addition to the $V_{S30}$, the slope and the geology are used to account for ground motion amplification due to local site effects. In the comparisons using aggregated exposure models, the

exposure models used are the ESRM20 exposure, an aggregated exposure model based on French statistical data, and a building-by-building exposure model based on the field damage observations. The probabilities of the damages estimated based on the calculations are compared to the corresponding probabilities based on damage observations and expert judgement. The steps leading up to these comparisons are summarized in Figure 1.

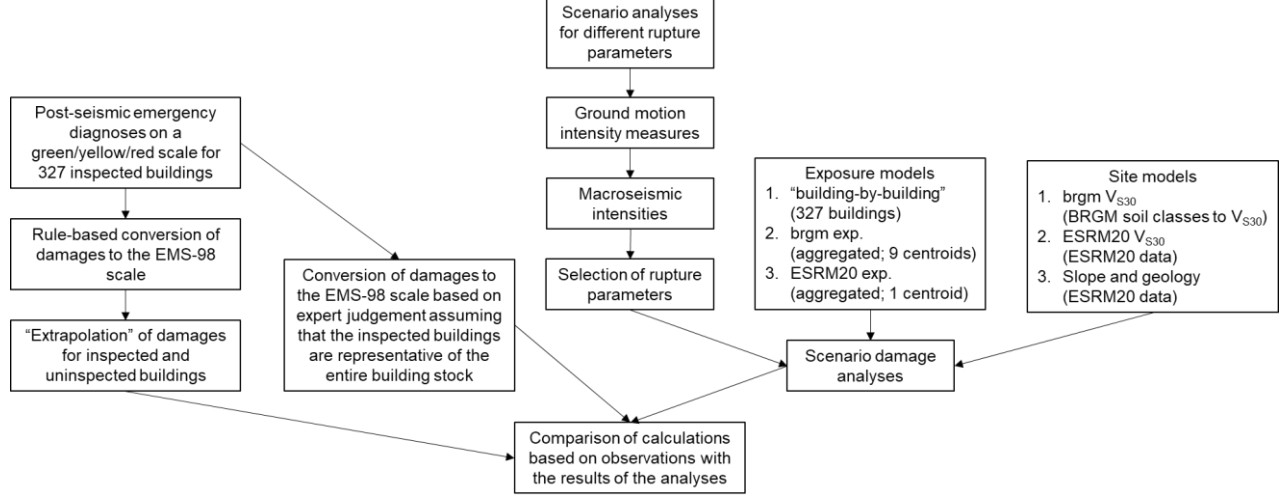


**Figure 1 Overview of the steps leading up to the comparison of the different estimations of the damages**

## 2 Seismological and damage data

### 2.1 Seismic hazard and information for 2019 Le Teil earthquake

The municipality of Le Teil is located in southeastern mainland France, a region that corresponds to low and moderate risk categories, according to the French Seismic Zonation. For Le Teil in particular, the ESHM20 estimates a mean Peak Ground Acceleration (PGA) of 0.04 g with a 0.21 % probability of exceedance in 1 year (475 years mean return period) on rock site conditions ($V_{s,30} = 800$ m/s).

The Le Teil earthquake took place on the 11th of November 2019, and its epicentre is located at 44.518° N 4.671° E (Ritz et al., 2020), with a focal depth of 1 km and a moment magnitude $M_w$ 4.9 (Ritz et al., 2020), in close proximity to the municipality of Le Teil and the town of Montélimar in the Lower Rhône valley in France. A private power plant accelerometer, located 15 km north-northeast of the epicentre, recorded PGA of 0.045 g (Schlupp et al., 2022), as the closest seismic station to the earthquake. Three stations of the French seismological and geodetic network (Résif / EPOS-FR) at 24-44 km from the epicentre

recorded PGAs in the range of 0.004-0.007 g. These four stations are at such a distance from the epicentre and the municipality of Le Teil, that they cannot accurately constrain the predicted IMs. Causse et al. (2021) used numerical modelling, including physics-based rupture modelling and modelling of near-fault wave propagation, and estimated near-fault PGAs with a 68 % confidence interval of 0.3-1.9 g in the fault projection on ground surface. They argued that their estimations are compatible with displacements of rigid block objects such as rocks and ledger stones. Moreover, they suggested that existing ground

motion models may not be useful in the case of earthquakes such as this one, with a rarely observed shallow hypocentral depth, and with rupture parameters such as stress drop that are usually associated with earthquakes not only at larger depths, but of larger magnitudes too. However, it should be noted that some branches in the ESHM20 ground motion models logic tree should be able to account for the possibility of having extreme stress parameter values, by treating uncertainty in the stress drop as a source of epistemic uncertainty (Kotha et al., 2020; Weatherill et al., 2020). As far as the attenuation of the intensity

of the PGA is concerned, the recorded value at 15 km was 0.04, which indicates a high attenuation probably due to the very shallow rupture: the Le Teil earthquake is a specific event, which generated very high large intensities right next to the epicentre, however the ground motion attenuated very quickly.

Schlupp et al. (2022) reported an EMS-98 macroseismic intensity of 7-8 for the municipality of Le Teil. This conclusion was

the product of expert judgement considering the EMS-98 definitions of the intensity degrees and damage grades, the field observations from the Macroseismic Response Group, and the EMS-98 vulnerability classes of the buildings based on land registration data. Based on this procedure, Schlupp et al. (2022) determined 765 macroseismic intensities covering the area affected by the earthquake. The isoseist line of the map by Schlupp et al. (2022) for intensity 7 includes the built area of Le

Teil: given the limited spatial extent of this area, there is practically no spatial variation of the macroseismic intensity within

this isoseist line, and the maximum is at Le Teil (7.5).

## 2.2    Post-seismic emergency assessments dataset

We produced the dataset used here by processing post-seismic emergency inspection forms, and by completing and editing an existing dataset (Perez, 2020) for 501 inspected buildings. The inspection forms were filled in by the French Association of

Earthquake Engineering (AFPS) during on-site inspections (Taillefer et al., 2021), which took place from the 3$^{rd}$ to the 5$^{th}$ of February 2020. Out of the 501 buildings, the produced dataset contains 327 entries with information about the coordinates of each inspected building, the number of storeys, the date of construction, and the description of damage for the entirety of each inspected building as well as for its structural and non-structural elements. The colour tags assigned by the post-seismic emergency assessments are on a three-level scale, i.e. green-yellow-red, which we converted to EMS-98 damage grades. The

174 entries that were not included in the produced dataset were left out due to the fact that, although they included the colour tag for the building, they lacked information with respect to the damage to the structural elements and the non-structural components, or with respect to the construction material, the year of construction, or the number of storeys. The distribution of the green / yellow / red tags across these entries (Table A5) has small differences from the distribution of the 327 entries (Table 2-2), which leads us to consider that their removal from the dataset does not introduce any significant bias.


For the conversion of the post-seismic emergency assessments to EMS-98 damage grades, we used the rules in Table 2-1. We defined these rules based on expert judgement, and they are based on the observed structural and non-structural damage, which are the criteria for classification according to the EMS-98 damage scale (Grünthal, 1998). Therefore, for this specific purpose, the essential data in the forms are the entries in the fields for the structural elements bearing vertical and horizontal loads

(which were considered separately), and for the non-structural elements as well. The rest of the fields on the forms are related to procedures for life safety, e.g. evacuation, and they were not required for classifying damage according to the EMS-98. In this way, we used the raw information from the inspection forms to classify buildings according to structural damage and not whether a building was usable or not. In the cases where a given parameter is red, the damage grade is assigned irrespective of the other parameters. As far as the column *Types of elements* in Table 2-1 is concerned, the four components are ordered

hierarchically. If both vertical and horizontal structural elements are red, then damage grade 5 is assigned, but if the horizontal structural elements are red and the vertical are yellow or green, then grade 4 is assigned. In the cases where everything is green, damage grade 1 is assigned (damage grade 1 corresponds to no structural damage and slight non-structural damage). This assignment is done based on our judgement. The dataset that we used contains only damage observations, which were made during inspections on request by the building owners. We consider that at least slight non-structural damage was the cause that

led the owners to request an inspection of their building. The results of this reclassification (which involves the distribution of

EMS-98 damage levels in the green, yellow, red tags) are presented in Table 2-2 for the entire dataset, independently of building typology.

**Table 2-1 Proposed classification of the observed damage in the EMS-98 damage grades as a function of the colour tags assigned by the inspectors.**

| Type of elements | Colour tag: G (green), Y (yellow), R (red) | | | | | | | | | | | | | | | | |
|---|---|---|---|---|---|---|---|---|---|---|---|---|---|---|---|---|---|
| Vertical load-bearing structural elements | R | | | | Y | Y | Y | Y | G | G | Y | Y | G | G | G | G | G |
| Horizontal load-bearing structural elements | | R | | | Y | Y | Y | Y | Y | Y | G | G | G | G | G | G | G |
| Internal non-structural elements | | | R | | R | Y | R | Y | R | Y | R | Y | R | Y | Y | G | G |
| External non-structural elements | | | | R | R | R | Y | Y | R | R | R | R | Y | R | Y | Y | G |
| EMS-98 damage grade | 5 | 4 | 2 | 2 | 4 | 4 | 3 | 3 | 3 | 3 | 4 | 3 | 2 | 2 | 2 | 2 | 1 |

**Table 2-2 Percentage of buildings in each damage grade as a function of the building's final tag for the entire dataset**

| Building tag | Damage grade | Count | Percentage (%) |
|---|---|---|---|
| Green | 1 | 91 | 61 |
| Green | 2 | 22 | 15 |
| Green | 3 | 35 | 24 |
| Yellow | 3 | 95 | 90 |
| Yellow | 4 | 8 | 8 |
| Yellow | 5 | 2 | 2 |
| Red | 4 | 47 | 64 |
| Red | 5 | 27 | 36 |

In the following sections, we will compare results of calculations against three different sets of damage distribution that are based on the post-seismic emergencyassessments. An overview of the estimation of the three different sets is given in Table 2-3. The first set, labelled DD1, consists of EMS-98 damage grades resulting from the conversion based on the post-seismic emergency assessments (with respect to the 327 inspected buildings), by applying the rules from Table 2-1. The damage distributions in DD2 and DD3 are estimated for the entirety of the 2778 buildings in Le Teil (according to the national statistics database): to this end, an adjustment of the distribution in DD1 is performed in order to account for the fact that only a part of the buildings in Le Teil have been inspected, by applying probabilities of damage grades given the inspection or not of the building.

**Table 2-3 Description of the different calculations of damage**

| Calculation ID | Exposure resolution | Exposure data | Damage estimation method | Damage conversion method |
|---|---|---|---|---|
| DD1 | Building-by-building (327 buildings) | AFPS emergency survey | Observations on 327 buildings (Green / Yellow/ Red tags) | Conversion to EMS-98 damage grades (Tab. 2-1) |
| DD2 | Infra-municipality districts (2778 buildings) | National statistics database | Observations on 327 buildings (Green / Yellow / Red tags) + Adjustment | Conversion to EMS-98 damage grades (Tab. 2-1) + Bias adjustment on total number of 2778 buildings (accounting for non-surveyed buildings) |
| DD3 | Infra-municipality districts (2778 buildings) | National statistics database | Observations on 327 buildings (Green / Yellow / Red tags) + Adjustment | Conversion to EMS-98 damage grades with expert judgment (Tab. 2-8) |

The calculation of the probabilities of the damage grades for DD2 are given in Table 2-4 to Table 2-7. Table 2-4 includes the probabilities of the colour tags in the original dataset for 501 buildings. Table 2-4 also includes the probabilities of the damage

grades conditioned on the colour tags, which result from the conversion of the post-seismic emergencyassessments (Table 2-2). In Table 2-5, the total probabilities of the damage grades are calculated assuming that the probabilities of the damage grades conditioned on the colour tags are representative of the 501 buildings in the original dataset. Table 2-6 gives the damage grade probabilities conditioned on whether a building has been inspected. The first line of Table 2-6 includes the probabilities based on the damage observations, while the second line includes probabilities of the damage grades for the uninspected buildings, which were selected based on our judgement and our assumption that the damage grade probabilities for the buildings that have not been inspected are different, because the inspections were made upon owner request. The probabilities selected for the buildings that have not been inspected are based on our assumption that the probabilities of damage grades 3-5 are significantly smaller than for the inspected buildings. Moreover, we make the assumption that all buildings are at least in damage grade 1. We consider that this assumption is reasonable with respect to the inspected buildings and we acknowledge that it is conservative in the case of uninspected buildings. In the case of inspected buildings, given that the inspections were made upon request by the owners, we consider that the reason behind the requests was the existence of at least non-structural damage. Furthermore, we consider that this assumption is not excessively conservative given that a large portion of the building stock in Le Teil is masonry buildings, in which non-structural cracks are commonly encountered, and whose cause is difficult to determine. The calculation of the total probabilities of the damage grades for inspected and uninspected buildings is given in Table 2-7. Given that these probabilities are practically the probabilities in Table 2-6 weighted by the probability of a building to have been inspected ($P(Insp.=False)$), they depend to a large degree on the probabilities for the uninspected buildings, because most of the buildings were not inspected ($P(Insp.=True) \gg P(Insp.=False)$).

As far as the probabilities for DD3 are concerned, they are calculated using the Table 2-8 in combination with the probabilities of the green/yellow/red tags (*P(tag)* in Table 2-4). In specific, they result if we take a 1-row vector of the values in P(tag) in Table 2-4, and do a matrix multiplication with the values in Table 2-8. This calculation differs from the calculation of the probabilities in DD2 in that it implies that the damage observations are representative of the damage over the entire town of Le Teil. This is implied by the fact that there is no conditioning on whether a building has been inspected. The probabilities in Table 2-8 reflect the judgement of experts, who participated in the post-seismic emergency survey in Le Teil. Note that these probabilities may only be applied to this particular earthquake and should not be generalized.

**Table 2-4 Probabilities of the damage grades conditioned on the colour tag assigned to a building that has been inspected during post-seismic emergency assessments**

| tag | n_buildings | P(tag) | P(DG1|tag) | P(DG2|tag) | P(DG3|tag) | P(DG4|tag) | P(DG5|tag) |
|---|---|---|---|---|---|---|---|
| Green | 238 | 0.475 | 0.610 | 0.150 | 0.240 | 0.000 | 0.000 |
| Yellow | 157 | 0.313 | 0.000 | 0.000 | 0.900 | 0.080 | 0.020 |
| Red | 106 | 0.212 | 0.000 | 0.000 | 0.000 | 0.640 | 0.360 |

**Table 2-5 DD1 calculation of the total probability of the damage grades for buildings inspected during the post-seismic emergency assessments**

| tag | P(DG1|tag)·P(tag) | P(DG2|tag)·P(tag) | P(DG3|tag)·P(tag) | P(DG4|tag)·P(tag) | P(DG5|tag)·P(tag) |
|---|---|---|---|---|---|
| Green | 0.290 | 0.071 | 0.114 | 0.000 | 0.000 |
| Yellow | 0.000 | 0.000 | 0.282 | 0.025 | 0.006 |
| Red | 0.000 | 0.000 | 0.000 | 0.135 | 0.076 |
| Sum: | 0.290 | 0.071 | 0.396 | 0.160 | 0.082 |

**Table 2-6 Probabilities of the EMS-98 damage grades conditioned on whether a building has been inspected (the probabilities for inspected buildings are based on the damage observations, the probabilities for the uninspected buildings are based on expert judgement)**

| Inspected | P(Insp.) | P(DG1|Insp.) | P(DG2|Insp.) | P(DG3|Insp.) | P(DG4|Insp.) | P(DG5|Insp.) |
|---|---|---|---|---|---|---|
| TRUE | 0.180 | 0.290 | 0.071 | 0.396 | 0.160 | 0.082 |
| FALSE | 0.820 | 0.500 | 0.300 | 0.100 | 0.050 | 0.050 |

**Table 2-7 DD2 calculation of the total probabilities of the EMS-98 damage grades accounting for both inspected and uninspected buildings**

| Inspected | P(DG1|Insp.)·P(Insp.) | P(DG2|Insp.)·P(Insp.) | P(DG3|Insp.)·P(Insp.) | P(DG4|Insp.)·P(Insp.) | P(DG5|Insp.)·P(Insp.) |
|---|---|---|---|---|---|
| TRUE | 0.052 | 0.013 | 0.071 | 0.029 | 0.015 |
| FALSE | 0.410 | 0.246 | 0.082 | 0.041 | 0.041 |
| Sum: | 0.462 | 0.259 | 0.153 | 0.070 | 0.056 |

**Table 2-8 Probabilities of EMS-98 damage grades conditioned on the building colour tag according to expert judgement and DD3 calculation of the total probabilities of the ESM-98 damage grades**

| tag | P(DG1|tag) | P(DG2|tag) | P(DG3|tag) | P(DG4|tag) | P(DG5|tag) |
|---|---|---|---|---|---|
| Green | 0.80 | 0.20 | 0 | 0 | 0 |
| Yellow | 0 | 0.40 | 0.60 | 0 | 0 |
| Red | 0 | 0 | 0.55 | 0.40 | 0.05 |

| tag | P(DG1|tag)·P(tag) | P(DG2|tag)·P(tag) | P(DG3|tag)·P(tag) | P(DG4|tag)·P(tag) | P(DG5|tag)·P(tag) |
|---|---|---|---|---|---|
| Green | 0.380 | 0.095 | 0.000 | 0.000 | 0.000 |
| Yellow | 0.000 | 0.125 | 0.188 | 0.000 | 0.000 |
| Red | 0.000 | 0.000 | 0.116 | 0.085 | 0.011 |
| Sum: | 0.380 | 0.220 | 0.304 | 0.085 | 0.011 |

## 3 Modelling assumptions

### 3.1 Rupture models

Various ground-motion scenarios are generated for different assumptions of rupture models, which are detailed in Table 3-1. The scenarios are named after the source of the data for the magnitude and the hypocentre location, i.e., *CEA* (CEA/LDG, 2011; Duverger et al., 2021), *EMSC* (EMSC, 2019), *RENASS* (Schlupp et al., 2022), *Ritz et al.* (Ritz et al., 2020) and USGS (USGS, 2019). The strike, dip, and rake angles of the focal mechanism solutions reported by *CEA* and *Ritz et al.* are arbitrarily assigned to the scenarios *EMSC* and *RENASS*, respectively. The surface of the rupture is estimated using the Wells and Coppersmith (1994) scaling relation, and the coordinates of the points defining the rupture geometry are calculated in order to be used in the OpenQuake Engine simulations and in the conversion of ground motion IMs to macroseismic intensity. In the case of the rupture model according to the parameters based on Ritz et al. (2020), the area of the rupture model is equal to 6.49 km$^2$. To calculate the coordinates of the corners of the rupture geometry, we assume that its geometric centroid is located at the hypocentre. This assumption leads in some cases to an upper rupture edge above ground surface. This is amended by translating the rupture geometry on its plane so that its upper edge coincides with the fault trace on ground surface. The depths of the upper and lower edges of the rupture geometry are used to define in the Simple Fault model the upper and lower seismogenic depths, respectively. The coordinates of the ends of the trace of the fault on the ground surface required by the Simple Fault model are calculated by projecting the rupture geometry on the ground surface in the direction of the dip. Moreover, a maximum rupture mesh spacing of 0.5 km is used, which leads to a 6 by 6 grid in all scenario calculations, which we consider sufficient.

**Table 3-1 Rupture parameters associated with the five source models**

| Rupture model | $M_W$ | Hypocentre longitude (°E) | Hypocentre latitude (°N) | Hypocentre depth (km) | Strike (°) | Dip (°) | Rake (°) |
|---|---|---|---|---|---|---|---|
| CEA | 4.9 | 4.65 | 44.53 | 2.0 | 47 | 65 | 93 |
| EMSC | 4.9 | 4.62 | 44.57 | 10.0 | 47 | 65 | 93 |
| RENASS | 4.8 | 4.64 | 44.53 | 2.0 | 50 | 45 | 89 |
| Ritz et al. | 4.9 | 4.671 | 44.518 | 1.0 | 50 | 45 | 89 |
| USGS | 4.84 | 4.638 | 44.612 | 11.5 | 53 | 57 | 99 |

## 3.2 Exposure and fragility models

In the components, three different exposure models are considered in order to characterise the built area of Le Teil. A main

distinction is made between aggregated models (i.e., distribution of building classes within a geographical unit) and models at

the level of single buildings.

The first aggregated exposure model (ESRM20 exp.), which is based on the ESRM20 exposure (Crowley et al., 2019, 2020,

2021b), consists of a single area containing a total of 1679 residential buildings. This exposure model results from the

260 simplification of the ESRM20 exposure model, by fusing similar building types with a small portion of the overall number of

buildings in the original ESRM20 exposure (Table A1) into 7 building classes (Table A2). Given that the original ESRM20

exposure includes classes with a small percentage of the total number of buildings, which could be grouped with similar

classes, we opted for such mergers in order to reduce the total number of classes and simplifying the comparisons. For example,

we decided to group in Class 1 (revised Table A1) buildings categories with 6 or more storeys, which have a small number of

265 buildings, together with buildings with 3-5 storeys on the basis of the similarity of their lateral load-bearing systems. The effect

of the simplification of the ESRM20 model is checked with an additional calculation using the original ESRM20 exposure and

the corresponding fragility models.

The second aggregated exposure model (BRGM exp.) is based on national statistical data, and it includes 9 distinct areas

(Figure 2) with 2778 residential buildings. In this exposure model, the buildings are categorized in 12 ESRM20 classes (Table

A3), which we selected based on the exposure model in Sedan et al. (2013).

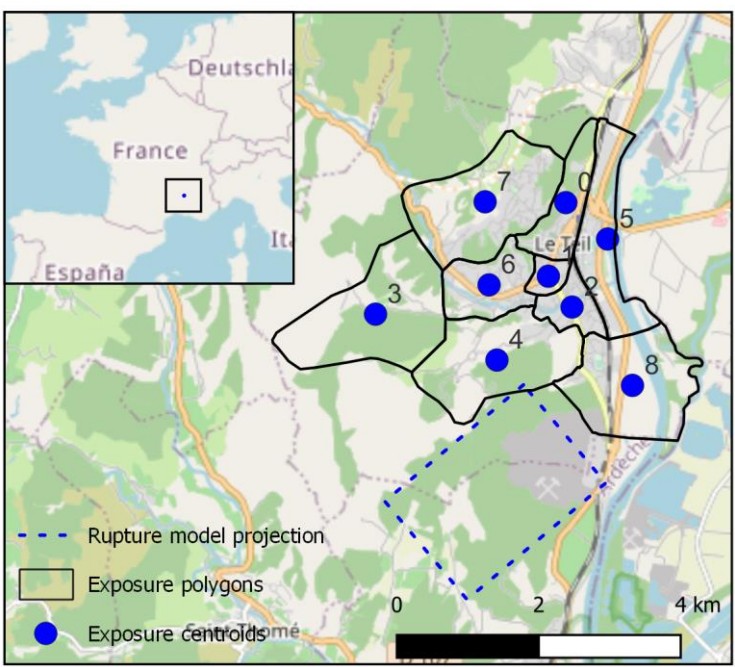

**Figure 2 Location of the 9 exposure centroids in the BRGM exposure model and surface projection of the "Ritz et al." rupture model [the map includes an OpenStreetMap layer (OpenStreetMap contributors, 2017)].**

Finally, the building-by-building exposure model includes 327 buildings located at the coordinates of the buildings in the damage dataset DD1, for which the information in the dataset is sufficient for determining the building class and damage grade. In the simulations, the fragility model consists of fragility curves from the ESRM20, which we selected according to the information in the damage dataset. Initially, we defined building classes in terms of the GEM Building Taxonomy v3.0 (Silva et al., 2022) based on the building materials and the number of storeys (Table 3-2). Moreover, we assigned an EMS-98 vulnerability class according to the building material, the year of construction, as well as the building types in Le Teil and their vulnerability class reported by Schlupp et al. (2022). Based on the building type and the vulnerability class, we then selected fragility models from the ESRM20. It is noted that the lateral force coefficientcould have been estimated based on the date of construction according to Crowley et al. (2021c), but was not considered. Moreover, it was not considered during the selection of the fragility models. An EMS-98 vulnerability class was assigned based on the year of construction, and then we selected fragility models, which we considered to be in agreement with the construction material and the EMS-98 vulnerability classes.

**Table 3-2 Assigned GEM Building Taxonomy v3.0, ESM-98 vulnerability, and ESRM20 building classes for the buildings in the post-seismic emergency assessments dataset. The fragility curves in ESRM for the selected classes are function of the listed intensity measure types (IMT)**

| GEM Building Taxonomy v3.0 class | EMS-98 vuln. class | ERSM20 class | IMT | Number of buildings |
|---|---|---|---|---|
| MUR+STDRE/LWAL+DNO/HAPP:2 | A | MUR-STDRE_LWAL-DNO_H2 | $S_a(0.3s)$ | 124 |
| MUR+STDRE/LWAL+DNO/HAPP:2 | B-D | MCF_LWAL-DUL_H2 | PGA | 20 |
| MUR+STDRE/LWAL+DNO/HAPP:4 | A | MUR-STDRE_LWAL-DNO_H3 | $S_a(0.6s)$ | 122 |
| MUR+STDRE/LWAL+DNO/HAPP:4 | B,D | MCF_LWAL-DUL_H3 | $S_a(0.3s)$ | 6 |
| CR/HAPP:2 | C | CR_LFINF-CDL-10_H2 | $S_a(0.6s)$ | 23 |
| CR/HAPP:2 | E-D | CR_LFINF-CDM-0_H1 | $S_a(0.3s)$ | 2 |
| CR/HAPP:4 | C | CR_LFINF-CDL-15_H4 | $S_a(1.0s)$ | 29 |
| CR/HAPP:4 | E | CR_LFINF-CDM-10_H1 | $S_a(0.3s)$ | 1 |

## 3.3 Ground-motion models and site amplification

In order to generate the ground motion fields in the scenario calculations, we use two GMMs in the OpenQuake Engine named: KothaEtAl2020Site , (a version of the GMM by Kotha et al. (2020) with a polynomial site amplification as a function of the $V_{S30}$)., and KothaEtAl2020ESHM20SlopeGeology. The GMMs KothaEtAl2020Site and KothaEtAl2020ESHM20SlopeGeology are based on site amplification modelling as a function of $V_{S30}$ and as a function of slope and geology, respectively. The effect of the $V_{S30}$ mapping on the estimated probabilities of the damage grade is investigated by using two different site models, which are described below.

The first site model (BRGM $V_{S30}$) uses a map of Eurocode 8 (European Commitee for Standardization, 2004) site classes, which has been assembled at BRGM for the French territory (Monfort and Roullé, 2016). This map of soil classes has then been converted into a $V_{S30}$ map by taking the median value of each soil class. The resolution in the *BRGM $V_{S30}$* model is based on a geological map at the 1/50000 scale. $V_{S30}$ values are then directly extracted at the coordinates of the entries in the exposure model, i.e., the 9 centroids in the BRGM exposure model, the one centroid in the ESRM20 exposure model, or the 327 points in the building-by-building exposure model.

The second site model (ESRM $V_{S30}$) uses the values of the *$V_{S30}$* that are returned for the coordinates of the exposure centroids by the *point* workflow in the *exposure to site tool* in the ESRM20 (Dabbeek et al., 2021). In the case of the building-by-building scenario calculations, the $V_{S30}$ values for the *ESRM $V_{S30}$* model are obtained by using the *exposure to site tool* in the ESRM20, in which the *point* workflow is applied, which returns the $V_{S30}$ value associated with the 30-arcsec cell, where the query points are found. It should be noted that these two different ways to collect $V_{S30}$ valuesbased on the coordinates the centroids (weighted mean of $V_{S30}$ values across the area versus punctual value at the centroid) may constitute an additional source of discrepancy, in addition to the initial differences between the two $V_{S30}$ models. In addition to the $V_{S30}$ values, the

exposure to site tool returned the type of geology and the slope, which are used subsequently in combination with the GMM KothaEtAl2020ESHM20SlopeGeology.

The $V_{S30}$ values from the two site models at the coordinates of the centroids in the *BRGM exp.* exposure model are compared in Table 3-3. Both site models (BRGM $V_{S30}$ and ESRM $V_{S30}$), when used in combination with the exposure model BRGM exp., consider one point for each exposure centroid, which has identical coordinates with its corresponding exposure centroid. The *BRGM $V_{S30}$* model includes $V_{S30}$ values corresponding to soft soils, while the lowest $V_{S30}$ values in the *ESRM $V_{S30}$* model

are typical of hard soil sites. The same applies to the $V_{S30}$ values for the two site models, when they are used in combination with the ESRM20 exposure model (Table 3-4).

**Table 3-3 Site parameters in the site models *ESRM $V_{S30}$* and *BRGM $V_{S30}$* used in combination with the BRGM exposure model (9 centroids)**

| Centroid | Latitude | Longitude | Region | BRGM $V_{S30}$ (m·s$^{-1}$) | ESRM20 $V_{S30}$ (m·s$^{-1}$) | $V_{S30}$ Type | Geology | Slope |
|---|---|---|---|---|---|---|---|---|
| 0 | 44.5546 | 4.6835 | 1 | 800 | 807 | inferred | CRETACEOUS | 0.0823 |
| 1 | 44.5453 | 4.6804 | 1 | 270 | 831 | inferred | CRETACEOUS | 0.0645 |
| 2 | 44.5414 | 4.6846 | 1 | 270 | 730 | inferred | HOLOCENE | 0.0487 |
| 3 | 44.5405 | 4.6498 | 1 | 800 | 726 | inferred | CRETACEOUS | 0.0768 |
| 4 | 44.5347 | 4.6713 | 1 | 800 | 831 | inferred | CRETACEOUS | 0.0467 |
| 5 | 44.5500 | 4.6909 | 1 | 270 | 699 | inferred | HOLOCENE | 0.0160 |
| 6 | 44.5442 | 4.6699 | 1 | 800 | 830 | inferred | CRETACEOUS | 0.0522 |
| 7 | 44.5547 | 4.6692 | 1 | 580 | 840 | inferred | CRETACEOUS | 0.0503 |
| 8 | 44.5315 | 4.6953 | 1 | 270 | 644 | inferred | HOLOCENE | 0.0439 |

**Table 3-4 Site parameters in the site models *ESRM $V_{S30}$* and *BRGM $V_{S30}$* used in combination with the ESRM20 exposure model (one centroid)**

| Site ID | Latitude | Longitude | Region | BRGM $V_{S30}$ (m·s$^{-1}$) | ESRM20 $V_{S30}$ (m·s$^{-1}$) | $V_{S30}$ Type | Geology | Slope |
|---|---|---|---|---|---|---|---|---|
| 0 | 44.54307 | 4.66441 | 1 | 270 | 834 | inferred | CRETACEOUS | 0.0304 |

## 4    Comparisons of estimated intensities

### 4.1    Comparison based on ground-motion parameters

Here we compare intensity measures of the seismic ground motion resulting from scenario calculations and one shake-map derivation. The scenario calculations are conducted for five different rupture models using the OpenQuake Engine (Pagani et al., 2014; Silva et al., 2014), the ground motion model (GMM) KothaEtAl2020Site and the BRGM $V_{S30}$ site model. The geometries of the ruptures in the shake-map as well as in the scenario calculations are all modelled as *Simple Faults* of flat square geometry, each defined by the set of parameters in Table 3-1. As far as the shake-map for this scenario is concerned, it was re-calculated with the USGS ShakeMap v4 engine (Wald et al. 2022), using the rupture parameters according to Ritz et al. (2020) (i.e., *Ritz et al.* model in Table 3-1), and it was constrained with ground motion measurements only (no "Did You Feel It" reports were used). However, the closest stations are over 15 km from the epicentre, which leads to practically no constraint.

To account for the uncertainty in the intensity of the ground motion, 1000 ground motion fields are generated, i.e. samples of IMs at the location of the 9 centroids of the aggregated exposure model. The ground motion fields are generated by the OpenQuake Engine for the IMs peak ground acceleration (PGA), spectral pseudo-acceleration at 0.3, 0.6, 1.0 and 3.0 s. Furthermore, the spatial correlation of the IMs is taken into account in the generation of the IM samples by using the Jayaram and Baker (2009) model in the OpenQuake Engine, assuming no clustering of the $V_{S30}$ values in the study area. As far as the correlation between spectral periods is concerned, the default correlation model *BakerJayaram2008* by Baker and Jayaram (2008) is used by the OpenQuake Engine.

On the other hand, the shake-map estimates parameters of the lognormal distributions of the IMs (PGA, spectral acceleration at 0.3, 0.6, and 1.0 s) at the 9 centroids, which are then used to generate ground motion fields, i.e. 1,000 samples for each IM at each centroid, using R (R Core Team, 2023). For the generation of the samples, we use our implementation of the correlation models for the spatial correlation (Jayaram and Baker, 2009) and the correlation between spectral accelerations at different periods (Baker and Jayaram, 2008)), as in the calculations with the OpenQuake Engine. Based on the correlation models, we define a symmetrical correlation matrix containing one row (and one column) for each spectral period at each site. The sites are defined based on the coordinates of the exposure centroids (in Section 5.1, the sites are defined using the coordinates of the individual buildings in the case of the calculations using the building-by-building exposure). Additionally, for the sampling, we use the Nearest Positive Definite Matrix using the approach by Higham (2002) as implemented in the R package *Matrix* (Matrix package authors and Oehlschlägel, 2023) in order to overcome the problem of a non-positive definite correlation matrix. The sampling is done using the R package *faux* (DeBruine, 2023).

Figure 3 shows box plots for the samples of the considered IMs, which were generated at the locations of the exposure centroids. For a specific IM, the median and the mean of the entirety of the samples for all centroids are represented by the line at the middle of the box and the point marker, respectively. The boundaries of a box mark the 1$^{st}$ and 3$^{rd}$ quartile, while the whiskers approximate the 95 % confidence interval. If we consider only the boxplots corresponding to the five scenario calculations, the dispersions of the samples are equivalent, as expected due to the use of the same GMM. However, the differences with respect to the means of these five IM samples have to be attributed to the differences between the epicentre locations, the depth of the hypocentre, and the focal solution, because these are the parameters affecting the distance between the exposure centroids and the geometry of the rupture. Moreover, the means for the scenarios *EMSC* and *USGS* are consistently the lowest. We attribute this primarily to the hypocentral depths in these two scenarios (10.0 and 11.5 km), which are significantly larger those in the other three scenarios, leading to distances from the rupture between 10.0 and 25.0 km, when the corresponding distances in the other three scenarios are less than 5.0 km. Regarding the samples based on the shake-map derivation, the boxplot whiskers are relatively shorter than those for the five scenarios, signifying smaller dispersions of the IM logarithms. This difference should primarily originate from the differences between the GMMs in the shake-map configuration and in the scenario calculations.

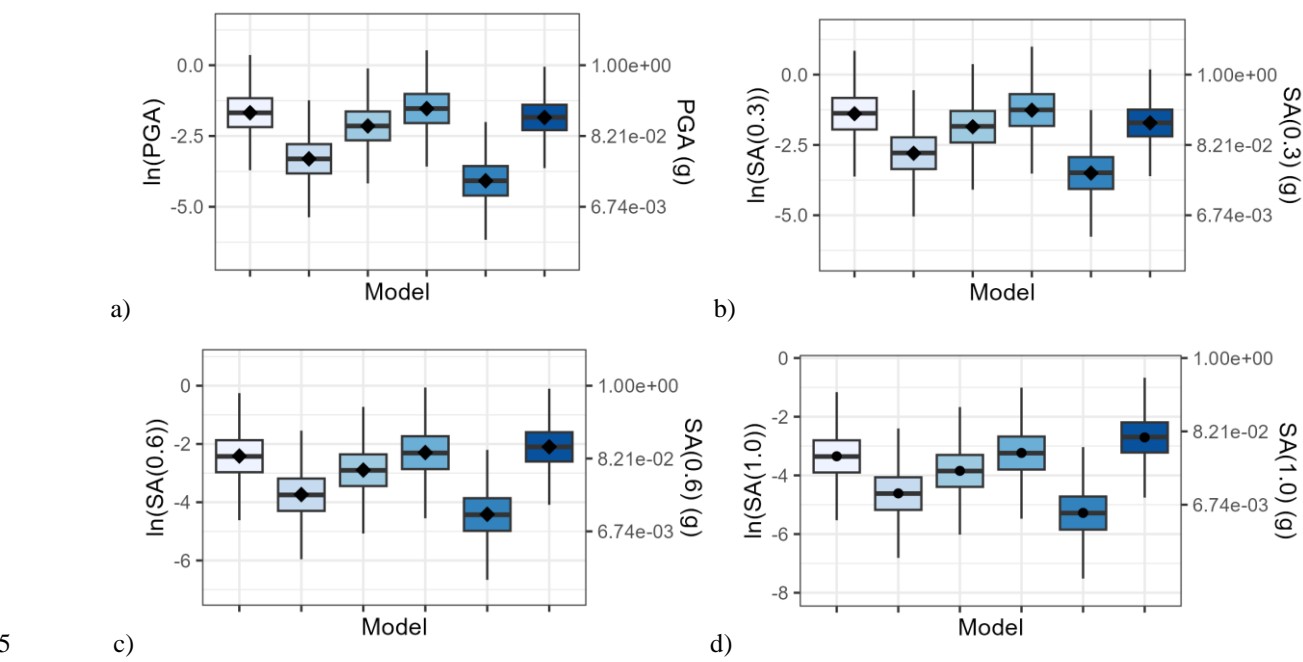

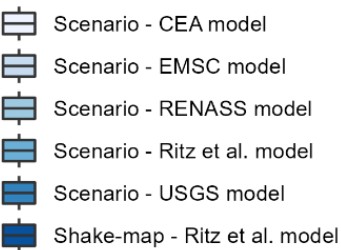

**Figure 3 Ground motion intensity measures aggregated from all exposure centroids (the edges of the box are located at the first and third quartile, respectively, the line at the middle of the box is located at the median, the point marker is located at the mean of the sample, the whiskers extend up to 1.5 times the distance between the first and third quartile approximating the 95 % confidence interval).**

## 4.2 Comparisons based on macroseismic intensity

The generated IM samples are subsequently converted to macroseismic intensities using GMICEs and they are compared with the macroseismic intensity reported by Schlupp et al. (2022). The aim of this comparison is to identify the rupture models leading to macroseismic intensities closest to the reported ones. Moreover, another motive for this comparison is the fact that it is difficult to compare the models with measured observations (i.e., recordings of seismic stations), since such measures are very sparse (the nearest station is around 15km from the epicentre). Therefore, in the absence of measures in the epicentral area, it is difficult to compare the effects of different rupture distances in this area to measured ground-motions (this is where the relative differences in rupture distance are the largest, as they are greatly reduced further away from the epicentre). This is why we use macroseismic intensity (precise estimates obtained from field surveys) for the comparison. Two GMICEs are used for this comparison, which we consider compatible with the study area. These are the GMICEs by Faenza and Michelini (2010) (Equation 1) and by Caprio et al. (2015) (Equation 2).

$$MCS = a + b \cdot logIM + \sigma$$

Where MCS is the Mercalli-Cancani-Sieberg intensity, IM is PGA (in cm·s$^{-2}$) or PGV (in cm·s$^{-1}$), and $\sigma$ is the model's standard deviation.

$$INT = a + b \cdot logIM + \sigma$$

Where *INT* is a combination of the Modified Mercali Intensity (MMI) and the Mercalli-Cancani-Sieberg intensity (MCS), IM is the ground motion intensity measure, i.e., PGA (in cm·s$^{-2}$) or PGV (in cm·s$^{-1}$), *a* and *b* are the model's parameters, and σ is the model's standard deviation. The Caprio et al. (2015) model is bilinear and its parameters are found in Table 4-1, while the

Faenza and Michelini (2010) model is the single line model in Faenza and Michelini (2010), whose parameters are found in Table 4-2. To account for model uncertainty during the conversions with Eq. 1-2, random residuals were generated from zero-centred normal distributions with the corresponding standard deviation and added to the means given by the equations.

**Table 4-1 Parameters used in the implementation of the model by Caprio et al. (2015)**

| IM type | IM range | a | b | σ |
|---------|----------|------|-------|-----|
| PGA (cm·s$^{-2}$) | $\log_{10}(IM) < 1.6$ | 2.270 | 1.589 | 0.6 |
| | $\log_{10}(IM) \geq 1.6$ | -1.361 | 2.671 | 0.5 |

**Table 4-2 Parameters used in the implementation of the model by Faenza and Michelini (2010)**

| IM type | a | b | σ |
|---------|------|------|------|
| PGA (cm·s$^{-2}$) | 1.68 | 2.58 | 0.35 |

Figure 4 shows the boxplots for the MCS and the INT, respectively, which result from the conversion of the IM samples. Despite the fact that the MMI and MCS have differences, we adopt here the guidelines by Musson et al. (2010), which take the two scales as equivalent (to each other and to the EMS-98 scale) up to intensity 10. We make this assumption to distinguish the effects of the employed GMICEs on the distributions of the generated samples of macroseismic intensities in Figure 4 from the differences due to the underlying hazard model components.

In order to assess the usefulness of the distribution for each scenario in Figure 4, we are using the 7.5 EMS-98 intensity estimated by Schlupp et al. (2022) for the municipality of Le Teil. The MCS distributions resulting from the Faenza and Michelini (2010) model, whose median is closer to the 7.5 observation-based estimation, are those for the *CEA*, *RENASS*, and *Ritz et al.* scenarios, and the shake-map derivation. As far as the application of the Caprio et al. (2015) model (Figure 4b) is concerned, it leads to macroseismic intensity distributions with larger dispersions and lower medians compared to the distributions calculated using the model by Faenza and Michelini (2010) (Figure 4a) in the cases considered. In the cases examined here, the distributions whose median is closest to the 7.5 observation-based estimation, are those from the scenarios *CEA*, *RENASS*, and *Ritz et al.* and from the shake-map. Based on this, the *Ritz et al.* rupture model is used in the calculations that follow.

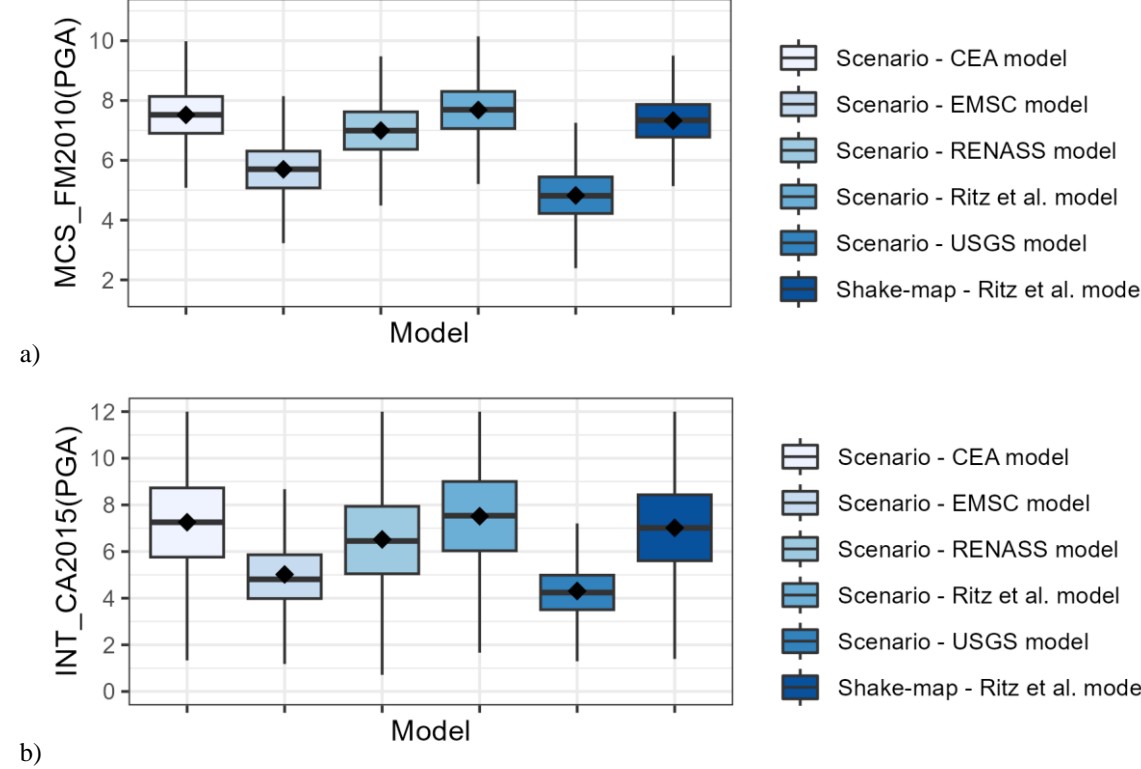

a)

b)


**Figure 4 Boxplots for a) the Mercalli-Cancani-Sieberg (MCS) macroseismic intensity as a function of the PGA given by the ground motion-to-intensity conversion equation by Faenza and Michelini (2010) (FM2010), and b) the macroseismic intensity (INT) as a function of the PGA given by the ground motion-to-intensity conversion equation by Caprio et al. (2015) (CA2015) (the edges of the box are located at the first and third quartile, respectively, the line at the middle of the box is located at the median, the point marker**

**is located at the mean of the sample, the whiskers extend up to 1.5 times the distance between the first and third quartile approximating the 95 % confidence interval).**

## 5 Comparisons of estimated damages

### 5.1 Estimated damage based on a building-by-building exposure model

First, we perform *scenario damage* calculations using the OpenQuake Engine and the building-by-building exposure model, which includes 327 buildings with classes defined in Table 3-2. The ground motion fields in the calculations are generated using four different configurations (Table 5-1), which include the two different GMMs, i.e. KothaEtAl2020Site, and KothaEtAl2020ESHM20SlopeGeology, and three different site models, i.e. *BRGM* $V_{S30}$ and *ESRM* $V_{S30}$. In all cases, the rupture is modelled according to the *Ritz et al.* scenario (Table 3-1). A scenario calculation is also performed using as input

ground motion fields generated from the shake-map procedure described in Section 4.1 (GM4 in Table 5-1).

**Table 5-1 The configurations (GM Map IDs) used to generate the ground motion fields in the scenario damage calculations based on a building-by-building exposure model**

| GM Map ID | Type | GMM | Site model | Rupture model | Observations |
|---|---|---|---|---|---|
| GM1 | ground-motion field | KothaEtAl2020Site | BRGM soil classes to Vs30 | Ritz et al. | No |
| GM2 | ground-motion field | KothaEtAl2020ESHM20 SlopeGeology | ESRM site model (Slope & Geology) | Ritz et al. | No |
| GM3 | ground-motion field | KothaEtAl2020Site | ESRM20 Vs30 model | Ritz et al. | No |
| GM4 | shake-map | KothaEtAl2020Site | BRGM soil class to Vs30 | Ritz et al. | Seismic stations |

The damages based on the *scenario damage* calculations are firstly calculated on the damage scale of the ESRM20 fragility models, and then they are converted to the ESM-98 damage scale using as criterion the structural damage according to Table A4. Due to this conversion, all buildings in the calculation have at least non-structural damage. In this case, the building-by-building exposure model includes the inspected buildings, and as discussed in Section 2.2, it is reasonable to assume that completely undamaged buildings are underrepresented in the in the sample of inspected buildings.


Figure 5 gives the distribution of the damage grades and the corresponding number of buildings based on the calculations. First, it is worth noting that the GM4 simulation leads to similar, but somewhat lower probabilities for the damage grades 3-5 than the GM1 simulation. GM1 and GM4 use the same GMM and site model, the difference lies in the fact that the GM4 uses ground motion fields based on a shake-map. The main drivers of the probabilities of the damage grades are the buildings in
the classes MUR-STDRE_LWAL-DNO_H2 and MUR-STDRE_LWAL-DNO_H3, which include 38 % and 37 %, respectively, of the total number of buildings in the model. These two classes are also the most vulnerable among the classes in the model, as indicated by the fact that they were classified in the EMS-98 vulnerability class A. The fragility curves of these two building classes are functions of $S_a(0.3s)$ and $S_a(0.6s)$, respectively. Based on the results in Figure 3, we consider that the $S_a(0.3s)$ is on average higher in the calculation *Scenario – Ritz et al. model* (GM1) than in *Shake-map – Ritz et al.*
*model* (GM4), and that there are no significant differences between the two with respect to the $S_a(0.6s)$. This is the factor to which we attribute the differences in the probabilities of the damage grades based on the simulations GM1 and GM4.

The GM3 calculation leads to the lowest probabilities for the damage grades 3-5 amongst all computations in Figure 5. In this simulation, 68 % of the buildings are located on sites with $V_{S30} \geq 800\ m \cdot s^{-1}$, while in GM1 72 % of the buildings are on sites
with $V_{S30} \leq 360\ m \cdot s^{-1}$, which is expected to lead to higher ground motion intensities due to site amplification. It interesting to note that the GM2 calculation, which uses the KothaEtAl2020ESHM20SlopeGeology GMM, gives results which are practically halfway between the results of the calculations GM1 and GM3.

Figure 5 also includes the estimation DD1 (Table 2-3) of actual damages, which is based on our conversion of the damage observation. For damage grades 4 and 5, there are significant differences between the probabilities based on DD1 and the probabilities based on the scenario calculations and the shake-maps (GM1-4), however, they are not as important as the differences in the case of the damage grades 2 and 3. We presume that the rule that we used for the translation of the damage observations to damage grades (Table 2-1) is the source of these discrepancies. Moreover, DD1 leads to a distribution in Figure 5 that has an unusual valley for damage grade 2. The proposed mapping of damage observations assigns damage grade 3, when the vertical or the horizontal structural elements have a yellow tag (see Table 2-1). We believe that a yellow tag with respect to the structural elements signifies moderate structural damage, hence damage grade 3. The fact that in these cases a green tag was assigned (Table 2-2), perhaps indicates that a further inspection took place, which either reclassified the damage as green structural damage, or as yellow non-structural damage. Such cases could be taken into account by a future refinement of the proposed mapping scheme.

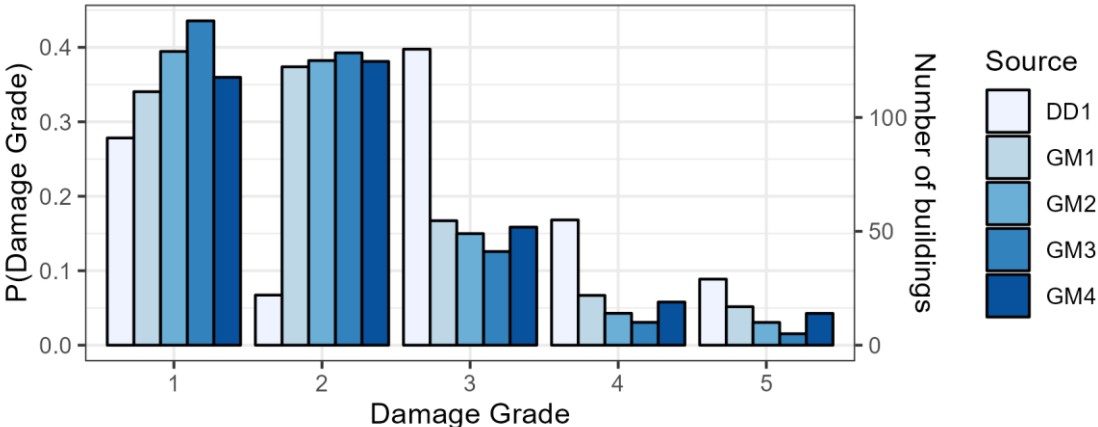

**Figure 5 Distribution of the damage grades based on the calculations with the building-by-building exposure model compared with the DD1 estimation of actual damages**

## 5.2 Estimated damage based on aggregated exposure models

In addition to the building-by building calculations, we perform a series of *scenario damage* calculations with the two aggregated exposure models that include the total number of residential buildings in the municipality of Le Teil. In the calculations with the aggregated exposure models, the ground motion intensity measures are modelled with nine different combinations of GMMs, site models and exposure models, as shown in Table 5-2.

**Table 5-2 Combinations of ground motion map IDs with the exposure models for each damage scenario ID**

| Damage scenario ID | GM Map ID | Exposure model |
|---|---|---|
| DS1 | GM1 | BRGM exposure |
| DS2 | GM1 | ESRM20 exposure |
| DS3 | GM2 | BRGM exposure |
| DS4 | GM2 | ESRM20 exposure |
| DS5 | GM3 | BRGM exposure |
| DS6 | GM3 | ESRM20 exposure |
| DS7 | GM4 | BRGM exposure |
| DS8 | GM4 | ESRM20 exposure |
| DS9 | GM3 | Original ESRM20 exposure |


As in the calculations based on the building-by-building exposure model, the damages based on the *scenario damage* calculations are converted from the ESRM20 damage grades to the ESM-98 damage grades using Table A4. In this case, this assumption may lead to an overestimation of non-structural damage. However, as discussed in Section 2.2, this overestimation may not be excessive due to possible non-seismic pre-existing non-structural damage, especially in masonry buildings, which

make up the biggest part of the building stock in Le Teil.

In Figure 6, we may see the effect of the different exposure models on the distribution of the damage grades and on the corresponding number of buildings. Figure 6a includes the distributions of the damage grades for the damage scenarios DS5, DS6, and DS9, which all use the same rupture model, GMM, and site model (GM3). Compared to DS5, the DS6 calculation

for the ESRM20 exposure leads to somewhat higher probabilities for damage grades 3-5. The differences between DS5 and DS6 are due to the use of the BRGM or ESRM20 exposure model, respectively. Moreover, Figure 6a includes the results of damage scenario DS9, which uses the original ESRM20 exposure and fragility model to check the effect of the simplification of the ESRM20 exposure and fragility models. By comparing the results between DS6 and DS9, we conclude that the simplification has a minor effect on the results. Figure 6a also includes our estimations DD2 and DD3. It is reminded that DD2

depends mostly on expert judgement and on the damage observation on a lesser extent, while DD3 is entirely based on expert judgement (see Section 2.2). Note that, in DD3, the probabilities for damage grades 3-5 depend heavily on the probabilities of these damage grades conditioned on a red tag, which were assigned based on expert judgement. In hindsight, it may have been too optimistic to assign a 55 % probability of damage grade 3 in case of a red tag. Alternative assignments of the probabilities for a red tag may smooth out in DD3 the peak for damage grade 3. The probabilities of the damage grades 3 and 4 calculated

by the damage scenario calculations DS5, DS6, and DS9 are lower than the DD2 and DD3 estimations. However, for damage grade 5, the results of the damage scenarios are found in the range between the DD2 and DD3 estimations. The same trends

may be observed (not shown here) by comparing the calculations DS1 and DS2 (based on GM1), or DS3 and DS4 (based on GM2), or DS7 and DS8 (based on GM4).

The numbers of buildings in Figure 6b are calculated by multiplying the total number of buildings in the exposure model by the probabilities in Figure 6a. In the case of the DD2 and DD3 estimations, we chose to calculate the number of buildings by multiplying with the number of buildings in the BRGM exposure model. Despite the difference in the total number of buildings in the BRGM and in the ESRM20 exposures (2778 versus 1679), the results of the damage scenarios for damage grades 3-5 present minor differences.


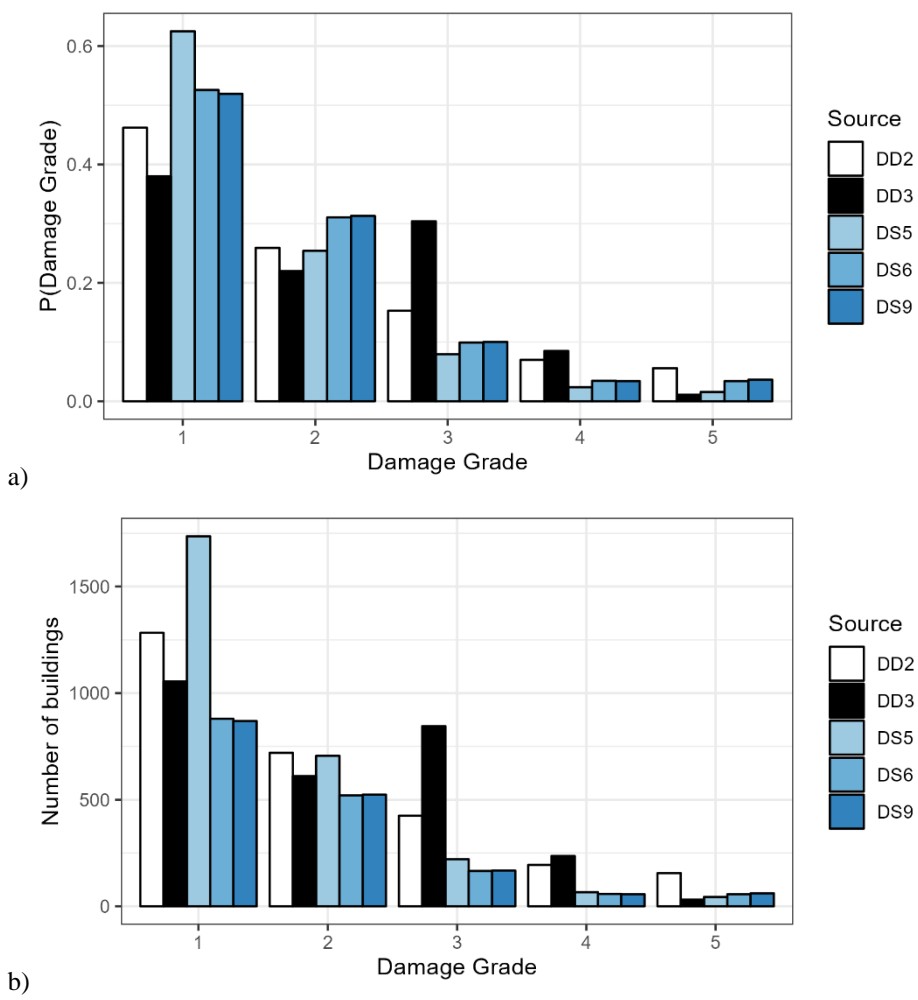

a)

b)

**Figure 6 Effect of the exposure model on the a) probabilities and b) number of buildings per EMS-98 damage grade for the calculations with an aggregated exposure including the total number of buildings in Le Teil**

The comparison with respect to the site amplification models is done using the damage scenario calculations DS1, DS3, DS5, DS7 (Figure 7a), where the same exposure model is used, i.e., the BRGM exposure model, but each time we use one of the four different GM Maps (GM1 to GM4 in Table 5-1 and Table 5-2). The effect of using the BRGM $V_{S30}$ model instead of the ESRM20 $V_{S30}$ model may be seen by comparing DS1 with DS5. The probabilities of the damage grades 2-5 are slightly lower in the scenario DS5. This may be explained by the fact that the $V_{S30}$ values are higher in GM3 than in GM1, however we would expect more important differences considering the differences in the $V_{S30}$ values shown Table 3-3. The damage grade probabilities in the scenario DS3, which uses the KothaEtAl2020ESHM20SlopeGeology GMM, are between the results for

DS1 and DS5 for all damage grades. As far as the results based on DS7, which uses a shake-map, they present small differences

from those from DS1, which is reasonable considering that they use the same ground motion and site model, and that the

updating based on ground motion observations in the shake-map is minor. The results based on the ESRM20 exposure and

fragility model (Figure 7b) show a similar image with the exception of the difference between the DS2 and DS8. Again, DS2

and DS8 use the same ground motion and site model, so the origin of this difference may be the consideration of observations

in the shake-map used by DS8.

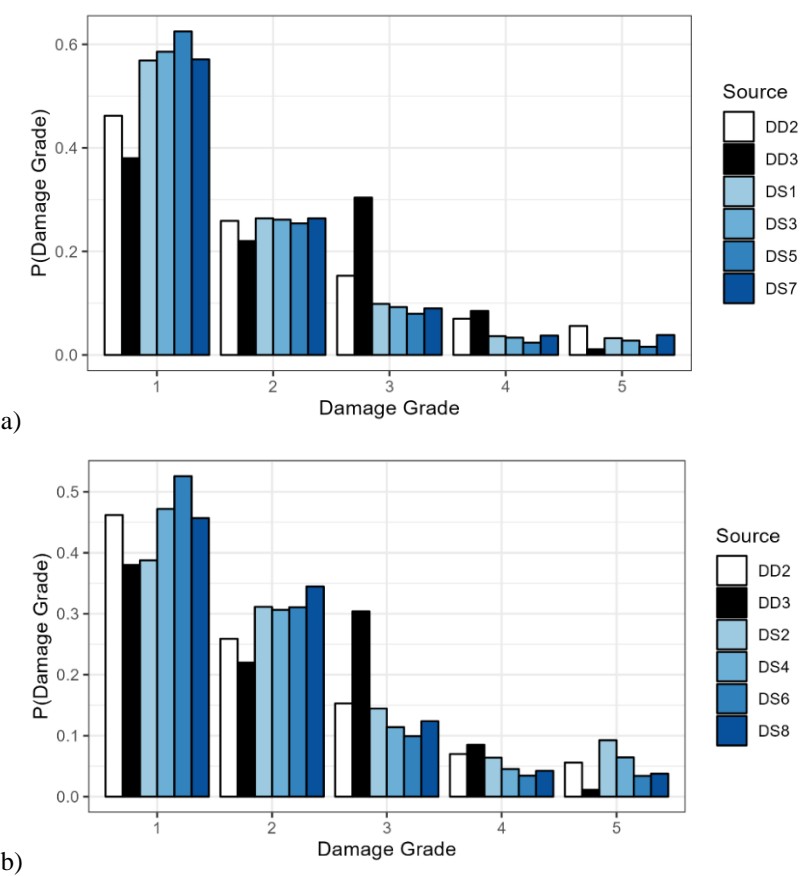

a)

b)

**Figure 7 Effect of the GMM and site model on the probabilities of EMS-98 damage grade for the calculations with a) the BRGM**
**and b) the ESRM20 aggregated exposure models including the total number of buildings in Le Teil**

### 5.3 Estimated damage based on a semi-empirical vulnerability approach

For the comparison with respect to the distribution of damages according to different calculation methodologies, we compare the estimated damages using the seismic risk analysis tool *Armagedom* (Sedan et al. 2013), running on the VIGIRISKS platform (Negulescu et al. 2023), with an estimation made with the DS1 scenario calculation with the OpenQuake Engine.

The Armagedom tool implements the semi-empirical macroseismic method developed by the RISK-UE project (Lagomarsino and Giovinazzi, 2006). In contrast to the scenario calculations with the OpenQuake Engine, where 1,000 ground motion realizations are used to account for ground motion uncertainty, the calculation with Armagedom takes as input a third-party pre-calculated map of macroseismic intensity. For the calculation with Armagedom, we use the macroseismic intensity map produced by Schlupp et al. (2022). The semi-empirical macroseismic method applied by Armagedom calculates the mean EMS-98 damage grade as a function of the macroseismic intensity and two parameters, i.e. the vulnerability and the ductility index. These indices have been assigned to building classes in the exposure model used for the calculation using Armagedom based on criteria such as the material and the year of construction (Sedan et al. 2013). Subsequently, the semi-empirical macroseismic method applied in Armagedom assumes a binomial distribution to calculate the probabilities of exceeding the EMS-98 damage grades as a function of macroseismic intensity. On the other hand, the OpenQuake Engine uses ground motion realizations in combination with fragility curves to generate realizations of damages.

The estimated distribution of buildings in each damage grade based on the two calculations is given in Figure 8, along with the distribution from the damage datasets DD2 and DD3. The percentage of buildings with Heavy and Very Heavy damage is 1.1 % and 0.0 % with Armagedom, and 3.7 % and 3.3 % with the OpenQuake Engine, respectively. Both the DS1 and the Armagedom calculation lead to estimations for damage grades 3 and 4, which are lower than the estimations DD2 and DD3. As far as damage grade 5 is concerned, the DS1 calculation estimates a probability of 3.3 % which lies between the DD2 and DD3 estimations, i.e., 5.6 % and 1.1% respectively. On the other hand, the Armagedom calculation globally underestimates damages when compared to the DS1 calculation. It should be noted that DS1 is based on the GM1 map, which corresponds to macroseismic intensity ranges (see Figure 4) that are well in line with the estimates by Schlupp et al. (2022), i.e. intensity around 7.5. Therefore, differences between DS1 and Armagedom may be mostly attributed to the different methods of damage estimation, i.e. the conversion between building vulnerability classes and corresponding fragility functions.

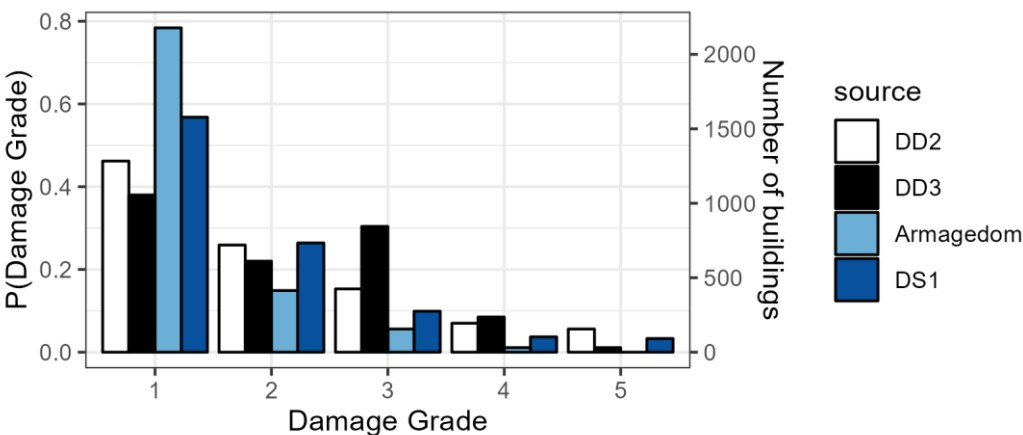

**Figure 8 Estimation of damage grades using the Armagedom tool compared to the estimations DD2-DD3 and the results of the DS1 calculation.**

 **6 Conclusion**

Using simulations of earthquake scenarios and shake-maps, we conducted comparisons based on ground motion intensity, macroseismic intensity, and the estimated number of damages based on different risk model compoenents. Moreover, we produced a dataset of 327 entries containing damage on the EMS-98 scale based on emergency post-seismic assessments on a 3-level (red/orange/green) scale, which were made after the Le Teil 2019 earthquake. The damage on the EMS-98 scale in the dataset is the result of a conversion based on a proposed rule, which considers structural and non-structural damage. The produced dataset was also used to make estimations for the entirety of the residential building stock in Le Teil.

Based on scenario calculations using the OpenQuake engine, as well as shake-maps, we calculated the ground motion intensity at a series of points of interest on the town of Le Teil, and then we converted the ground motion intensity to macroseismic intensity. This was done for different models of the earthquake rupture to select the model to be used in subsequent damage scenarios.

The damage scenarios used different ground motion models, site models, and exposure and fragility models to study the effect of these modelling assumptions. The GMMs used are KothaEtAl2020Site and KothaEtAl2020ESHM20SlopeGeology, while the site models include a site model based on $V_{S30}$ values based on a map of site classes produced by the BRGM, and a site model based on the ESRM20. As far as the exposure models are concerned, they include the BRGM exposure for Le Teil, a model based on French national statistical data, and the ESRM20 exposure.

The scenario damage calculaitons lead to probabilities for damage grades 3-5 based on the ESRM20 with small differences from the probabilities based on the BRGM exposure and fragility model.. Furthermore, the damage scenarios using the ESRM20 exposure and fragility model are overall in better agreement with the calculations DD2 and DD3 (see Section 2.2 for the details of the calculations). In general, the scenario damage calculations estimate lower probabilities for damage grades 3-4 than the DD2 and DD3 calculations using the damage dataset, while they are in better agreement in the case of damage grade 5. The estimation based on the Armagedom tool results in probabilities of damage grades 3-5 which are even lower than those based on the damage scenario using the BRGM exposure and fragility model. As far as the ground motion and site models are concerned, the damage grade probabilities based on the KothaEtAl2020ESHM20SlopeGeology model lead in general to results between those obtained with KothaEtAl2020Site in combination with the BRGM and the ESRM20 site model. This is observed in the scenario calculations with the building-by-building and the aggregated exposure models.

At this point it is worth referring to the difficulties, limitations, and challenges related to the presented comparisons. A first and obvious one is the conversion of emergency post-seismic diagnoses assessments into ESM-98 damage grades. The proposed rule (Table 2-1) that uses the red/orange/green tags for structural and non-structural elements may have a significant effect on the damage grades resulting from the conversion, although we did not study the effect of possible alternative conversion rules on the results of calculation DD1. We acknowledge that the proposed rule can be refined, especially if we consider the valley in damage grade 2 in calculation DD1. The results of calculation DD2 are affected by proposed rule, but to a lesser extent, given that DD2 mostly depends on expert judgement with respect to the probabilities of damage in the uninspected buildings (Table 2-7). The conversion in calculation DD3 is purely subjective, and it reflects the experts' judgement with respect to this particular earthquake. One refinement could be a probabilistic rule which would return damage grade probabilities instead of a single value for the damage grade as a function of the colour tags for structural and non-structural elements.

Recommending a model that is used in the comparisons here is difficult. However, we will attempt to offer some guidance to the reader and propose the DS1 calculation, which uses the GM1 and the BRGM exposure. As far as site effects are concerned in the context of calculations with aggregate exposure models, we consider that the combination of the BRGM $V_{S30}$ model and BRGM's (infra-communal) exposure is the best choice at the city scale. This choice is supported by the values of the $V_{S30}$ in tables 3.3 and 3.4, where the values of this combination are closest to the site effects expected in the area. There are two reasons for this: the resolution of the exposure (nine points instead of one) and the resolution of the site effect zones in the BRGM $V_{S30}$ model is better than that of the ESRM20, which is expected since the ESRM20 has been developed for application on the European scale. We would also like to underline that resolution of the exposure (extent of the polygons) is also important for the representation of the site effects, in terms of the parameters exactly at the centroid and their averages over the exposure polygon. If we were to allocate research and development resources for seismic risk analysis, we would prioritize the detailed description of site effects and the assignment of building classes to relatively small exposure zones.

As far as the calculations using the building-by-building exposure model are concerned, using them to calibrate the scenarios based on the aggregated exposure models is challenging. This is due to the need to convert tags into degrees of damage, or to reinterpret collected data. In France (as in Italy), emergency post-seismic assessments (by the AFPS or by the firefighters) tag buildings on a three-colour scale (red, yellow and green), which is common practice and indeed useful in an emergency context. One recommendation is to add to the forms, which are used to collect data during the post-seismic emergency assessments, the classification of the building according to the EMS-98 damage grade or to the damage scale in the ESRM20.

Finally, we note how future seismic risk testing could be improved. A challenge in comparing the results of calculation DD2 with the damage observations is the estimation of damage in the entire building stock based on the damage observed over a sample of buildings. We believe that the buildings that were included in the emergency post-seismic inspections in Le Teil are not a representative sample of the entire building stock. We presume that this could be true in other cases too. Not only buildings in Le Teil were inspected upon request, but we believe that undamaged or completely destroyed buildings were not inspected, because that would be meaningless in emergency post-seismicassessments, which aim to inform about the risk associated with the use of impacted buildings. Therefore, there is no available information with respect to the damage in the uninspected buildings and one may use expert judgement (as we did in calculations DD2 and DD3), which may be biased, or seek more rigorous solutions. In order to estimate based on the sample of inspected buildingsto the damage in the entire building stock one may consider resorting to remote sensing or solutions such as rapid damage assessments based on data collected by numerous pre-installed low-cost sensors, which may be exploited by data-driven learning and forecasting methods, as proposed by Goulet et al. (2015), to estimate damage at the scale of a city.

## 7    Author contribution

KT: Conceptualization, Data curation, Formal analysis, Investigation, Methodology, Software, Validation, Visualization, Writing – original draft preparation, Writing – review & editing

PG: Conceptualization, Data curation, Formal analysis, Investigation, Methodology, Software, Supervision, Validation, Writing – original draft preparation, Writing – review & editing

CN: Conceptualization, Data curation, Funding acquisition, Methodology, Project administration, Supervision, Writing – original draft preparation, Writing – review & editing

HC, LD: Writing – review & editing

## 8 Competing interests

The authors declare that they have no competing interests.

## 9 Code/Data availability

Code and data are available upon request.

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

**Table A1 Selected ESRM20 fragility classes based on the building types in Le Teil according to the ESRM20**

| Original ESRM20 type | N. buildings | Selected ESRM20 frag. class | # class |
|---|---|---|---|
| CR+PC/LWAL+CDN/HBET:3-5 | 53 | CR_LDUAL-DUL_H4 | 1 |
| CR/LDUAL+CDL+LFC:4.0/HBET:3-5 | 7 | CR_LDUAL-DUL_H4 | 1 |
| CR/LDUAL+CDM+LFC:4.0/HBET:3-5 | 3 | CR_LDUAL-DUL_H4 | 1 |
| CR/LDUAL+CDL+LFC:4.0/HBET:6- | 3 | CR_LDUAL-DUL_H4 | 1 |
| CR/LDUAL+CDN/HBET:6- | 2 | CR_LDUAL-DUL_H4 | 1 |
| CR+PC/LWAL+CDN/HBET:6- | 1 | CR_LDUAL-DUL_H4 | 1 |
| CR/LDUAL+CDM+LFC:4.0/HBET:6- | 1 | CR_LDUAL-DUL_H4 | 1 |
| CR/LFINF+CDL+LFC:4.0/H:1 | 76 | CR_LFINF-CDL-10_H2 | 2 |
| CR/LFINF+CDL+LFC:4.0/H:2 | 67 | CR_LFINF-CDL-10_H2 | 2 |
| CR/LFINF+CDM+LFC:4.0/H:1 | 42 | CR_LFINF-CDM-10_H2 | 3 |
| CR/LFINF+CDM+LFC:4.0/H:2 | 37 | CR_LFINF-CDM-10_H2 | 3 |
| CR/LFINF+CDN/HBET:3-5 | 38 | CR_LFINF-CDL-15_H4 | 4 |
| CR/LFLS+CDN/HBET:6- | 9 | CR_LFINF-CDL-15_H4 | 4 |
| MUR+CL/LWAL+CDN/H:2 | 378 | MUR-CL99_LWAL-DNO_H2 | 5 |
| MUR+ST/LWAL+CDN/H:2 | 130 | MUR-CL99_LWAL-DNO_H2 | 5 |
| MUR+CL/LWAL+CDN/H:1 | 690 | MUR-CL99_LWAL-DNO_H1 | 6 |
| W/LWAL+CDN/H:1 | 100 | W_LFM-DUL_H2 | 7 |
| W/LWAL+CDN/H:2 | 43 | W_LFM-DUL_H2 | 7 |

**Table A2 Summary of the exposure based on the European Exposure model for the municipality of Le Teil**

| # | Selected ESRM20 class | N. of buildings |
|---|---|---|
| 1 | CR_LDUAL-DUL_H4 | 70 |
| 2 | CR_LFINF-CDL-10_H2 | 143 |
| 3 | CR_LFINF-CDM-10_H2 | 78 |
| 4 | CR_LFINF-CDL-15_H4 | 46 |
| 5 | MUR-CL99_LWAL-DNO_H2 | 508 |
| 6 | MUR-CL99_LWAL-DNO_H1 | 690 |
| 7 | W_LFM-DUL_H2 | 143 |


**Table A3 Summary of the BRGM exposure model for the municipality of Le Teil**

| # | Selected ESRM20 class | Number of buildings |
|---|---|---|
| 1 | CR_LFINF-CDL-10_H1 | 296 |
| 2 | CR_LFINF-CDL-10_H2 | 138 |
| 3 | CR_LFINF-CDL-15_H2 | 348 |
| 4 | CR_LFINF-CDL-15_H3 | 631 |
| 5 | CR_LFINF-CDL-15_H4 | 12 |
| 6 | CR_LFINF-CDM-0_H1 | 27 |
| 7 | CR_LFINF-CDM-10_H1 | 8 |
| 8 | MCF_LWAL-DUL_H2 | 127 |
| 9 | MCF_LWAL-DUL_H3 | 278 |
| 10 | MUR-STDRE_LWAL-DNO_H1 | 130 |
| 11 | MUR-STDRE_LWAL-DNO_H2 | 483 |
| 12 | MUR-STDRE_LWAL-DNO_H3 | 300 |

Table A4 Conversion of the damage scale of the ESRM20 fragility models to the EMS-98 damage scale used for the comparisons

| ESRM20 | EMS-98 |
|---|---|
| D0 no damage (combined structural and non-structural damage) [implied damage state] | Grade 1: Negligible to slight damage (no structural damage, slight non-structural damage |
| D1 slight (combined structural and non-structural damage) | Grade 2: Moderate damage (slight structural damage, moderate non-structural damage |
| D2 moderate (combined structural and non-structural damage) | Grade 3: Substantial to heavy damage (moderate structural damage, heavy non-structural damage) |
| D3 extensive (combined structural and non-structural damage) | Grade 4: Very heavy damage (heavy structural damage, very heavy non-structural damage) |
| D4 complete (combined structural and non-structural damage) | Grade 5: Destruction (very heavy structural damage) |

**Table A5 Empirical probabilities of the colour tags for the 174 entries that were excluded from the damage dataset used for the calculations**

| tag | P(tag) |
|---|---|
| Green | 0.518 |
| Yellow | 0.294 |
| Red | 0.188 |