# Peer review of "Comparing components for seismic risk modelling using data from the 2019 Le Teil (France) earthquake"

_EGUsphere, 2023_

## Referee Comment (RC1)

**Review of Manuscript egusphere-2023-1740**

**Testing the 2020 European Seismic Hazard and Risk Models using data from the 2019 Le Teil (France) earthquake**

This manuscript presents a comparison between building damage states as observed in the field after the 2019 Le Teil earthquake and those calculated by means of combining different components of existing risk models from different sources (not just the 2020 European Seismic Hazard and Risk Models). The damage survey has been processed by the authors according to expert judgment to obtain damage in terms of the EMS-98 scale. Different rupture models from the literature, as well as the USGS ShakeMap for this earthquake, are used to generate several realisations of ground motion fields in terms of peak ground and spectral acceleration (PGA, SA). The PGA values are then converted to macroseismic intensities using conversion equations, and these macroseismic intensities results are compared against the 7.5 value obtained in existing literature from field surveys with the purpose of selecting one rupture model to be used for the subsequent damage calculations carried out using the OpenQuake engine. Three main comparisons in terms of damage are carried out, combining different components (e.g., exposure, fragility, site effects) of different risk models as well as different risk calculation methods/software, and contrasting them against the results of the processed damage survey of the 2019 Le Teil earthquake.

While the work presented in this manuscript is of interest to the research community to understand how different existing models and modelling choices affect the calculated damage and, most importantly, how the calculated damage compares against observations from a real earthquake, the manuscript has many significant shortcomings that would need to be addressed before it can be published in NHESS. I thus recommend that the manuscript be reconsidered for publication after major revisions.

**Main Comments**

1. In my view, the title of the paper does not accurately describe its contents, due to three main reasons:

    I. The word "testing" is being used loosely throughout the manuscript (see point 2 below).

    II. The paper makes comparisons using a variety of sources of model components (exposure, fragility, ruptures) that are not just from the 2020 European Seismic Hazard and Risk Models (ESHM20, ESRM20). The ground motion model used and labelled as being the ESHM20 one does not seem to be the model actually implemented in ESHM20 but a previous version. When using the ESRM20 exposure model the building classes are "simplified", effectively changing the ESRM20 exposure model. To my understanding (as such an outline is missing in the introduction), three comparisons in terms of damage are carried out:

    1) Section 3.3.1: Comparison between (a) damage calculated with the Armagedom software, using the vulnerability index approach, EMS-98 vulnerability classes, and an in-house exposure model, and (b) damage calculated with OpenQuake, using fragility models from the European Seismic Risk Model 2020 (ESRM20) selected to be equivalent to the EMS-98 vulnerability classes, and the in-house exposure model converted onto ESRM20 building classes.

    2) Section 3.3.3: Comparison between (a) damage processed from the field survey, (b) damage calculated using the USGS ShakeMap, (c) damage calculated with OpenQuake, (seemingly) using the Kotha et al. (2020) GMPE (not the version used in ESHM20/ESRM20), and the BRGM $V_{S30}$ model (which I infer is the ESRM20 $V_{S30}$ model derived from geology, not used in ESRM20), and (d) the same as (c) but using the ESRM20 $V_{S30}$ model derived from topography (used in ESRM20 for cratonic and subduction areas, but not for shallow crustal areas, which is the case of France). All cases use the same exposure model, a building-by-building model based on the individual buildings from the damage survey to which ESRM20 building classes were assigned by the authors. All cases use the ESRM20 fragility models.

3) Section 3.3.4: Comparison between (a) damage processed from the field survey and (b through g) six combinations of the following components:

    i. Exposure models: (i) the ESRM20 aggregated exposure model defined by administrative unit (one administrative unit), but with a large modification to the building classes that makes it different from the ESRM20 exposure model, and (ii) an in-house model derived from statistical data (8 or 9 centroids), to which ESRM20 building classes were assigned.

    ii. Site models: (i) the BRGM $V_{S30}$ model (which I infer is the ESRM20 $V_{S30}$ model derived from geology, not used in ESRM20), values retrieved for the centroid of the administrative unit or 8-9 points of the exposure models, and (ii) the ESRM20 $V_{S30}$ model derived from topography (used in ESRM20 for cratonic and subduction areas, but not for shallow crustal areas, which is the case of France), with the value for the ESRM20 exposure being a population-weighted average of the whole administrative unit and the values for the in-house exposure model being retrieved from the 30 arc-sec cell that contains each of the 8-9 points.

    iii. Ground motions: (i) the USGS ShakeMap, and (ii) calculated with OpenQuake using the Kotha et al. (2020) GMPE (not the version used in ESHM20/ESRM20).

As can be seen, no "pure" components of ESHM20/ESRM20 appear to have been being used ("pure" = exactly as they have been used in the ESHM20/ESRM20 models) and several components from other sources are being used as well. The title should reflect that the models being compared come from a variety of sources and decisions from the authors.

III. Finally, "testing […] hazard and risk models" may be misleading, as it can be easily interpreted as testing the full probabilistic seismic hazard and risk models (i.e., probabilities of exceedance of ground motion, average annual losses, etc.), which is not what is done in the paper (and, furthermore, cannot be done using data from one single earthquake).

To sum up, the paper shows comparisons (no statistical tests) of observed damage against damage calculated using components of risk models from different sources.

I believe it is fundamental that a new title be assigned to the manuscript, taking into consideration the comments above.

2) I have found the word "testing" being used loosely throughout the manuscript as a synonym of "comparing", "validating", "verifying", "carrying out quality assurance", etc. The word "testing" usually implies a formal statistical procedure using statistical indicators of goodness of fit, similarity between distributions, etc., which are not what is presented in the paper. The paper mostly carries out comparisons, without quantifying differences across different models/components. Please avoid over-using and over-stretching in meaning the word "testing", rewording where necessary. Some outstanding examples:

a. The title in itself. The European Seismic Hazard and Risk Models are probabilistic models. The paper uses some of their components to carry out ground motion and damage calculations that are compared against damage observations from one earthquake. One earthquake cannot test or validate a probabilistic model, only its components.

b. Line 34: Bommer et al. (2013) call their work "quality assurance" and not "testing". Throughout the paper they use the word "check" far more than they use the word "test".

c. Sections 3.1 and 3.2: These sections are not testing ground motions or macroseismic intensities, they are comparing ground motions and macroseismic intensities calculated with different rupture models (against one value of macroseismic intensity) with the purpose of selecting one rupture to use in the remaining comparisons of the paper. The PGA and SA values are not compared against instrumental measurements at all (values of PGA are mentioned in lines 64-66 but not marked on the plots or mentioned again in Section 3.1). The sections are presented as "tests" when, in reality, they are an intermediate comparative step to select rupture parameters.

3) In line with the first point above, and with the purpose of aiding the reader to navigate comparisons carried out across so many different options, please re-phrase the last paragraph of the introduction to describe more accurately the work contained in the paper:

   a. Lines 46-47: This sentence states that the work is done "to test components of the ESHM20 and the ESRM20" models, giving the impression that only ESHM20 and ESRM20 components will be used, but components from other models are used as well, and these are not mentioned at all here. Please mention the other models used.

   b. Line 48: I suggest not using the expression "scenario simulations" to refer to ground motion scenarios calculated by means of ground motion models, as the word "simulations" is usually used to refer to physics-based ground motion simulations (this is not critical).

   c. Lines 49-50: This sentence may give the impression that "the most compatible scenario simulation" is selected in terms of the one that gives the results closest to the USGS ShakeMap, but this is not what is stated in lines 50-52 or in Sections 3.1/3.2 (and further along in the paper), which show comparisons of all rupture models with respect to each other (including the USGS ShakeMap) and finally comparing intensities against the value reported by Schlupp et al. (2022).

   d. Lines 49-52: The meaning of "the most compatible scenario simulation" and "the most plausible scenario simulation" is not clear. After reading the paper, I believe the authors mean "the most compatible earthquake rupture", or "the earthquake rupture that leads to the most compatible macroseismic intensities".

   e. Lines 46-54: While several sentences are dedicated to explaining the comparison of ground motions and macroseismic intensities (which is only a preliminary step to select a suitable rupture to carry out the damage comparisons), very little is said about the core of the work. Please consider delineating the content of the three damage comparisons in a similar fashion to what I have written above under point (1), or perhaps with a figure. This is relevant to help the user navigate the paper, as so many different considerations/decisions are being made in each case.

4) The authors state (lines 113 and 315) that they are using the Kotha et al. (2020) ground motion prediction equation (GMPE) in the form of its **KothaEtAl2020Site** implementation in OpenQuake. However, all ESHM20/ESRM20 sources indicate that this is not the final GMPE used in ESHM20 and ESRM20. This being the case, the **KothaEtAl2020Site** GMPE should not be labelled as "ESHM20 GMF" (e.g., line 314), as this can be misleading for the reader. A more fundamental implication is that, with this GMPE being used, it is not the ESHM20 ground motion model that is being "tested", as implied in the title. Weatherill et al. (2020) and the ESHM20 report (Danciu et al., 2021) explain that a series of modifications were introduced to the Kotha et al. (2020) GMPE for the implementation in ESHM20 and ESRM20. Fundamentally, and given that the authors of the present manuscript emphasise the comparison of different $V_{S30}$ models, **KothaEtAl2020Site** has a different amplification function for site effects, and the site-to-site variability of the GMPE was calibrated only on measured $V_{S30}$, which means that an incompatibility arises when using it with inferred values of $V_{S30}$. As explained in the OpenQuake documentation[1]:

   a. **KothaEtAl2020Site** is a "preliminary adaptation of the Kotha et al. (2020) GMPE using a polynomial site amplification function dependent on Vs30 (m/s)".

   b. **KothaEtAl2020ESHM20** is an "adaptation of the Kotha et al. (2020) GMPE for application to the 2020 European Seismic Hazard Model, as described in Weatherill et al. (2020)". Page 89 of the ESHM20 report (Danciu et al., 2021) explains that **KothaEtAl2020ESHM20** is the GMPE used in ESHM20. Site effects in this implementation depend on $V_{S30}$ and whether that $V_{S30}$ is a measured quantity or inferred from proxies (e.g., slope), so as to account for the uncertainty associated with using inferred values. Page 69 of Danciu et al. (2021) specifies that ESHM20 refers to ground
* * *
[1] https://docs.openquake.org/oq-engine/master/reference/openquake.hazardlib.gsim.html#openquake.hazardlib.gsim

motions on the "reference rock" ($V_{S30}$ of 800 m/s everywhere). The ESHM20 logic tree input file[2] also shows that **KothaEtAl2020ESHM20** is being used for the calculations.

   c. **KothaEtAl2020ESHM20SlopeGeology** is an "adaptation of the ESHM20-implemented Kotha et al. (2020) model for use when defining site amplification based on slope and geology rather than inferred/measured Vs30". The ESRM20 logic tree input file[3] and its "cut" version used for shallow-crustal areas when comparing against past earthquakes[4] indicate that this is the GMPE used in ESRM20 to calculate losses. Site effects in this implementation depend on slope and geology, not $V_{S30}$ (e.g., second paragraph of Section 3.2 of the ESRM20 report, page 16). ESRM20 uses this model together with the slope and geology of the ESRM20 model, which can be retrieved with the "exposure-to-site" tools cited in the present manuscript.

As a consequence, reference to the **KothaEtAl2020Site** GMPE should be modified so that it is not named as "the ESHM20 GMPE" or "the ESRM20 GMPE". Alternatively, the analyses could be re-done using the **KothaEtAl2020ESHM20SlopeGeology** GMPE and associated ESRM20 site model (slope and geology, not $V_{S30}$), as in ESRM20. One should also note that using **KothaEtAl2020ESHM20** with $V_{S30}$ values other than 800 m/s would not necessarily be representative of either the ESHM20 or ESRM20 models.

5) Associated with the previous point, I believe it is very important that clarity is added with respect to the site models used in the comparisons. When comparing against Weatherill et al. (2023) (cited by the authors) and the ESHRM20 documentation, the explanations (e.g., lines 266-272) in the paper lack from some clarity:

   a. It is not fully clear what the "BRGM's $V_{S30}$ database" refers to, as there are two $V_{S30}$ models in the cited reference Weatherill et al. (2023): one based on topography alone, and another based on geology alone. The ESRM20 exposure-to-site tools (which the authors use and cite in the present manuscript) return the $V_{S30}$ values from the topography-based model, as the comparisons in Weatherill et al. (2023) showed that it performed better than the geology-based one. As Table 3-5 (line 310) shows different $V_{S30}$ values for the two (and quite round values for the BRGM case), I infer that the "BRGM's $V_{S30}$ database" refers to the geology-based $V_{S30}$ model presented in Weatherill et al. (2023). Please clarify in the manuscript.

   b. The manuscript would benefit from adding some sentences regarding the resolution of each of the two models, as this is relevant for the reader to understand what is being compared (e.g., in lines 267-272). From Fig. 7 of Weatherill et al. (2023) it looks like in the "BRGM's $V_{S30}$ database" there are three geologic units, associated with three ranges of $V_{S30}$ values (is the uncertainty being sampled to assign values in the paper?). The "point" workflow of the ESRM20 exposure-to-site tool returns the values associated with the 30-arcsec cell to which the target point belongs, as 30-arcsec is the resolution of the model.

   c. It is noted that the $V_{S30}$ values returned by the exposure-to-site tool are not used in ESRM20 in France (non-cratonic shallow seismicity). These $V_{S30}$ values are used with the craton and subduction GMPEs selected for the areas of Europe where the shallow-crustal ESHM20 GMPE (i.e., **KothaEtAl2020ESHM20SlopeGeology**) is not applicable (e.g., see page 16 of the ESRM20 report, Crowley et al., 2021). The GMPE used for ESRM20 (i.e., **KothaEtAl2020ESHM20SlopeGeology** in OpenQuake) calculates site amplification based on slope and geology directly, not $V_{S30}$. Please clarify in the manuscript that the $V_{S30}$ values labelled as ESHM20 are actually not used in ESHM20/ESRM20 in France.
* * *
[2] https://gitlab.seismo.ethz.ch/efehr/eshm20/-
/blob/master/oq_computational/oq_configuration_eshm20_v12e_region_main/gmpe_complete_logic_tree_5br.xml
[3] https://gitlab.seismo.ethz.ch/efehr/esrm20/-/blob/main/Hazard/gmpe_logic_tree_5br_slope_geology.xml
[4] https://gitlab.seismo.ethz.ch/efehr/esrm20_scenario_tests/-
/blob/main/models/esrm20/GMPE/gmpe_logic_tree_5br_shallow_default.xml

d. From my understanding, the site amplification model and $V_{S30}$ maps are part of ESRM20 and not ESHM20, as ESHM20 focused on hazard on the reference rock. Please name them as ESRM20, not ESHM20.

6) In my view, it is necessary to add a map that shows the resolution/locations of the different exposure models and site models, the spatial extent of the municipality of Le Teil, the location of the selected rupture plane, etc. This is important for the reader to be able to understand the different models that are being compared and interpret the differences observed.

7) The conclusions section is too short and does not discuss the results with depth. It only focuses on marginal observations. It consists of three paragraphs, the first (and longest) of which focuses extensively on the comparison of macroseismic intensities (which is not the core of this work), the second of which briefly mentions that the exposure model was a key difference-maker in the results, without elaborating on reasons, and the third paragraph discusses potential improvements to the analysis by changing the criteria used to post-process the field damage survey, highlights the need for more standardised field survey practices, and comments about the importance of accounting for buildings not included in the survey, which has not been discussed in the paper and for which explanations are not given. Please re-write the conclusions focusing on the large number of different model components that have been compared, to reflect the work done.

I have found the statement about the effect of the exposure model (lines 359-362) quite hard to see in Fig. 5, which shows so many different models. Moreover, lines 323-333 focus on the differences due to the $V_{S30}$ model, not the exposure. I strongly recommend to find alternative ways to show and compare these results (perhaps several plots "grouping" results according to exposure, or $V_{S30}$), and potentially even to quantify the differences between models, so that it becomes clearer to the reader whether exposure or site effects have had a greater influence in the discrepancies with observed values.

The importance of including in the calculations buildings that were not part of the damage survey is mentioned in the conclusions (lines 368-369), but I cannot find it discussed before. Please explain why it is important to include those buildings and comment on why the damage survey seems to cover such a small proportion of the buildings of the municipality of Le Teil. Did they only survey buildings on-demand from the owner? Can it be assumed that the rest of the buildings were undamaged? This is important as well to interpret the plots in Fig. 5.

Apart from this, the first paragraph (lines 350-357) talks extensively about macroseismic intensities calculated with the AS2000 model. The acronym AS2000 is not defined at all within the text. Line 354 suggests the AS2000 has been used to convert from SA(1 s) to macroseismic intensity, and , lines 355-357 highlight that SA(1 s) is not representative of the buildings in Le Teil, but Section 3.2 discusses two models that convert from PGA/PGV (not SA) to macroseismic intensity. I thus infer AS2000 stands for Atkinson and Sonley (2000), one of the conversion models used by the Armagedom software, according to Sedan et al. (2013). However, no macroseismic intensity values calculated using the Atkinson and Sonley (2000) conversion equation are presented in the paper. Please revise and correct as needed.

8) Similarly to the conclusions, the abstract would need a revision to include mention of all other models that have been used, as per my previous comments. Please revise the last sentence of the abstract (lines 17-19), which vaguely hints on conclusions that do not match the conclusions section or the content of the work.

**Other Comments on Content**

1. Line 56: Please remove "and risk" from the title, as the section does not describe seismic risk in the area.

2. Lines 70-74: While this statement can be generally valid, it is noted that the ground motion model used in ESHM20 is a backbone model whose central tendency is derived from European data that may be lacking representation of such shallow earthquakes with a relatively large stress drop, but whose different branches account for the possibility of having more "unusual" stress parameters (i.e., uncertainty in the

stress drop is treated as an epistemic uncertainty). Please see Kotha et al. (2020) and Weatherill et al. (2020) and consider rephrasing (otherwise it suggests that the authors agree with Causse et al. 2021 in this particular case and believe a priori that the ESHM20 ground motion model cannot be able to represent this earthquake).

3. Line 101, Table 2-1: There are some aspects of the table that would benefit from clarification in the text:

   a. How should the reader interpret the first four columns that contain "R" and empty spaces? Does it mean that while a certain parameter is red, the EMS-98 damage grade is as indicated, irrespective of the other parameters? Are the four components ordered as per a hierarchy? I.e. if both vertical and horizontal structural elements are red, then it is damage grade 5, but if the horizontal structural elements are red and the vertical ones are yellow or green, then it is 4?

   b. The far right column shows all components in green and the damage grade resulting in 1. Is this because all entries in the survey have some sort of damage and thus "green" is to be interpreted as "damaged, but usable" and not include "undamaged"? It calls the reader's attention that everything is green and the damage grade is not zero. Please comment in the paper.

4. Line 106, Table 2-2:

   a. In the caption, please clarify this is the buildings' "final" tag (as opposed of tags by components). "… as a function of the *buildings' final tags* for the entire dataset".

   b. It calls my attention that several green buildings end up classified as ESM-98 damage grade 3, which corresponds to moderate structural damage and heavy non-structural damage. I would expect moderate structural damage to lead to the need of further inspection and repair before the building can be used, while "green" means that the building can be used again immediately. This could be the reason why in Fig. 4 the "observation based" probabilities for damage grade 2 are notably low when compared against damage grades 1 and 3 (the distribution has an unusual "valley" in damage grade 2). Can it be that several of the green buildings that ended up classified as damage grade 3 are, actually, damage grade 2? Moreover, Table 3-6 suggests the authors also believe green should map only to damage grade 1 or 2.

5. Associated with the previous point, there seem to be different probabilities of damage and numbers of damaged buildings from observations presented in different plots and the text, which I have found confusing. I have found/observed:

   a. The probabilities of damage from observations differ in Fig. 4 with respect to Fig. 5.

   b. The numbers of buildings from observations in Fig. 5b are much larger than the 327 buildings included in the damage survey. Why is this the case?

   c. At the same time, the plots in Fig. 5 have two separate categories, "Exp. judg.-based" and "Observation-based", but I have found no explanation regarding what this means, as lines 324-326 only say "*Two of the sources consist of probabilities based on expert judgement ("Exp. judg.-based"), and probabilities based on our conversion of the damage observations to damage grades ("Observation-based")*", but the meaning of "based on expert judgement" is not explained. It is noted as well that "*our conversion of the damage observations to damage grades*" is also "*expert judgment*", and thus the difference between the two requires a more detailed clarification.

   d. The above makes me wonder if one of the two "observation" labels in the plots in Fig. 5 has been created using Table 3-6. I have been unable to find any reference to Table 3-6. Please clarify if Table 3-6 is being used and reference it within the text if this is the case.

   e. If more than one method has been used to obtain damage grades from the survey data (apart from the one described in Section 2.2), all methods need to be specified (and given distinct names/labels) in Section 2.2.

f. The conclusions state "*The proposed testing procedure based on the observed damages could be improved by introducing a probabilistic rule for the conversion of damage observations on the three-level colour tag (red, yellow, green) scale to the EMS-98 damage scale*" (lines 364-365). To my understanding, this is exactly what Table 3-6 is showing. If this is the case, and it has been used, then please adjust the conclusions.

g. I cannot find any reason for Table 3-6 not to be used. Showing and discussing "observed" damage results obtained using both strategies (Table 2-1 and Table 3-6), which is potentially what is shown in Fig. 5 but not sufficiently explained, would convey to the reader the inherent uncertainty involved in the comparison between the models and the observations (i.e., "observations" are not a ground truth), which is fundamental in any comparison between models and data (i.e., the uncertainties do not only exist in the models).

6. Associated with the previous point, please explain in the paper how the ESRM20 damage scale (associated with the ESRM20 fragility models) was converted into the EMS-98 scale, as this is another source of uncertainty in the comparison.

7. Lines 110 and 161: The titles of Sections 3.1 and 3.2 need to be changed, as they do not reflect the content of these sections. Neither section presents a test. They are both a procedure to select a rupture model to carry out the damage comparisons. The first sentence of Section 3.1 needs to be changed as well, as the section does not present a comparison against macroseismic intensities.

8. Line 111 (and other instances): Although the citation of the Wald et al. (2022) paper indicates that it is the USGS ShakeMap that is being used, it would be good to be explicit (by saying "USGS ShakeMap"), as the USGS ShakeMap software is also used by other organisations with their own configuration (e.g., the European ShakeMap, the Italian ShakeMap).

9. Line 114: Which site model was used for the ground motion comparisons?

10. Line 139 states that the ground motions were "aggregated over all exposure centroids", but it is not specified whether the values shown are means or medians (of all points). Please specify.

11. Line 139: It is stated that ground motions are calculated at the exposure centroids. However:

    a. To my understanding, OpenQuake does not calculate the ground motions at the exposure points themselves but at the points of the site model that are closest neighbours to the exposure points (and assigns the ground motions to the exposure points by closest neighbours, not interpolation). This can be checked by looking at the sitemesh_XXX.csv output by OpenQuake, as this shows the locations at which ground motions were calculated. If this is the case, it would be relevant to know what site model is being used and its resolution with respect to the resolution of the exposure points.

    b. At this stage, the exposure model has not been described, and different exposure models are used later on in the paper. Please indicate if the "exposure centroids" refer to the building-by-building data of the post-earthquake damage survey or other locations.

12. Lines 149-150: It would be relevant to comment on whether the USGS ShakeMap for this earthquake was constrained with direct ground motion measurements (from stations) and/or Did You Feel It macroseismic intensity observations. For reproducibility, please include as well the version of the USGS ShakeMap used, as the USGS recalculates ShakeMaps when new data or new algorithms become available.

13. Line 151, Fig. 1: It would help the reader if the vertical axis contained the non-logarithmic values of the IM (potentially side by side with the logarithmic ones, or as a scale on the right side of the plot).

14. Line 181, Table 3-2: Is it relevant to show the parameters for the CA2015 model and not the FM2010 model?

15. Lines 193 and 197 use the acronym "KO2020", which has not been defined.

16. Lines 210-226: There are some aspects of the comparison shown in Section 3.3.1 that are not explained and are relevant for interpreting the results. Please specify in the paper:

   a. Lines 212-213 state that the "ESHM20 ground motion logic tree" was used, but so far there has been no reference to the ESHM20 ground motion logic tree, only to the **KothaEtAl2020Site** implementation of the Kotha et al. (2020) GMPE, which, as explained earlier, is not the one used in ESHM20. Please clarify which logic tree is being used.

   b. Lines 214-215: If "equivalent" exposure and fragility models are being used "*so as to limit the effect of these two factors on the differences between the two estimations*", what is the purpose of this comparison? Comparing a model in Armagedom against a model in OpenQuake? Is the equivalence between the models fully guaranteed? Please clarify the purpose of the comparison presented in Section 3.3.1.

   c. Lines 215-216: Please clarify in the paper the meaning of "the exposure model in Armagedom". I am not familiar with the software, but the paper of Sedan et al. (2013) gives the impression that Armagedom is a software and the user can input any exposure model as desired. Please clarify in the paper how this exposure model was defined.

   d. Lines 215-221: Does the exposure model used in OpenQuake maintain the 9 centroids mentioned in line 217?

   e. Please comment in the paper (a paragraph would suffice) about the details of the damage calculation in Armagedom: use of conversion models to transform PGA into macroseismic intensity, calculation of a mean damage grade as a function of macroseismic intensity, distribution into damage grades under the assumption of a Beta distribution, etc. This method is fundamentally different from the calculation carried out in OpenQuake in terms of PGA/SA, with damage grades directly retrieved from the fragility model, conversion of ESRM20 damage grades into ESM-98 damage grades, etc. Without these details and comparisons, it may not be fully evident to the reader what the purpose of this section is.

   f. Lines 224-225: These sentences compare the values obtained against observations, but the percentages of "heavy" and "very heavy" damage observed are not reported. Please add them in the text. It is also not clear why the observed values are not shown in Fig. 3, given that they are shown later in Figs. 4 and 5 (converting number of buildings into proportions, as in the other plots, or using a right-hand axis with a different scale on the same plot).

   g. Do the OpenQuake damage results correspond to the average damage resulting from all 1,000 ground motion realisations (only mentioned in Section 2.1) and all logic tree branches (if a ground motion logic tree was indeed used)? Please specify.

   h. Does Armagedom calculate different ground motion fields (1,000 as well?) to account for ground motion uncertainty?

17. Line 240: To my knowledge, the most recent reference of GED4ALL is Silva et al. (2022), and the preferred name for this building taxonomy is "GEM Building Taxonomy v3.0":

   Silva V, Brzev S, Scawthorn C, Yepes C, Dabbeek J, Crowley H (2022) A building classification system for multi-hazard risk assessment. International Journal of Disaster Risk Science 13:161–177. https://www.doi.org/10.1007/s13753-022-00400-x

18. Line 240: I would suggest to re-phrase "we selected a GED4ALL building class based on…" as "we defined building classes in terms of the GEM Building Taxonomy v3.0 (Silva et al., 2022), based on the building materials and the number of storeys". The current phrasing may erroneously convey that the taxonomy consists of a pre-defined list of building classes to choose from, instead of a classification system of attributes to be concatenated.

19. Line 245, Table 3-4: It is interesting that fragility models for infilled frames ("CR_LFINF") were selected for dual frame-wall systems ("CR/LDUAL"), instead of using the "CR_LDUAL" fragility models directly (one of which is mentioned in Table 3-3). Please comment in the paper on this choice. Moreover, the reinforced concrete ESRM20 classes selected correspond to different values of the lateral force coefficient, and it is not clear how this could be selected from the damage dataset. Please comment.

20. Lines 249-254: Please specify the GMPE used.

21. Lines 254-256: The label "SM – brgm $V_{S30}$" suggests that the BRGM model was used together with the USGS ShakeMap. How was this site model incorporated to the ShakeMap? Does this mean the ShakeMap used in the paper is not the one downloaded from the USGS but the authors have run the ShakeMap software themselves? Please clarify in the manuscript.

22. Line 283 (Fig. 4) and Line 341 (Fig. 5): Please clarify if the proportions of buildings in each damage grade stemming from the calculations have been calculated with respect to the total number of buildings (including undamaged ones) or only the number of damaged buildings (which I understand is the case for the observation values).

23. Line 284 (caption of Fig. 4), and Table A3: Please clarify what the acronym "BRGM/CCR" refers to. I find it confusing that it is named in Fig. 4, which corresponds to analyses carried out using the building-by-building exposure based on the 327 surveyed buildings, and then in Table A3, which lists 2,778 buildings, which is the number reported in both Sections 3.3.1 (line 216, "the exposure model in Armagedom, which includes 2778 buildings") and 3.3.4 (lines 293-294, "the second exposure model ("brgm exp.") is based on national statistical data, and includes 9 centroids with 2778 buildings"). Please clarify the relation between the exposure models used in Sections 3.3.1 and 3.3.4: are they the same? Please add reference to Table A3 within the text.

24. Lines 291-293, and Tables A1 and A2: It is not clear why the ESRM20 exposure model is not being used directly as it is, including its exposure-to-vulnerability mapping. The changes introduced by the authors mean that the calculations carried out with this model may not necessarily reflect what would have been obtained with the "original" ESRM20 model. Moreover, the choice of fragility classes for each exposure class shown in Table A1 appears as contradictory. In the screenshot of Table A1 below, I have marked the differences in the classes and annotated the classes used in ESRM20, which can be consulted in the *esrm20_exposure_vulnerability_mapping.csv* file of the ESRM20 v1.0 repository[5]. The differences are associated with the number of storeys (e.g., a 4-storey class has been selected for a 6-and-above-storey class, first row) and the lateral force coefficient and/or design code level (e.g., a low code class with 15% lateral force coefficient has been selected for a no-code class, seventh row). Please justify the need to use a "simplified" version of the exposure model (instead of the original ESRM20 exposure) and explain the criteria used to assign new classes in Table A1 (in the main body of the paper).

Table A1 Selected ESRM20 fragility classes based on the building types in Le Teil according to the ESRM20

| Original ESRM20 type | N. buildings | Selected ESRM20 frag. class | # class |
|---|---|---|---|
| CR/LDUAL+CDL+LFC:4.0/HBET:6- | 3 | CR_LDUAL-DUL H4 **H6** | 1 |
| CR/LDUAL+CDL+LFC:4.0/HBET:3-5 | 7 | CR_LDUAL-DUL_H4 | 1 |
| CR/LDUAL+CDN/HBET:6- | 2 | CR_LDUAL-DUL_H4 **H6** | 1 |
| CR/LFINF+CDL+LFC:4.0/H:2 | 67 | CR_LFINF-CDL-10 H2 **CDL-5** | 2 |
| CR/LFINF+CDM+LFC:4.0/H:1 | 42 | CR_LFINF-CDM-10 H2 **CDM-5_H1** | 3 |
| CR/LDUAL+CDM+LFC:4.0/HBET:6- | 1 | CR_LDUAL-DUL_H4 **H6** | 1 |
| CR/LFLS+CDN/HBET:6- | 9 | CR_LFINF-CDL-15 H4 **CDN-0_H6** | 4 |
| CR/LFINF+CDL+LFC:4.0/H:1 | 76 | CR_LFINF-CDM-10 H2 **CDL-5_H1** | 2 |
| CR/LDUAL+CDM+LFC:4.0/HBET:3-5 | 3 | CR_LDUAL-DUL_H4 | 1 |
| MUR+CL/LWAL+CDN/H:2 | 378 | MUR-CL99_LWAL-DNO_H2 | 5 |
| CR/LFINF+CDM+LFC:4.0/H:2 | 37 | CR_LFINF-CDM-10 H2 **CDM-5** | 3 |
| MUR+CL/LWAL+CDN/H:1 | 690 | MUR-CL99_LWAL-DNO_H1 | 6 |
| MUR+ST/LWAL+CDN/H:2 | 130 | MUR-CL99_LWAL-DNO_H2 **STDRE** | 5 |
| CR+PC/LWAL+CDN/HBET:3-5 | 53 | CR_LDUAL-DUL_H4 | 1 |
| W/LWAL+CDN/H:1 | 100 | W_LFM-DUL H2 **H1** | 7 |
| W/LWAL+CDN/H:2 | 43 | W_LFM-DUL_H2 | 7 |
| CR+PC/LWAL+CDN/HBET:6- | 1 | CR_LDUAL-DUL H4 **H6** | 1 |
| CR/LFINF+CDN/HBET:3-5 | 38 | CR_LFINF-CDL-15 H4 **CDN-0** | 4 |
* * *
[5] https://gitlab.seismo.ethz.ch/efehr/esrm20/-/blob/v1.0/Vulnerability/esrm20_exposure_vulnerability_mapping.csv

25. Lines 291, 294: Please clarify in the manuscript that only residential buildings from the ESRM20 exposure model are being included in the calculation (I have deduced this from looking at the ESRM20 exposure model for France). Please clarify as well if the BRGM exposure considers only residential buildings as well, and whether it covers the same spatial extent (even better if using a map). Please clarify if the damage observations only cover residential buildings as well.

26. Lines 300-304: By using a weighting scheme for the so-called "ESHM $V_{S30}$" model but not for the BRGM model, this comparison becomes not just about the $V_{S30}$ models but the different ways of assigning values to an aggregated area. It would be useful to highlight this further in the text.

27. Line 310, Table 3-5: The table shows 8 locations but the text (line 294) says "9 centroids". Please correct where needed.

**Language Use, Typos**

Please make the following changes.

1. What do the authors mean with "ShakeMap analyses"? It seems to me that, in most cases, the authors simply mean "ShakeMaps". Please revise and re-phrase all instances along the paper. Examples:

   a. Line 14: Just "ShakeMaps in order to…".

   b. Line 49: Just "to distributions given by ShakeMaps".

2. Line 10: "validated individually, *although* testing and validating".

3. Line 12: "*damage from* past earthquakes".

4. Line 15: "components of the 2020 *European Seismic Hazard Model*" (not "Euro-Mediterranean").

5. Line 16: "the *degree* of damage" or "the *damage grade*".

6. Line 22: "insured and uninsured *direct economic* losses". I assume this was the intention, as only indirect economic losses are mentioned otherwise.

7. Line 23: "(PSHA, *PSRA* are…" (not "PSHR").

8. Line 53: Please define $V_{S30}$ in its first appearance (this line).

9. Line 77: "*v*ulnerability *c*lasses" (small letters).

10. Line 93: "data in the *forms that we used are*" (no commas).

11. Line 101, Table 2-1: "Vertical load-bearing" and "Horizontal load-bearing" (not "loads").

12. Line 115: "the ruptures in the *ShakeMap* as well as".

13. Line 121: "scaling *relation*".

14. Line 123: "we assume that its *geometric centroid* is located at the hypocentre".

15. Line 131, Table 3-1: In the caption, "Rupture *parameters associated with* the five source models".

16. Line 156, Fig. 1, caption: "ground motion intensity measures *aggregated from* all exposure centroids".

17. Line 164: "to identify the *ruptures* leading to".

18. Line 168: The equation starts "MCS =" but the subscript of the standard deviation says "MMI". Is this correct? (See line 170 as well).

19. Line 177: "The *CA2015* model".

20. Line 201: "(FM2010), *and* b) the macroseismic intensity".

21. Line 203: "PGA given by  the ground motion-to-intensity".

22. Line 210, caption: "at the exposure centroids *of the BRGM exposure* in the site models…" (or appropriate name for the exposure model).

23. Line 350: "closer to the estimation *of EMS-98 macroseismic intensity* by Schlupp et al. (2022)". The text before that statement had not yet mentioned macroseismic intensity.

**Issues with References**

1. Line 384: There are numbers at the end of "Munson" and "Stamatakos".

2. Lines 396-397: The citation of Crowley et al. (2021) is incomplete (no initials of first names, no DOI, mention of EFEHR Technical Report 002 missing). Please cite as (apply journal formatting style):

Crowley, H., Dabbeek, J., Despotaki, V., Rodrigues, D., Martins, L., Silva, V., Romão, X., Pereira, N., Weatherill, G. and Danciu, L., 2021. *European Seismic Risk Model (ESRM20)*, EFEHR Technical Report 002, V1.0.1, 84 pp, https://doi.org/10.7414/EUC-EFEHR-TR002-ESRM20

3. Lines 408-411: The citation of Danciu et al. (2021) is not fully correct. Please cite as :

Danciu, L., Nandan, S., Reyes, C., Basili, R., Weatherill, G., Beauval, C., Rovida, A., Vilanova, S., Sesetyan, K., Bard, P.-Y., Cotton, F., Wiemer, S., and Giardini, D.: *The 2020 update of the European Seismic Hazard Model: Model Overview*, EFEHR Technical Report 001, V1.0.0, https://doi.org/10.12686/A15, 2021.

---

## Referee Comment (RC2)

**Review of Manuscript egusphere-2023-1740**
Testing the 2020 European Seismic Hazard and Risk Models using data from the 2019 Le Teil (France) earthquake

The manuscript is a research study devoted to carry out a testing and validation study of components involved in the seismic hazard and seismic risk estimation. The testing of ground motion and damage to building is done using several models, observations of ground shaking and observed damage from past earthquakes. The authors investigate if the obtained scenarios are consistent with observations and the reason for the obtained differences.

The topic of the paper is very interesting and suitable for the readers of the journal. However, the title and the redaction of the manuscript do not help to get this goal. The focus on European Seismic Hazard and Risk Models distracts from the very interesting part of the manuscript. The manuscript should be focused as a sensitivity study of the ground motion estimation and damage estimation using different input models and how these are closest or not to the observed data from Le Teil earthquake.
Therefore, each section must be introduced with the models that are going to be compared, why are those comparisons going to be done in that section?. Additionally, each comparison must be explained more in detail so the reader can see clearly which models are kept constant and which are compared. Finally, the author must try to rewrite the conclusions according to the comparisons they are doing. My final recommendation is to reconsider the publication of the manuscript after major revisions.

**MAIN COMMENTS**

The concept ShakeMap analysis is not clear. The authors cite Wald et al. 2022, but they should explain better.

**Line 62.** When describing the earthquake, you have to indicate also the registered magnitude and focal depth. Also, they indicate a estimated near-faults PGAs with a 68% confidence interval of 0.3-1.9g . Is this a range in the rupture area? Which is the size of the rupture? How can you explain such a high attenuation because the at 15 km the recorded PGA was only 0.04 g (that is a reduction of 77% of the PGA in 15 km if compared with 0.3g).

**Line 75.** Do not use number for macroseismic intensity, it is better to say VII-VIII instead 7-8

**Line 110.** Regarding the test based on the intensity of the seismic ground motion.
The authors compare the different scenarios pointing that the lowest PGA and Sa0.3s must be due to differences in the rupture distance but they do not say anything about which scenarios is closest to the observed ground motion. Which models fit better the observations?

**Line 160.** Regarding the test based on the macroseismic intensity.

I do not understand what the authors are trying to demonstrate. If you are using correlations from Ground Motion to Intensity the results that you are going to obtain should be similar to the obtained in the previous section. If the idea is to see which is the best GMICEs for the region, then using only those scenarios is not enough, the authors should look for the most recent correlation (using a higher number of observations ground motions and macroseismic intensity) and simply use that relationship with the corresponding standard deviation and probably the observed intensity at Le Tail will be in that range.

**Line 209.** Estimation of damage using different risk analsys tools

Here the authors compare the damage results using Armagedom and OpenQuake but the section should be explained better. As far as I understand the damage obtained with Armagedom is obtained using the ground motion modelled by the deterministic scenarios (all of used in the previous sections?, one of them?) and the semi-emprical macroseismic method, but regarding Openquake the authors indicate the use the ESHM20 ground motion logic tree (is this meaning you are comparing damage using a deterministic scenarios with damage from a probabilistic hazard map? It sound strange to me. Can you clarify? Which is the method used in OPENQUAKE for the damage estimation is also the same used in Armagedom? Is it a different method? You have explained how this is done to be sure that you can compare the results.

**Line 237.** Regarding the Damage based on observations.

Again, this is rather difficult to understand. The paragraph starts speaking about test related to vulnerability and risk modelling, but the conclusion of the paragraph is simply a table assigning building taxonomies to the building database. If the author wants to create different taxonomies to their database, they should name the section: Vulnerability estimation or something related to that.

**Line 248.** Regarding Estimated damage based on a "building-by-building"

Here the authors, compare the building-by-building damage results using OPENQUAKE when using Ritz et al. scenario and Shakemap analysis (try to find a better name for this). Initially those analysis use the same Vs30 model and they also include a new Vs30 model (named ESHM20 Vs30) to the Ritz et al. scenario. Again, this is very messy. If you want to compare the influence of the ground motion scenario, it is clear the comparison between Ritz and Shakemap using the same Vs30 model but if you want to compare the Vs30 influence you should also include the Shakemap scenario with the ESHM20 Vs30 model to be consistent.

**Line 287.** Regarding Estimated damage based on aggregated exposure model.

Here the authors carry out many different comparisons. Again, it is very messy, and it is not clear why you are doing it and what are you looking for.

**Conclusions:** The first conclusion is that the FM2010 model is the best to estimate macroseismic intensity since it is closer to Schlupp et al. (2022). Is this the model used in your national seismic hazard maps or shakemaps to convert from ground motion to macroseismic intensity? Is it only appropriate for the Le Teil region?

Along the paper you have made multiple comparison, so it would be nice if the conclusions also indicate the main conclusion about those comparisons. At the moment, 11 lines are conclusions regarding the ground motion comparisons (sections 3.1 and 3.2) and 11 lines are conclusions regarding the rest of comparisons (3.3.1 to 3.3.4).

---

## Author Response (AR1)

Response to reviewer comments
Preprint egusphere-2023-1740
https://doi.org/10.5194/egusphere-2023-1740
**7    Reviewer 1**

Review of Manuscript egusphere-2023-1740
Testing the 2020 European Seismic Hazard and Risk Models using data from the 2019 Le
Teil (France) earthquake
This manuscript presents a comparison between building damage states as observed in the
field after the 2019 Le Teil earthquake and those calculated by means of combining different
components of existing risk models from different sources (not just the 2020 European
Seismic Hazard and Risk Models). The damage survey has been processed by the authors
according to expert judgment to obtain damage in terms of the EMS-98 scale. Different
rupture models from the literature, as well as the USGS ShakeMap for this earthquake, are
used to generate several realisations of ground motion fields in terms of peak ground and
spectral acceleration (PGA, SA). The PGA values are then converted to macroseismic
intensities using conversion equations, and these macroseismic intensities results are
compared against the 7.5 value obtained in existing literature from field surveys with the
purpose of selecting one rupture model to be used for the subsequent damage calculations
carried out using the OpenQuake engine. Three main comparisons in terms of damage are
carried out, combining different components (e.g., exposure, fragility, site effects) of different
risk models as well as different risk calculation methods/software, and contrasting them
against the results of the processed damage survey of the 2019 Le Teil earthquake.
While the work presented in this manuscript is of interest to the research community to
understand how different existing models and modelling choices affect the calculated
damage and, most importantly, how the calculated damage compares against observations
from a real earthquake, the manuscript has many significant shortcomings that would need to
be addressed before it can be published in NHESS. I thus recommend that the manuscript
be reconsidered for publication after major revisions.
We thank the reviewer for their constructive and helpful comments. We have tried to address
them to the best of our knowledge, as detailed below.
**40   Reviewer 1 - Main Comments**

1. In my view, the title of the paper does not accurately describe its contents, due to three
main reasons:
1.I. The word "testing" is being used loosely throughout the manuscript (see point 2
below).
Review round 1 reply
We agree with your comment (i.e., leaving the word "testing" to the context of actual
statistical tests), and we will revise the manuscript accordingly, by replacing the word
"testing" by "comparison" or "evaluation" wherever it is applicable.
Description of revision

The terms "test" and "testing" have been replaced throughout the manuscript.

1.II. The paper makes comparisons using a variety of sources of model components (exposure, fragility, ruptures) that are not just from the 2020 European Seismic Hazard and Risk Models (ESHM20, ESRM20). The ground motion model used and labelled as being the ESHM20 one does not seem to be the model actually implemented in ESHM20 but a previous version. When using the ESRM20 exposure model the building classes are "simplified", effectively changing the ESRM20 exposure model. To my understanding (as such an outline is missing in the introduction), three comparisons in terms of damage are carried out:

1.II.1) Section 3.3.1: Comparison between (a) damage calculated with the Armagedom software, using the vulnerability index approach, EMS-98 vulnerability classes, and an in-house exposure model, and (b) damage calculated with OpenQuake, using fragility models from the European Seismic Risk Model 2020 (ESRM20) selected to be equivalent to the EMS-98 vulnerability classes, and the in-house exposure model converted onto ESRM20 building classes.

1.II.2) Section 3.3.3: Comparison between (a) damage processed from the field survey, (b) damage calculated using the USGS ShakeMap, (c) damage calculated with OpenQuake, (seemingly) using the Kotha et al. (2020) GMPE (not the version used in ESHM20/ESRM20), and the BRGM VS30 model (which I infer is the ESRM20 VS30 model derived from geology, not used in ESRM20), and (d) the same as (c) but using the ESRM20 VS30 model derived from topography (used in ESRM20 for cratonic and subduction areas, but not for shallow crustal areas, which is the case of France). All cases use the same exposure model, a building-by-building model based on the individual buildings from the damage survey to which ESRM20 building classes were assigned by the authors. All cases use the ESRM20 fragility models.

1.II.3) Section 3.3.4: Comparison between (a) damage processed from the field survey and (b through g) six combinations of the following components:

1.II.3.i. Exposure models: (i) the ESRM20 aggregated exposure model defined by administrative unit (one administrative unit), but with a large modification to the building classes that makes it different from the ESRM20 exposure model, and (ii) an in-house model derived from statistical data (8 or 9 centroids), to which ESRM20 building classes were assigned.

1.II.3.ii. Site models: (i) the BRGM VS30 model (which I infer is the ESRM20 VS30 model derived from geology, not used in ESRM20), values retrieved for the centroid of the administrative unit or 8-9 points of the exposure models, and (ii) the ESRM20 VS30 model derived from topography (used in ESRM20 for cratonic and subduction areas, but not for shallow crustal areas, which is the case of France), with the value for the ESRM20 exposure being a population-weighted average of the whole administrative unit and the values for the inhouse exposure model being retrieved from the 30 arc-sec cell that contains each of the 8- 9 points.

1.II.3.iii. Ground motions: (i) the USGS ShakeMap, and (ii) calculated with OpenQuake using the Kotha et al. (2020) GMPE (not the version used in ESHM20/ESRM20). As can be seen, no "pure" components of ESHM20/ESRM20 appear to have been being used ("pure" = exactly as they have been used in the ESHM20/ESRM20 models) and several components from other sources are

| 109 | being used as well. The title should reflect that the models being compared come |
| 110 | from a variety of sources and decisions from the authors. |

| 112 | 1.III. Finally, "testing […] hazard and risk models" may be misleading, as it can be easily |
| 113 | interpreted as testing the full probabilistic seismic hazard and risk models (i.e., |
| 114 | probabilities of exceedance of ground motion, average annual losses, etc.), which is not |
| 115 | what is done in the paper (and, furthermore, cannot be done using data from one single |
| 116 | earthquake). |

To sum up, the paper shows comparisons (no statistical tests) of observed damage against
damage calculated using components of risk models from different sources.I believe it is
fundamental that a new title be assigned to the manuscript, taking into consideration the
comments above.

Review round 1 reply
Indeed, there are no "pure" components of ESHM20/ESRM20 that have been used, and there
are no statistical tests in the manuscript. We will revise it according to comments 1.I-1.III.

We propose a new title for the manuscript:

"Comparing components of the 2020 European Seismic Hazard and Risk Models using data
from the 2019 Le Teil (France) earthquake"

Description of revision
The title of the manuscript has been revised.

2. I have found the word "testing" being used loosely throughout the manuscript as a
synonym of "comparing", "validating", "verifying", "carrying out quality assurance", etc. The
word "testing" usually implies a formal statistical procedure using statistical indicators of
goodness of fit, similarity between distributions, etc., which are not what is presented in the
paper. The paper mostly carries out comparisons, without quantifying differences across
different models/components. Please avoid over-using and overstretching in meaning the
word "testing", rewording where necessary. Some outstanding examples:

| 145 | 2.a. The title in itself. The European Seismic Hazard and Risk Models are probabilistic |
| 146 | models. The paper uses some of their components to carry out ground motion and |
| 147 | damage calculations that are compared against damage observations from one |
| 148 | earthquake. One earthquake cannot test or validate a probabilistic model, only its |
| 149 | components. |

| 151 | 2.b. Line 34: Bommer et al. (2013) call their work "quality assurance" and not "testing". |
| 152 | Throughout the paper they use the word "check" far more than they use the word "test". |

| 154 | 2.c. Sections 3.1 and 3.2: These sections are not testing ground motions or |
| 155 | macroseismic intensities, they are comparing ground motions and macroseismic |
| 156 | intensities calculated with different rupture models (against one value of macroseismic |
| 157 | intensity) with the purpose of selecting one rupture to use in the remaining comparisons |
| 158 | of the paper. The PGA and SA values are not compared against instrumental |
| 159 | measurements at all (values of PGA are mentioned in lines 64-66 but not marked on the |
| 160 | plots or mentioned again in Section 3.1). The sections are presented as "tests" when, in |
| 161 | reality, they are an intermediate comparative step to select rupture parameters. |

Review round 1 reply

We agree with comments 2 and 2a-2c and we will revise the manuscript accordingly.
Specifically, we will replace "testing" with terms such as "comparison" or "evaluation", and "test"
with "check" or "compare" or a comparable term.
Moreover, the revised manuscript will state that the comparisons in Sections 3.1-3.2 serve the
purpose of selecting rupture parameters.
Description of revision
The manuscript has been revised based on the reviewer's comments and our reply (lines 55).
3. In line with the first point above, and with the purpose of aiding the reader to navigate
comparisons carried out across so many different options, please re-phrase the last
paragraph of the introduction to describe more accurately the work contained in the paper:
3.a. Lines 46-47: This sentence states that the work is done "to test components of the
ESHM20 and the ESRM20" models, giving the impression that only ESHM20 and
ESRM20 components will be used, but components from other models are used as well,
and these are not mentioned at all here. Please mention the other models used.
3.b. Line 48: I suggest not using the expression "scenario simulations" to refer to ground
motion scenarios calculated by means of ground motion models, as the word
"simulations" is usually used to refer to physics-based ground motion simulations (this is
not critical).
Review round 1 reply
The last paragraph of the introduction will be rephrased according to comments 3a-3b.
Lines 46-47: "to compare components of the ESHM20 and the ESRM20 with local site
effects models, exposure models and damage estimation methods,…"
We also plan to expand this paragraph by using the summary made by the Reviewer in
Comment 1.
Line 48: we replace "simulations' by "computations".
Description of revision
The last paragraph of the introduction has been revised. It mentions the other models
used, and it includes a summary of the comparisons in the paper (lines 50-63).
The term "scenario simulations" has been replaced with "scenario computations"
throughout the manuscript.
3.c. Lines 49-50: This sentence may give the impression that "the most compatible
scenario simulation" is selected in terms of the one that gives the results closest to the
USGS ShakeMap, but this is not what is stated in lines 50-52 or in Sections 3.1/3.2 (and
further along in the paper), which show comparisons of all rupture models with respect to
each other (including the USGS ShakeMap) and finally comparing intensities against the
value reported by Schlupp et al. (2022).
Review round 1 reply

Actually, the so-called "USGS ShakeMap" is a shake-map generated by us, using our data (seismic stations measurements, site effect model, specific ground-motion model), with the USGS ShakeMap v4 code.

In order to avoid any confusion, we will use the word "shake-map" (lower case) when it is our own product (although it has been generated using the USGS ShakeMap algorithm), as opposed to the wording "USGS ShakeMap" (trademark product downloaded from the USGS website). We will correct this sentence in order to clarify this.

We will also add a table that summarizes all the shale-maps / ground-motion fields that have been generated:

| GM Map ID | Type | GMM | Site model | Rupture model | Observations |
|---|---|---|---|---|---|
| GM1 | ground-motion field | KothaEtAl2020Site | BRGM soil classes to $V_{S30}$ | Ritz et al. | No |
| GM2 | ground-motion field | KothaEtAl2020ESHM20SlopeGeology | Slope & Geology (ESRM20 data) | Ritz et al. | No |
| GM3 | ground-motion field | KothaEtAl2020Site | ESRM20 $V_{S30}$ data | Ritz et al. | No |
| GM4 | shake-map | KothaEtAl2020Site | BRGM Soil class to $V_{S30}$ | Ritz et al. | Seismic stations |

Description of revision
The "shake-map" is used throughout the revised manuscript to refer to our analyses using the ShakeMap algorithm.
The table above has been added to the revised manuscript (Table 5-1) and the text has been revised (lines 324-328).

3.d. Lines 49-52: The meaning of "the most compatible scenario simulation" and "the most plausible scenario simulation" is not clear. After reading the paper, I believe the authors mean "the most compatible earthquake rupture", or "the earthquake rupture that leads to the most compatible macroseismic intensities".

Review round 1 reply
Thank you for suggesting a clear and precise term. We will use it to revise the manuscript according to comments 3c-3d.

Description of revision
The manuscript has been revised based on your suggestion (lines 69-40).

3.e. Lines 46-54: While several sentences are dedicated to explaining the comparison of ground motions and macroseismic intensities (which is only a preliminary step to select a suitable rupture to carry out the damage comparisons), very little is said about the core of the work. Please consider delineating the content of the three damage comparisons in a similar fashion to what I have written above under point (1), or perhaps with a figure. This is relevant to help the user navigate the paper, as so many different considerations/decisions are being made in each case.

Review round 1 reply

Yes, we will do so based on your comments under point 1. We will also add a figure to
summarize the various steps and comparisons.

Description of revision
The revised manuscript delineates the content of the damage comparisons (lines 55-83;
text added in response to comments 3a-b).
The revised manuscript includes also a figure (Figure 1), which summarizes the various
steps leading up to the comparisons.

4. The authors state (lines 113 and 315) that they are using the Kotha et al. (2020) ground
motion prediction equation (GMPE) in the form of its KothaEtAl2020Site implementation in
OpenQuake. However, all ESHM20/ESRM20 sources indicate that this is not the final GMPE
used in ESHM20 and ESRM20. This being the case, the KothaEtAl2020Site GMPE should
not be labelled as "ESHM20 GMF" (e.g., line 314), as this can be misleading for the reader.

A more fundamental implication is that, with this GMPE being used, it is not the ESHM20
ground motion model that is being "tested", as implied in the title. Weatherill et al. (2020) and
the ESHM20 report (Danciu et al., 2021) explain that a series of modifications were
introduced to the Kotha et al. (2020) GMPE for the implementation in ESHM20 and ESRM20.
Fundamentally, and given that the authors of the present manuscript emphasise the
comparison of different VS30 models, KothaEtAl2020Site has a different amplification
function for site effects, and the site-to-site variability of the GMPE was calibrated only on
measured VS30, which means that an incompatibility arises when using it with inferred
values of VS30. As explained in the OpenQuake documentation[1]:

4.a. KothaEtAl2020Site is a "preliminary adaptation of the Kotha et al. (2020) GMPE
using a polynomial site amplification function dependent on Vs30 (m/s)".

4.b. KothaEtAl2020ESHM20 is an "adaptation of the Kotha et al. (2020) GMPE for
application to the 2020 European Seismic Hazard Model, as described in Weatherill et al.
(2020)". Page 89 of the ESHM20 report (Danciu et al., 2021) explains that
KothaEtAl2020ESHM20 is the GMPE used in ESHM20. Site effects in this
implementation depend on VS30 and whether that VS30 is a measured quantity or
inferred from proxies (e.g., slope), so as to account for the uncertainty associated with
using inferred values. Page 69 of Danciu et al. (2021) specifies that ESHM20 refers to
ground motions on the "reference rock" (VS30 of 800 m/s everywhere). The ESHM20
logic tree input file[2] also shows that KothaEtAl2020ESHM20 is being used for the
calculations.

4.c. KothaEtAl2020ESHM20SlopeGeology is an "adaptation of the ESHM20-
implemented Kotha et al. (2020) model for use when defining site amplification based on
slope and geology rather than inferred/measured Vs30". The ESRM20 logic tree input
file[3] and its "cut" version used for shallowcrustal areas when comparing against past
earthquakes[4] indicate that this is the GMPE used in ESRM20 to calculate losses. Site
effects in this implementation depend on slope and geology, not VS30 (e.g., second
* * *
[1] https://docs.openquake.org/oq-
engine/master/reference/openquake.hazardlib.gsim.html#openquake.hazardlib.gsim
[2] https://gitlab.seismo.ethz.ch/efehr/eshm20/-
/blob/master/oq_computational/oq_configuration_eshm20_v12e_region_main/gmpe_complete_logic_t
ree_5br.xml
[3] https://gitlab.seismo.ethz.ch/efehr/esrm20/-
/blob/main/Hazard/gmpe_logic_tree_5br_slope_geology.xml
[4] https://gitlab.seismo.ethz.ch/efehr/esrm20_scenario_tests/-
/blob/main/models/esrm20/GMPE/gmpe_logic_tree_5br_shallow_default.xml paragraph of Section 3.2 of the ESRM20 report, page 16). ESRM20 uses this model together with the slope and geology of the ESRM20 model, which can be retrieved with the "exposure-to-site" tools cited in the present manuscript.

As a consequence, reference (, not VS30), as in ESRM20. One should also note that using KothaEtAl2020ESHM20 with VS30 values other than 800 m/s would not necessarily be representative of either the ESHM20 or ESRM20 models.

Review round 1 reply
In the revised manuscript, we will now apply the correct KothaEtAl2020ESHM20SlopeGeology GMM when applying the "ESHM20 model", according to Comment 4. However, we will also apply the KothaEtAl2020Site when using the Vs30-based site effect model available at BRGM. These differences will be detailed in the Table above (answer to Comment 3).

As far as Comment 4.c is concerned, the ESRM20 uses a collapsed version of the ESHM20 source model logic tree for 2 reasons: 1) to avoid high computational costs for calculations with respect to the generation of stochastic event sets and the associated ground motion fields, 2) to avoid undesirable correlations in the source parameters due to the approach for propagating uncertainty, which assigns to all sources the same category of activity rate. In our manuscript, we are assessing damage after a single event. Therefore, no source logic tree is used.

5. Associated with the previous point, I believe it is very important that clarity is added with respect to the site models used in the comparisons. When comparing against Weatherill et al. (2023) (cited by the authors) and the ESHRM20 documentation, the explanations (e.g., lines 266-272) in the paper lack from some clarity:

    5.a. It is not fully clear what the "BRGM's VS30 database" refers to, as there are two
    VS30 models in the cited reference Weatherill et al. (2023): one based on topography
    alone, and another based on geology alone. The ESRM20 exposure-to-site tools (which
    the authors use and cite in the present manuscript) return the VS30 values from the
    topography-based model, as the comparisons in Weatherill et al. (2023) showed that it
    performed better than the geology-based one. As Table 3-5 (line 310) shows different
    VS30 values for the two (and quite round values for the BRGM case), I infer that the
    "BRGM's VS30 database" refers to the geology-based VS30 model presented in
    Weatherill et al. (2023). Please clarify in the manuscript.

    Review round 1 reply
    We apologize that there has been a confusion regarding the reference and origin of the
    "BRGM's VS30 database". The model that we used in the manuscript is an EC8 soil class
    map assembled at BRGM for the French territory: this map of soil classes has then been
    converted into a Vs30 map by taking the median value of each EC8 soil class. The
    associated reference is a BRGM report (Roullé & Monfort, 2016), where the map is based
    on local knowledge of geology and soil classes. It is not linked to the Weatherill et al. (2023)
    reference. We will add some sentences to clarify this aspect.
    Associated reference:
    Monfort, C., & Roullé, A. (2016). Estimation statistique de la répartition des classes de sol
    Eurocode 8 sur le territoire français - Phase 1 : Rapport final. BRGM Report RP-66250-
    FR.

    Revision description
    The creation of the site model based on the soil classes in Monfort and Roullé (2016) is
    described in the revised manuscript (lines 286-290).

5.b. The manuscript would benefit from adding some sentences regarding the resolution
of each of the two models, as this is relevant for the reader to understand what is being
compared (e.g., in lines 267- 272). From Fig. 7 of Weatherill et al. (2023) it looks like in
the "BRGM's VS30 database" there are three geologic units, associated with three
ranges of VS30 values (is the uncertainty being sampled to assign values in the paper?).
The "point" workflow of the ESRM20 exposure-to-site tool returns the values associated
with the 30-arcsec cell to which the target point belongs, as 30-arcsec is the resolution of
the model.
Review round 1 reply
The resolution of the BRGM Vs30 model is based on a geological map at the (1/50000
scale). We will add a sentence comparing this value to the resolution of the ESRM20
exposure-to-site tool (30-arcsec).
Description of revision
The revised manuscript mentions the resolutions of the two models (line 288).
5.c. It is noted that the VS30 values returned by the exposure-to-site tool are not used in
ESRM20 in France (non-cratonic shallow seismicity). These VS30 values are used with
the craton and subduction GMPEs selected for the areas of Europe where the shallow-
crustal ESHM20 GMPE (i.e., KothaEtAl2020ESHM20SlopeGeology) is not applicable
(e.g., see page 16 of the ESRM20 report, Crowley et al., 2021). The GMPE used for
ESRM20 (i.e., KothaEtAl2020ESHM20SlopeGeology in OpenQuake) calculates site
amplification based on slope and geology directly, not VS30. Please clarify in the
manuscript that the VS30 values labelled as ESHM20 are actually not used in
ESHM20/ESRM20 in France.
Review round 1 reply
We agree with this comment: as stated above, we will now use the
KothaEtAl2020ESHM20SlopeGeology GMM to represent the "ESHM20 model" (GM2 in
the above table). As a result, we will use the ESRM20 slope and geology data directly.
For the generation of other ground-motion scenarios, we will still use the BRGM Vs30
model for France associated with the KothaEtAl2020Site GMM in order to be consistent.
We will clarify this in the revised manuscript.
Description of revision
The Sections 3.3 and 5.1 describe how this GMM is used in the revised manuscript.
d. From my understanding, the site amplification model and VS30 maps are part of
ESRM20 and not ESHM20, as ESHM20 focused on hazard on the reference rock.
Please name them as ESRM20, not ESHM20.
Review round 1 reply
We will rename them accordingly in the revised manuscript.
Description of revision
The revised manuscript uses the name ESRM20 $V_{S30}$ instead of ESHM20 $V_{S30}$.
6. In my view, it is necessary to add a map that shows the resolution/locations of the different
exposure models and site models, the spatial extent of the municipality of Le Teil, the
location of the selected rupture plane, etc. This is important for the reader to be able to understand the different models that are being compared and interpret the differences observed.

Review round 1 reply
Thank you for suggesting this. Such maps will be added to the revised manuscript.

Description of revision
A map has been added to the manuscript (Figure 2).

7.a The conclusions section is too short and does not discuss the results with depth. It only focuses on marginal observations. It consists of three paragraphs, the first (and longest) of which focuses extensively on the comparison of macroseismic intensities (which is not the core of this work), the second of which briefly mentions that the exposure model was a key difference-maker in the results, without elaborating on reasons, and the third paragraph discusses potential improvements to the analysis by changing the criteria used to post-process the field damage survey, highlights the need for more standardised field survey practices, and comments about the importance of accounting for buildings not included in the survey, which has not been discussed in the paper and for which explanations are not given. Please rewrite the conclusions focusing on the large number of different model components that have been compared, to reflect the work done.

Review round 1 reply
The conclusions will be revised based on this comment. Thank you for your comment and your guidance. The points around which the conclusions will be revised:

- The comparison of macroseismic intensities, as well as the other comparisons will be discussed in the conclusions;
- The effect of the exposure model on the results will be discussed in terms of the number of estimated damages, and in terms of the included building classes and their fragility;
- The effect of accounting for buildings not included in the survey will be discussed in the manuscript and in the conclusions.

Description of revision
The conclusions have been revised.

7.b I have found the statement about the effect of the exposure model (lines 359-362) quite hard to see in Fig. 5, which shows so many different models. Moreover, lines 323-333 focus on the differences due to the VS30 model, not the exposure. I strongly recommend to find alternative ways to show and compare these results (perhaps several plots "grouping" results according to exposure, or VS30), and potentially even to quantify the differences between models, so that it becomes clearer to the reader whether exposure or site effects have had a greater influence in the discrepancies with observed values.

Review round 1 reply
Once more we would like to thank you for your comment and your guidance. In the revised manuscript, we will describe the effect of the different exposure models in Section 3.3.4. We will add different plots, which will group results by exposure or $V_{S30}$. Moreover, the differences between models will be quantified by selecting one case as the reference, and by subsequently calculating the ratio of the probability of a damage grade in the other cases to the probability of a damage grade in the reference case.

Description of revision

In the revised manuscript, the effect of the exposure and the $V_{S30}$ models are discussed with
the help of figures 6 and 7, respectively.

7.c The importance of including in the calculations buildings that were not part of the damage
survey is mentioned in the conclusions (lines 368-369), but I cannot find it discussed before.
Please explain why it is important to include those buildings and comment on why the
damage survey seems to cover such a small proportion of the buildings of the municipality of
Le Teil. Did they only survey buildings on demand from the owner? Can it be assumed that
the rest of the buildings were undamaged? This is important as well to interpret the plots in
Fig. 5.

Review round 1 reply
Indeed, surveys were done upon requests from the owners. Because of this, there is a potential
bias in the damage distribution based on the observations. On the other hand, it cannot be
guaranteed that the rest of the buildings were undamaged. This issue will be discussed in the
revised manuscript. Thank you very much for raising this point.

We propose to add a table that will clarify the way the buildings have been surveyed:

| Observed Damage Data ID | Exposure resolution | Exposure data | Damage estimation method | Damage conversion method | |
|---|---|---|---|---|---|
| DD1 | Building-by-building (327 buildings) | AFPS emergency survey | AFPS emergency observations on 327 buildings (Green/Yellow/Red tags) | Conversion to EMS-98 damage grades (Tab. 2.1) | Related to Fig. 4 |
| DD2 | Infra-municipality districts (2778 buildings) | National statistics database (BRGM-CCR) | AFPS emergency observations on 327 buildings (Green/Yellow/Red tags) + "Extrapolation" | Conversion to EMS-98 damage grades with expert judgment (Tab. 3.6) | Related to Fig. 5 |
| DD3 | Infra-municipality districts (2778 buildings) | National statistics database (BRGM-CCR) | AFPS emergency observations on 327 buildings (Green/Yellow/Red tags) + "Extrapolation" | Conversion to EMS-98 damage grades (Tab. 2.1) + Bias adjustment on total number of 2778 buildings (accounting for non-surveyed buildings) | Related to Fig. 5 |

Description of revision
The table above has been added to the manuscript (Table 2-3). Moreover, importance of
including in the calculations buildings that were not part of the damage survey is explained in the revised manuscript after the revision with respect to the comment 5 in the section
”Reviewer 1 - Other Comments on Content”.

Apart from this, the first paragraph (lines 350-357) talks extensively about macroseismic
intensities calculated with the AS2000 model. The acronym AS2000 is not defined at all
within the text. Line 354 suggests the AS2000 has been used to convert from SA(1 s) to
macroseismic intensity, and , lines 355- 357 highlight that SA(1 s) is not representative of the
buildings in Le Teil, but Section 3.2 discusses two models that convert from PGA/PGV (not
SA) to macroseismic intensity. I thus infer AS2000 stands for Atkinson and Sonley (2000),
one of the conversion models used by the Armagedom software, according to Sedan et al.
(2013). However, no macroseismic intensity values calculated using the Atkinson and Sonley
(2000) conversion equation are presented in the paper. Please revise and correct as needed.
Review round 1 reply
We apologize for this confusion: the reference to the Atkinson & Sonley (2000) GMICE comes
from a previous working version of the manuscript. Eventually, this GMICE has not been used
in the intensity computations (we confirm that the SA(1s) ground-motion parameter is of little
interest to the studied building stock). The manuscript will be corrected by removing references
to this model.
Description of revision
Any reference to the Atkinson & Sonley (2000) GMICE has been removed.
8. Similarly to the conclusions, the abstract would need a revision to include mention of all
other models that have been used, as per my previous comments. Please revise the last
sentence of the abstract (lines 17-19), which vaguely hints on conclusions that do not match
the conclusions section or the content of the work.
Review round 1 reply
The abstract will be revised so that it takes into account your comments, and the closing
statements will match the content. Thank you for this comment.
Description of revision
The abstract has been revised.

**Reviewer 1 - Other Comments on Content**

1. Line 56: Please remove "and risk" from the title, as the section does not describe seismic risk in the area.

Review round 1 reply
This will be removed from the title. Thank you for this comment.

Description of revision
This has been removed from the mansucript.

2. Lines 70-74: While this statement can be generally valid, it is noted that the ground motion model used in ESHM20 is a backbone model whose central tendency is derived from European data that may be lacking representation of such shallow earthquakes with a relatively large stress drop, but whose different branches account for the possibility of having more "unusual" stress parameters (i.e., uncertainty in the stress drop is treated as an epistemic uncertainty). Please see Kotha et al. (2020) and Weatherill et al. (2020) and consider rephrasing (otherwise it suggests that the authors agree with Causse et al. 2021 in this particular case and believe a priori that the ESHM20 ground motion model cannot be able to represent this earthquake).

Review round 1 reply
The manuscript will be revised according to this comment. We do not wish to express any agreement or disagreement with Causse et al. (2021), only to report their findings. However, we do acknowledge –and the revised manuscript will do so too– that the ESHM20 ground motion model may be able to represent the ground shaking generated by this earthquake. We propose to add the following sentence at the end of the paragraph:

"However, it should be noted that some branches in the ESHM20 GMM logic tree should be able to account for the possibility of having extreme stress parameter values, by treating uncertainty in the stress drop as a source of epistemic uncertainty (Kotha et al., 2020; Weatherill et al., 2020)."

Description of revision
The sentence above has been added to the manuscript (lines 107-109).

3. Line 101, Table 2-1: There are some aspects of the table that would benefit from clarification in the text:

   3.a. How should the reader interpret the first four columns that contain "R" and empty spaces? Does it mean that while a certain parameter is red, the EMS-98 damage grade is as indicated, irrespective of the other parameters? Are the four components ordered as per a hierarchy? I.e. if both vertical and horizontal structural elements are red, then it is damage grade 5, but if the horizontal structural elements are red and the vertical ones are yellow or green, then it is 4?

   Review round 1 reply
   Yes, in the cases where a given parameter is red the damage grade is assigned irrespective of the other parameters.

   Yes, the four components are ordered hierarchically. Yes, if both vertical and horizontal structural elements are red, then the damage grade 5 is assigned, but if the horizontal structural elements are red and the vertical are yellow or green, then the grade 4 is
assigned.
We will add this clarification in the revised manuscript.
Description of revision
The clarification above has been added to the manuscript (line 143).
3.b. The far right column shows all components in green and the damage grade resulting
in 1. Is this because all entries in the survey have some sort of damage and thus "green"
is to be interpreted as "damaged, but usable" and not include "undamaged"? It calls the
reader's attention that everything is green and the damage grade is not zero. Please
comment in the paper.
Review round 1 reply
Indeed, in the cases where everything is green, the damage grade 1 is assigned (damage
grade 1 corresponds to no structural damage and slight non-structural damage). This
assignment is done based on our judgement. The dataset that we used contains only
damage observations, which were made during inspections on request by the building
owners. We consider that slight non-structural damage was the cause that led the owners
to request an inspection of their building. We will add this clarification in the revised
manuscript.
Description of revision
The clarification above has been added to the manuscript (line 144).
4. Line 106, Table 2-2:
4.a. In the caption, please clarify this is the buildings' "final" tag (as opposed of tags by
components). "… as a function of the buildings' final tags for the entire dataset".
Review round 1 reply
This will be corrected in the revised manuscript. Thank you for this comment.
Description of revision
This has been corrected.
4.b. It calls my attention that several green buildings end up classified as ESM-98
damage grade 3, which corresponds to moderate structural damage and heavy non-
structural damage. I would expect moderate structural damage to lead to the need of
further inspection and repair before the building can be used, while "green" means that
the building can be used again immediately. This could be the reason why in Fig. 4 the
"observation based" probabilities for damage grade 2 are notably low when compared
against damage grades 1 and 3 (the distribution has an unusual "valley" in damage grade
2). Can it be that several of the green buildings that ended up classified as damage grade
3 are, actually, damage grade 2? Moreover, Table 3-6 suggests the authors also believe
green should map only to damage grade 1 or 2.
Review round 1 reply
This is a very good point, and we agree with this comment. Indeed, there may be green
buildings, which could have been assigned a damage grade 2. The classification that we
propose assigns damage grade 3, when the vertical or the horizontal structural elements have a yellow tag. We believe that a yellow tag with respect to the structural elements
signifies moderate structural damage, hence damage grade 3. The fact that in these cases
a green tag was assigned, perhaps indicates that a further inspection took place, which
either reclassified the damage as green structural damage, or as yellow non-structural
damage. We acknowledge that our mapping scheme can be refined to take into account
such cases.
The "valley" in damage grade 2, which you refer to, will be discussed in the revised
manuscript based on your comments and this response.
Description of revision
The revised manuscript comments on the "valley" in damage grade 2 (line 456).
5. Associated with the previous point, there seem to be different probabilities of damage and
numbers of damaged buildings from observations presented in different plots and the text,
which I have found confusing. I have found/observed:
5.a. The probabilities of damage from observations differ in Fig. 4 with respect to Fig. 5.
Review round 1 reply
Thank you for raising this issue. Please accept our apologies for omitting the calculation of
the probabilities in Fig. 5.a labelled as "Observation-based". These probabilities take into
account the probabilities in Fig. 4 as well as our presumption that the damage grade
probabilities for the buildings that have not been inspected are different, because the
inspections were made upon owner request. The calculation of the probabilities in Fig. 5.a is
done with the following tables (Tables 5.a.1-4). Table 5.a.1 includes the probabilities of the
damage grades conditioned on colour tags. In Table 5.a.2, the total probabilities of the
damage grades is calculated. Table 5.a.3 gives the damage grade probabilities conditioned
on whether a building has been inspected. The first line of Table 5.a.3 includes the
probabilities based on the damage observations. The second line includes values selected
based on our judgement. The calculation of the total probabilities of the damage grades for
inspected and uninspected buildings, which are the probabilities in Fig. 5.a labelled as
"Observation-based", is given in Table 5.a.4. The description of this calculation as well as
Tables 5.a.1-4 will be included in the revised manuscript.
Table 5.a.1: Probabilities of the damage grades conditioned on the colour tag assigned to a
building that has been inspected during the survey

| tag | n_buildings | P(tag) | P(DG1|tag) | P(DG2|tag) | P(DG3|tag) | P(DG4|tag) | P(DG5|tag) |
|---|---|---|---|---|---|---|---|
| Green | 238 | 0.475 | 0.610 | 0.150 | 0.240 | 0.000 | 0.000 |
| Yellow | 157 | 0.313 | 0.000 | 0.000 | 0.900 | 0.080 | 0.020 |
| Red | 106 | 0.212 | 0.000 | 0.000 | 0.000 | 0.640 | 0.360 |

Table 5.a.2: Calculation of the total probability of the damage grades for buildings inspected
during the survey

| tag | P(DG1|tag)·P(tag) | P(DG2|tag)·P(tag) | P(DG3|tag)·P(tag) | P(DG4|tag)·P(tag) | P(DG5|tag)·P(tag) |
|---|---|---|---|---|---|
| Green | 0.290 | 0.071 | 0.114 | 0.000 | 0.000 |
| Yellow | 0.000 | 0.000 | 0.282 | 0.025 | 0.006 |
| Red | 0.000 | 0.000 | 0.000 | 0.135 | 0.076 |
| Sum: | 0.290 | 0.071 | 0.396 | 0.160 | 0.082 |

Table 5.a. 3: Probabilities of the damage grades conditioned on whether a building has been
inspected (the probabilities for inspected buildings are based on the damage observations,
the probabilities for the uninspected buildings are based on expert judgement)

| Inspected | P(Insp.) | P(DG1\|Insp.) | P(DG2\|Insp.) | P(DG3\|Insp.) | P(DG4\|Insp.) | P(DG5\|Insp.) |
|---|---|---|---|---|---|---|
| TRUE | 0.180 | 0.290 | 0.071 | 0.396 | 0.160 | 0.082 |
| FALSE | 0.820 | 0.500 | 0.300 | 0.100 | 0.050 | 0.050 |

Table 5.a.4: Calculation of the total probabilities of the damage grades accounting for both
inspected and uninspected buildings

| Inspected | P(DG1\|Insp.)·P(Insp.) | P(DG2\|Insp.)·P(Insp.) | P(DG3\|Insp.)·P(Insp.) | P(DG4\|Insp.)·P(Insp.) | P(DG5\|Insp.)·P(Insp.) |
|---|---|---|---|---|---|
| TRUE | 0.052 | 0.013 | 0.071 | 0.029 | 0.015 |
| FALSE | 0.410 | 0.246 | 0.082 | 0.041 | 0.041 |
| Sum: | 0.462 | 0.259 | 0.153 | 0.070 | 0.056 |

Description of the revision
The tables 5.a.1-4 have been added to the manuscript (Tables 2-4 – 2-7), as well as a
paragraph (lines 161-191).
5.b. The numbers of buildings from observations in Fig. 5b are much larger than the 327
buildings included in the damage survey. Why is this the case?
Review round 1 reply
Thank for this question. The numbers of buildings in Fig. 5b are calculated by multiplying
the total number of buildings in the exposure model by the probabilities in Fig. 5a. The
numbers reported as "Observation-based" result from the multiplication with the
probabilities calculated according to our response to the previous comment (comment
5.a). We acknowledge that the figure may mislead the reader to think that the numbers in
Fig. 5b correspond to numbers of observations. Therefore, we will rename the label
"Observation-based" in the legends in Fig. 5a-5b to "Calc. on insp.", shorthand for
"Calculation based on the damage grade probabilities for inspected and uninspected
buildings".
Description of revision
The revised manuscript explains how the numbers of buildings in Fig. 6-8 are calculated
(lines 161-191), and the labels DD2 and DD3 have replaced the label "Observation-
based" in Fig 6-8.
5.c. At the same time, the plots in Fig. 5 have two separate categories, "Exp. judg.-
based" and "Observation-based", but I have found no explanation regarding what this
means, as lines 324-326 only say "Two of the sources consist of probabilities based on
expert judgement ("Exp. judg.- based"), and probabilities based on our conversion of the
damage observations to damage grades ("Observation-based"), but the meaning of
"based on expert judgement" is not explained. It is noted as well that "our conversion of
the damage observations to damage grades" is also "expert judgment", and thus the
difference between the two requires a more detailed clarification.
Review round 1 reply

It is true that both results labelled as "Exp. judg.-based" and "Observation based" have been calculated using expert judgment to different extents. Please see our responses to comments 5.a and 5.d, which also respond to comment 5.c.

Description of revision
The calculation of the probabilities labelled as DD2 (the label used in the revised manuscript instead of "Observation-based"), and DD3 is described in the revised manuscript (lines 161-191).

5.d. The above makes me wonder if one of the two "observation" labels in the plots in Fig. 5 has been created using Table 3-6. I have been unable to find any reference to Table 3-6. Please clarify if Table 3-6 is being used and reference it within the text if this is the case.

Review round 1 reply
Yes, Table 3-6 is used for one of the probabilities in Fig 5.a labelled "Exp. judg.-based". We will clarify this with a more precise nomenclature. We will add a table that explains how the "observations-based" damage distributions have been generated (see table in our answer to Comment 7.c in the section "Reviewer 1 - Main Comments"). That table will include a reference to Table 3-6.

Description of revision
The Table 3-6 is now numbered as Table 2-8 in the revised manuscript and its use is described in lines 161-191.

5.e. If more than one method has been used to obtain damage grades from the survey data (apart from the one described in Section 2.2), all methods need to be specified (and given distinct names/labels) in Section 2.2.

Review round 1 reply
We agree with your suggestion: see our answer above and our proposition to add a table describing these methods (answer to Comment 7.c in the section "Reviewer 1 - Main Comments").

Description of revision
These methods are described in Section 2.2.

5.f. The conclusions state "The proposed testing procedure based on the observed damages could be improved by introducing a probabilistic rule for the conversion of damage observations on the three level colour tag (red, yellow, green) scale to the EMS-98 damage scale" (lines 364-365). To my understanding, this is exactly what Table 3-6 is showing. If this is the case, and it has been used, then please adjust the conclusions.

Review round 1 reply
We acknowledge that the manuscript is not clear. The revised manuscript will say instead that one could introduce a conversion rule, which would return damage grade probabilities instead of a single value for the damage grade as a function of the colour tags for structural and non-structural elements. Thank you for this comment.

Description of revision
The conclusions have been revised (lines 593-595).

5.g. I cannot find any reason for Table 3-6 not to be used. Showing and discussing

"observed" damage results obtained using both strategies (Table 2-1 and Table 3-6), which is potentially what is shown in Fig. 5 but not sufficiently explained, would convey to the reader the inherent uncertainty involved in the comparison between the models and the observations (i.e., "observations" are not a ground truth), which is fundamental in any comparison between models and data (i.e., the uncertainties do not only exist in the models).

Review round 1 reply

Thank you for this comment. Table 3-6 is used to calculate the probabilities in Fig. 5.a labelled as "Exp. judg.-based".

Description of revision

The Table 3-6 is numbered Table 2-8 in the revised manuscript and its use is described in lines 187-191.

6. Associated with the previous point, please explain in the paper how the ESRM20 damage scale (associated with the ESRM20 fragility models) was converted into the EMS-98 scale, as this is another source of uncertainty in the comparison.

Review round 1 reply

Thank you very much for this comment. Indeed, this conversion can be a source of uncertainty.

It will be described in the revised manuscript. The conversion was done by matching the damage states/grades based on the structural damage since both scales assume the level of non-structural damage based on the level of structural damage. A table like the following will be added to the manuscript:

Table: Conversion of the damage scale of the ESRM20 fragility models to the EMS-98 damage scale on the basis of structural damage

| ESRM20 | EMS98 |
|---|---|
| D0 no damage (combined structural and non-structural damage) [This damage state is not explicitly mentioned by the damage scale, but it is implied] | Grade 0 No damage [This damage state is not explicitly mentioned by the damage scale, but it is implied] |
| | Grade 1: Negligible to slight damage (no structural damage, slight non-structural damage |
| D1 slight (combined structural and non-structural damage) | Grade 2: Moderate damage (slight structural damage, moderate non-structural damage |
| D2 moderate (combined structural and non-structural damage) | Grade 3: Moderate damage (moderate structural damage, heavy non-structural damage |
| D3 extensive (combined structural and non-structural damage) | Grade 4: Very heavy damage (heavy structural damage, very heavy non-structural damage) |
| D4 complete (combined structural and non-structural damage) | Grade 5: Destruction (very heavy structural damage) |

Description of revision

The conversion is described in the revised manuscript (lines 473-475, Table A4)

7. Lines 110 and 161: The titles of Sections 3.1 and 3.2 need to be changed, as they do not
reflect the content of these sections. Neither section presents a test. They are both a
procedure to select a rupture model to carry out the damage comparisons. The first sentence
of Section 3.1 needs to be changed as well, as the section does not present a comparison
against macroseismic intensities.
Review round 1 reply
The titles of Sections 3.1 and 3.2, as well as the first sentence of Section 3.1, will be changed
in the revised manuscript based on the comment. Indeed, these sections are a procedure to
select a rupture model to carry out the damage comparisons.
Description of revision
In the revised manuscript, the titles of Sections 3.1 and 3.2 have been changed (new Sections
4.1 and 4.2).
8. Line 111 (and other instances): Although the citation of the Wald et al. (2022) paper
indicates that it is the USGS ShakeMap that is being used, it would be good to be explicit (by
saying "USGS ShakeMap"), as the USGS ShakeMap software is also used by other
organisations with their own configuration (e.g., the European ShakeMap, the Italian
ShakeMap).
Review round 1 reply
We will clarify this sentence, as stated in our answer to Comment 3c. In this study, we have
generated the shake-map ourselves, using our specific configuration of the USGS ShakeMap
software.
Description of revision
Please see the description of the revision based on Comment 3c in the section "Reviewer 1 -
Main Comments".
9. Line 114: Which site model was used for the ground motion comparisons?
Review round 1 reply
Thank you for this question. The revised manuscript will describe the site model that was used.
It is a site model including one point for each exposure centroid, with the same coordinates as
its corresponding exposure centroid. The $V_{S30}$ was inferred based on the EC8 soil class map
by the BRGM for the French territory (Roullé & Monfort, 2016). Specifically, the median of each
class was taken as the $V_{s30}$. The outputs of the Vs30 site model for the exposure centroids are
given in the next table, which could be added to the revised manuscript:

| Centroid | latitude | longitude | region | brgm $V_{S30}$ (m·s⁻¹) | ESRM20 $V_{S30}$ (m·s⁻¹) | $V_{S30}$ Type | geology | slope |
|---|---|---|---|---|---|---|---|---|
| 0 | 44.5546 | 4.6835 | 1 | 800 | 807 | inferred | CRETACEOUS | 0.0823 |
| 1 | 44.5453 | 4.6804 | 1 | 270 | 831 | inferred | CRETACEOUS | 0.0645 |
| 2 | 44.5414 | 4.6846 | 1 | 270 | 730 | inferred | HOLOCENE | 0.0487 |
| 3 | 44.5405 | 4.6498 | 1 | 800 | 726 | inferred | CRETACEOUS | 0.0768 |
| 4 | 44.5347 | 4.6713 | 1 | 800 | 831 | inferred | CRETACEOUS | 0.0467 |
| 5 | 44.5500 | 4.6909 | 1 | 270 | 699 | inferred | HOLOCENE | 0.0160 |
| 6 | 44.5442 | 4.6699 | 1 | 800 | 830 | inferred | CRETACEOUS | 0.0522 |
| 7 | 44.5547 | 4.6692 | 1 | 580 | 840 | inferred | CRETACEOUS | 0.0503 |

| 8 | 44.5315 | 4.6953 | 1 | 270 | 644 | inferred | HOLOCENE | 0.0439 |

Associated reference :
Monfort, C., & Roullé, A. (2016). Estimation statistique de la répartition des classes de sol Eurocode 8 sur le territoire français - Phase 1 : Rapport final. BRGM Report RP-66250-FR.

Description of revision
The Table 3-5 has been revised. It is now numbered Table 3-3, and it includes the parameters for the site models. Moreover, the revised manuscript describes the site models in Section 3.

10. Line 139 states that the ground motions were "aggregated over all exposure centroids", but it is not specified whether the values shown are means or medians (of all points). Please specify.

Review round 1 reply
The scenario analyses generated samples of the ground motion intensity measures at the locations of the exposure centroids. The boxplots concern the entirety of the samples for all centroids. Thank you for this comment. This will be clarified in the revised manuscript.

Description of revision
The revised manuscript clarifies how the distributions, the means, and the medians of the values are represented (lines 342-345).

11. Line 139: It is stated that ground motions are calculated at the exposure centroids. However:

11.a. To my understanding, OpenQuake does not calculate the ground motions at the exposure points themselves but at the points of the site model that are closest neighbours to the exposure points (and assigns the ground motions to the exposure points by closest neighbours, not interpolation). This can be checked by looking at the sitemesh_XXX.csv output by OpenQuake, as this shows the locations at which ground motions were calculated. If this is the case, it would be relevant to know what site model is being used and its resolution with respect to the resolution of the exposure points.

Review round 1 reply
Thank you for this comment. The site model includes points with coordinates identical with those of the exposure points. The manuscript will be revised accordingly. See also the reply to Comment 9 in the section "Reviewer 1 - Other Comments on Content".

Description of revision
The lines 303-304 have been added to the manuscript.

11.b. At this stage, the exposure model has not been described, and different exposure models are used later on in the paper. Please indicate if the "exposure centroids" refer to the building-by-building data of the post-earthquake damage survey or other locations.

Review round 1 reply
The exposure centroids refer to the 9 centroids of the 9 intra-municipal districts in BRGM's exposure model for the town of Le Teil (Table 3-5). This will be clarified in the revised manuscript. Thank you for this comment.

Description of revision
The revised manuscript describes the exposure models in Section 3.2 before the
comparison with respect to intensity measures in Section 4.1.

12. Lines 149-150: It would be relevant to comment on whether the USGS ShakeMap for this
earthquake was constrained with direct ground motion measurements (from stations) and/or
Did You Feel It macroseismic intensity observations. For reproducibility, please include as
well the version of the USGS ShakeMap used, as the USGS recalculates ShakeMaps when
new data or new algorithms become available.
Review round 1 reply
As stated in previous answers to comments ,we have generated the shake-map ourselves,
using our specific configuration of the USGS ShakeMap software (version 4). The parameters
related to this shake-map are detailed in the table that we propose to add (see GM4 in the
table added in the answer to Comment 3c in the section "Reviewer 1 - Main Comments").
The shake-map for this earthquake was constrained with ground motion measurements only
(no DYFI). However, the closest stations are over 15 km from the epicentre, which leads to
practically no constraint. We will discuss this issue in the revised manuscript.
Description of revision
Please see the description of the revision based on Comment 3c in the section "Reviewer 1 -
Main Comments". Moreover, the lines 152-155 have been added to the manuscript.
13. Line 151, Fig. 1: It would help the reader if the vertical axis contained the non-logarithmic
values of the IM (potentially side by side with the logarithmic ones, or as a scale on the right
side of the plot).
Review round 1 reply
Fig. 1 will be revised according to this comment.
Description of revision
A scale on the right side of the plot has been added. The figure now is numbered as Figure
2.
14. Line 181, Table 3-2: Is it relevant to show the parameters for the CA2015 model and not
the FM2010 model?
Review round 1 reply
The parameters for the FM2010 model will be added to the revised manuscript as well.
Description of revision
The parameters for the FM2010 model are included in the revised manuscript (Table 3-3).
15. Lines 193 and 197 use the acronym "KO2020", which has not been defined.
Review round 1 reply
Any reference to KO2020 will be removed, and the rest of the manuscript will be revised
accordingly. We apologize that this was left after a revision of a working version of the
manuscript. Thank you for this comment.

Description of revision
The revised manuscript does not use the acronym KO2020.

16. Lines 210-226: There are some aspects of the comparison shown in Section 3.3.1 that are not explained and are relevant for interpreting the results. Please specify in the paper:

16.a. Lines 212-213 state that the "ESHM20 ground motion logic tree" was used, but so far there has been no reference to the ESHM20 ground motion logic tree, only to the KothaEtAl2020Site implementation of the Kotha et al. (2020) GMPE, which, as explained earlier, is not the one used in ESHM20. Please clarify which logic tree is being used.

Review round 1 reply
Thank you for requesting this clarification. Indeed, in Section 3.2 no ground motion logic tree is used. For the calculation in Section 3.3.1, the ESHM20 ground motion logic tree is being used, which employs the GMPE «KothaEtAl2020ESHM20». The revised manuscript will include this clarification.

From a technical point of view, the file gmpe_complete_logic_tree_5br.xml was edited by removing all other «logicTreeBranchSet» other than «branchSetID="Shallow_Def"», which corresponds to the regime of the study area, because errors related to the removed branches were preventing the completion of the analysis. In our opinion, this technical detail will not be of interest to the readers, but it will be included in the revised manuscript if you consider it should be.

Description of revision
The revised manuscript specifies the GMPE and the logic tree that is used in the comparison in section 3.3.1.

16.b. Lines 214-215: If "equivalent" exposure and fragility models are being used "so as to limit the effect of these two factors on the differences between the two estimations", what is the purpose of this comparison? Comparing a model in Armagedom against a model in OpenQuake? Is the equivalence between the models fully guaranteed? Please clarify the purpose of the comparison presented in Section 3.3.1.

Review round 1 reply
OpenQuake and Armagedom use different methods for the damage estimation.

As mentioned previously, Armagedom uses the RISK-UE semi-empirical macroseismic method. This is based on the intensity values and a vulnerability index for the calculation of the mean damage degree for the beta distribution.

OpenQuake uses ground motion intensities and fragility curves.

The two methods are obviously different, but, no matter what their path, the results of both methods have the same aim: asses the damages after an earthquake. Considering this same objective, the results from the two methods can be compared.

Nevertheless, we agree with your comment, and we will add a paragraph to summarise both methods.

A few articles attempt to address the issue (e.g. Lestuzzi et al. 2016).

Lestuzzi, P., Podestà, S., Luchini, C. et al. Seismic vulnerability assessment at urban scale
for two typical Swiss cities using Risk-UE methodology. Nat Hazards 84, 249–269 (2016).
https://doi.org/10.1007/s11069-016-2420-z

Description of revision
A paragraph has been added to the manuscript (lines 249-260), which underlines the main
differences between the two analysis tools and clarifies the purpose of this comparison.

16.c. Lines 215-216: Please clarify in the paper the meaning of "the exposure model in
Armagedom". I am not familiar with the software, but the paper of Sedan et al. (2013)
gives the impression that Armagedom is a software and the user can input any exposure
model as desired. Please clarify in the paper how this exposure model was defined.

Review round 1 reply
Yes, we will explain the exposure model used in Armagedom, based on vulnerability
indices of building classes. A more detailed answer and paragraph is available below (see
answer to Comment 25). Yes, Armagedom is able to treat any exposure model, as long as
the preliminary step of converting building class to vulnerability indices is carried out.

Description of revision
The revised manuscript gives details on the exposure model used in the calculation with
Armagedom (lines 251-253, 259-260).

16.d. Lines 215-221: Does the exposure model used in OpenQuake maintain the 9
centroids mentioned in line 217?

Review round 1 reply
Yes, it does.

Description of revision
This is specified in the revised manuscript (lines 246-247).

16.e. Please comment in the paper (a paragraph would suffice) about the details of the
damage calculation in Armagedom: use of conversion models to transform PGA into
macroseismic intensity, calculation of a mean damage grade as a function of
macroseismic intensity, distribution into damage grades under the assumption of a Beta
distribution, etc. This method is fundamentally different from the calculation carried out in
OpenQuake in terms of PGA/SA, with damage grades directly retrieved from the fragility
model, conversion of ESRM20 damage grades into ESM-98 damage grades, etc. Without
these details and comparisons, it may not be fully evident to the reader what the purpose
of this section is.

Review round 1 reply
Thank you for this comment. It is indeed worth describing the procedure used by
Armagedom and highlighting the differences from the calculation in OpenQuake. A
paragraph on this subject will be added to the revised manuscript.

Description of revision
A paragraph has been added to the revised manuscript (lines 249-255).

16.f. Lines 224-225: These sentences compare the values obtained against observations,
but the percentages of "heavy" and "very heavy" damage observed are not reported.
Please add them in the text. It is also not clear why the observed values are not shown in
Fig. 3, given that they are shown later in Figs. 4 and 5 (converting number of buildings
into proportions, as in the other plots, or using a right-hand axis with a different scale on
the same plot).
Review round 1 reply
Thank you for this comment. Indeed the values calculated based on the observations
should have been included in Fig. 3, and they will be included in the revised manuscript.
We should note that since the percentages concern the entire town of Le Teil, the
percentages calculated based on the observations are calculated according to our
response to comment 4 in the section "Reviewer 1 - Other Comments on Content". The
revised manuscript will also report the percentages for "heavy" and "very heavy" damage.
Description of revision
The percentages of "heavy" and "very heavy" damage are reported in the manuscript (line
268).
The observed values have been added to the figure (the new number of the figure is Fig.
4, line 277)
16.g. Do the OpenQuake damage results correspond to the average damage resulting
from all 1,000 ground motion realisations (only mentioned in Section 2.1) and all logic
tree branches (if a ground motion logic tree was indeed used)? Please specify.
Review round 1 reply
Yes, they do correspond to the average damage from all ground motion realisations for all
logic tree branches. The manuscript will be revised accordingly. Thank you for this
comment.
Description of revision
This is specified in the revised manuscript (line 266-267).
16.h. Does Armagedom calculate different ground motion fields (1,000 as well?) to
account for ground motion uncertainty?
No, currently, Armagedom does not generate stochastic samples of ground-motion fields.
It applies the GMM and estimates only the mean ground-motion parameters across the
map.
17. Line 240: To my knowledge, the most recent reference of GED4ALL is Silva et al. (2022),
and the preferred name for this building taxonomy is "GEM Building Taxonomy v3.0":
Silva V, Brzev S, Scawthorn C, Yepes C, Dabbeek J, Crowley H (2022) A building
classification system for multi-hazard risk assessment. International Journal of Disaster Risk
Science 13:161–177. https://www.doi.org/10.1007/s13753-022-00400-x
Review round 1 reply
Thank you for indicating the correct reference. It will be corrected in the revised manuscript.
Description of revision

In the revised manuscript, GED4ALL has been replaced by "GEM Building Taxonomy v3.0",
and the reference of Silva et al. (2022) has been added (line 284).
18. Line 240: I would suggest to re-phrase "we selected a GED4ALL building class based
on…" as "we defined building classes in terms of the GEM Building Taxonomy v3.0 (Silva et
al., 2022), based on the building materials and the number of storeys". The current phrasing
may erroneously convey that the taxonomy consists of a pre-defined list of building classes
to choose from, instead of a classification system of attributes to be concatenated.
Review round 1 reply
Thank you very much for this suggestion. We see how the phrasing may be misleading. As
suggested, we will rephrase this in the revised manuscript.
Description of revision
The manuscript has been rephrased as indicated (line 284).
19. Line 245, Table 3-4: It is interesting that fragility models for infilled frames ("CR_LFINF")
were selected for dual frame-wall systems ("CR/LDUAL"), instead of using the "CR_LDUAL"
fragility models directly (one of which is mentioned in Table 3-3). Please comment in the
paper on this choice. Moreover, the reinforced concrete ESRM20 classes selected
correspond to different values of the lateral force coefficient, and it is not clear how this could
be selected from the damage dataset. Please comment.
Review round 1 reply
Thank you for this comment. We made the arbitrary choice to classify the reinforced concrete
buildings in the dataset as CR/LDUAL. We should have simply assigned to them a CR class.
We agree that the lateral force coefficient may not be selected based on the damage dataset.
Moreover, we did not consider it during the selection of the fragility models. We assigned an
EMS98 vulnerability class based on the year of construction. Subsequently, we selected
fragility models, which we considered to be in agreement with the construction material and
the EMS98 vulnerability classes.
Description of revision
In Table 3-4 (line 295), the GEM Building Taxonomy v3.0 class the building classes
CR/LDUAL/HAPP:2 and CR/LDUAL/HAPP:4 have been replaced by CR/HAPP:2 and
CR/HAPP:4, respectively.
The revised manuscript comments on the lateral force coefficient related to the ESRM20
building classes (287-291).
20. Lines 249-254: Please specify the GMPE used.
Review round 1 reply
The GMPE KothaEtAl2020Site has been used; but with the proposed revisions, we will now
apply two GMMs (KothaEtAl2020Site and KothaEtAl2020ESHM20SlopeGeology). This will
be better explained thanks to the following table:

| GM Map ID | Type | GMM | Site model | Rupture model | Observations |
|---|---|---|---|---|---|
| GM1 | ground-motion field | KothaEtAl2020 Site | BRGM soil classes to Vs30 | Ritz et al. | No |

| | | | | | |
|---|---|---|---|---|---|
| GM2 | ground-motion field | KothaEtAl2020 ESHM20Slope Geology | Slope & Geology (ESRM20 data) | Ritz et al. | No |
| GM3 | ground-motion field | KothaEtAl2020 Site | ESRM20 Vs30 data | Ritz et al. | No |
| GM4 | shake-map | KothaEtAl2020 Site | BRGM Soil class to Vs30 | Ritz et al. | Seismic stations |

Description of revision
The GMPE is specified in the manuscript (lines 308-310), and the table above has been added (Table 3-6).

21. Lines 254-256: The label "SM – brgm VS30" suggests that the BRGM model was used together with the USGS ShakeMap. How was this site model incorporated to the ShakeMap? Does this mean the ShakeMap used in the paper is not the one downloaded from the USGS but the authors have run the ShakeMap software themselves? Please clarify in the manuscript.

Review round 1 reply
As stated in previous answers to comments, we have generated the shake-map ourselves, using our specific configuration of the USGS ShakeMap software (version 4). The parameters related to this shake-map are detailed in the table that we propose to add (see GM4 in the table added in the answer to Comment 3c in the section "Reviewer 1 - Main Comments"). We will revise the nomenclature of these labels ("SM – brgm Vs30") according to that new table.

22. Line 283 (Fig. 4) and Line 341 (Fig. 5): Please clarify if the proportions of buildings in each damage grade stemming from the calculations have been calculated with respect to the total number of buildings (including undamaged ones) or only the number of damaged buildings (which I understand is the case for the observation values).

Review round 1 reply
Thank you for requesting this clarification. In the revised manuscript, it will be clarified by a new table (as introduced in our answer above), which will explain the number of buildings considered in each comparison (differences between Fig. 4 and Fig. 5):

| Observed Damage Data ID | Exposure resolution | Exposure data | Damage estimation method | Damage conversion method | |
|---|---|---|---|---|---|
| DD1 | Building-by-building (327 buildings) | AFPS emergency survey | AFPS emergency observations on 327 buildings (Green/Yellow/Red tags) | Conversion to EMS-98 damage grades (Tab. 2.1) | Related to Fig. 4 |
| DD2 | Infra-municipality districts (2778 buildings) | National statistics database (BRGM-CCR) | AFPS emergency observations on 327 buildings (Green/Yellow/Red tags) + "Extrapolation" | Conversion to EMS-98 damage grades with expert judgment (Tab. 3.6) | Related to Fig. 5 |

| | | | | | |
|---|---|---|---|---|---|
| DD3 | Infra-municipality districts (2778 buildings) | National statistics database (BRGM-CCR) | AFPS emergency observations on 327 buildings (Green/Yellow/Red tags) + "Extrapolation" | Conversion to EMS-98 damage grades (Tab. 2.1) + Bias adjustment on total number of 2778 buildings (accounting for non-surveyed buildings) | Related to Fig. 5 |

Description of revision
The proportions of buildings in the calculations has been specified (Table 3-10, and lines 423-439). Please also see the revision with respect to comment 5a in the section "Reviewer 1 - Other Comments on Content".

23. Line 284 (caption of Fig. 4), and Table A3: Please clarify what the acronym "BRGM/CCR" refers to. I find it confusing that it is named in Fig. 4, which corresponds to analyses carried out using the buildingby-building exposure based on the 327 surveyed buildings, and then in Table A3, which lists 2,778 buildings, which is the number reported in both Sections 3.3.1 (line 216, "the exposure model in Armagedom, which includes 2778 buildings") and 3.3.4 (lines 293-294, "the second exposure model ("brgm exp.") is based on national statistical data, and includes 9 centroids with 2778 buildings"). Please clarify the relation between the exposure models used in Sections 3.3.1 and 3.3.4: are they the same? Please add reference to Table A3 within the text.

Review round 1 reply
The nomenclature of the exposure models will be clarified: the "BRGM/CCR" label refers to the same exposure model as "brgm-exp". This will also be clarified by the above table of observed damage data.

We will add a reference to Table A3 in the text.

Description of revision
The caption of Fig. 4 (Fig. 5 in the revised manuscript) has been corrected, as well as the caption of Table A3.
A reference to Table A3 has been added (line 369).

24. Lines 291-293, and Tables A1 and A2: It is not clear why the ESRM20 exposure model is not being used directly as it is, including its exposure-to-vulnerability mapping. The changes introduced by the authors mean that the calculations carried out with this model may not necessarily reflect what would have been obtained with the "original" ESRM20 model.

Moreover, the choice of fragility classes for each exposure class shown in Table A1 appears as contradictory. In the screenshot of Table A1 below, I have marked the differences in the classes and annotated the classes used in ESRM20, which can be consulted in the esrm20_exposure_vulnerability_mapping.csv file of the ESRM20 v1.0 repository[5]. The differences are associated with the number of storeys (e.g., a 4-storey class has been selected for a 6-and-above-storey class, first row) and the lateral force coefficient and/or design code level (e.g., a low code class with 15% lateral force coefficient has been selected for a no-code class, seventh row). Please justify the need to use a "simplified" version of the
* * *
[5] https://gitlab.seismo.ethz.ch/efehr/esrm20/-/blob/v1.0/Vulnerability/esrm20_exposure_vulnerability_mapping.csv exposure model (instead of the original ESRM20 exposure) and explain the criteria used to
assign new classes in Table A1 (in the main body of the paper).

**Table A1 Selected ESRM20 fragility classes based on the building types in Le Teil according to the ESRM20**

| Original ESRM20 type | N. buildings | Selected ESRM20 frag. class | # class |
|---|---|---|---|
| CR/LDUAL+CDL+LFC:4.0/HBET:6- | 3 | CR_LDUAL-DUL H4 **H6** | 1 |
| CR/LDUAL+CDL+LFC:4.0/HBET:3-5 | 7 | CR_LDUAL-DUL_H4 | 1 |
| CR/LDUAL+CDN/HBET:6- | 2 | CR_LDUAL-DUL H4 **H6** | 1 |
| CR/LFINF+CDL+LFC:4.0/H:2 | 67 | CR_LFINF-CDL-10 H2 **CDL-5** | 2 |
| CR/LFINF+CDM+LFC:4.0/H:1 | 42 | CR_LFINF-CDM-10 H2 **CDM-5_H1** | 3 |
| CR/LDUAL+CDM+LFC:4.0/HBET:6- | 1 | CR_LDUAL-DUL_H4 **H6** | 1 |
| CR/LFLS+CDN/HBET:6- | 9 | CR_LFINF-CDL-15 H4 **CDN-0_H6** | 4 |
| CR/LFINF+CDL+LFC:4.0/H:1 | 76 | CR_LFINF-CDM-10 H2 **CDL-5_H1** | 2 |
| CR/LDUAL+CDM+LFC:4.0/HBET:3-5 | 3 | CR_LDUAL-DUL_H4 | 1 |
| MUR+CL/LWAL+CDN/H:2 | 378 | MUR-CL99_LWAL-DNO_H2 | 5 |
| CR/LFINF+CDM+LFC:4.0/H:2 | 37 | CR_LFINF-CDM-10 H2 **CDM-5** | 3 |
| MUR+CL/LWAL+CDN/H:1 | 690 | MUR-CL99_LWAL-DNO_H1 | 6 |
| MUR+ST/LWAL+CDN/H:2 | 130 | MUR-CL99_LWAL-DNO_H2 **STDRE** | 5 |
| CR+PC/LWAL+CDN/HBET:3-5 | 53 | CR_LDUAL-DUL_H4 | 1 |
| W/LWAL+CDN/H:1 | 100 | W_LFM-DUL H2 **H1** | 7 |
| W/LWAL+CDN/H:2 | 43 | W_LFM-DUL_H2 | 7 |
| CR+PC/LWAL+CDN/HBET:6- | 1 | CR_LDUAL-DUL H4 **H6** | 1 |
| CR/LFINF+CDN/HBET:3-5 | 38 | CR_LFINF-CDL-15 H4 **CDN-0** | 4 |

Review round 1 reply
The ERSM20 model includes a number of building classes, which is higher than the number
of classes in the BRGM exposure model. Moreover, the ESRM20 model includes classes with
a small percentage of the total number of buildings, which could be grouped with similar
classes. For example, we decided to group in Class 1 (revised Table A1) buildings categories
with 6 or more storeys, which have a small number of buildings, together with buildings with 3-
5 storeys on the basis of the similarity of their load-bearing systems.
The merger of similar classes and the reduction of the total number of classes had the goal of
simplifying the comparisons. Moreover, we hoped that, if there were comparable classes, we
would be able to attribute differences in the results to specific classes based on the numbers
and probabilities of damage per building class.
Revised Table A1

| Original ESRM20 type | N. buildings | Selected ESRM20 frag. class | Class |
|---|---|---|---|
| CR+PC/LWAL+CDN/HBET:3-5 | 53 | CR_LDUAL-DUL_H4 | 1 |
| CR/LDUAL+CDL+LFC:4.0/HBET:3-5 | 7 | CR_LDUAL-DUL_H4 | 1 |
| CR/LDUAL+CDM+LFC:4.0/HBET:3-5 | 3 | CR_LDUAL-DUL_H4 | 1 |
| CR/LDUAL+CDL+LFC:4.0/HBET:6- | 3 | CR_LDUAL-DUL_H4 | 1 |
| CR/LDUAL+CDN/HBET:6- | 2 | CR_LDUAL-DUL_H4 | 1 |
| CR+PC/LWAL+CDN/HBET:6- | 1 | CR_LDUAL-DUL_H4 | 1 |
| CR/LDUAL+CDM+LFC:4.0/HBET:6- | 1 | CR_LDUAL-DUL_H4 | 1 |
| CR/LFINF+CDL+LFC:4.0/H:1 | 76 | CR_LFINF-CDL-10_H2 | 2 |
| CR/LFINF+CDL+LFC:4.0/H:2 | 67 | CR_LFINF-CDL-10_H2 | 2 |
| CR/LFINF+CDM+LFC:4.0/H:1 | 42 | CR_LFINF-CDM-10_H2 | 3 |
| CR/LFINF+CDM+LFC:4.0/H:2 | 37 | CR_LFINF-CDM-10_H2 | 3 |
| CR/LFINF+CDN/HBET:3-5 | 38 | CR_LFINF-CDL-15_H4 | 4 |
| CR/LFLS+CDN/HBET:6- | 9 | CR_LFINF-CDL-15_H4 | 4 |
| MUR+CL/LWAL+CDN/H:2 | 378 | MUR-CL99_LWAL-DNO_H2 | 5 |
| MUR+ST/LWAL+CDN/H:2 | 130 | MUR-CL99_LWAL-DNO_H2 | 5 |
| MUR+CL/LWAL+CDN/H:1 | 690 | MUR-CL99_LWAL-DNO_H1 | 6 |
| W/LWAL+CDN/H:1 | 100 | W_LFM-DUL_H2 | 7 |
| W/LWAL+CDN/H:2 | 43 | W_LFM-DUL_H2 | 7 |

However, in response to your suggestion, we propose to do an extra analysis using the original
ESRM20 exposure model, in order to check potential differences. This will be discussed in the
revised manuscript.
Revision description
Table A1 has been revised, and the manuscript justifies the simplification of the exposure
model (lines 367-373).
An additional analysis has been done using the original ESRM20 exposure model, whose
results are included in Fig. 6 (labelled "ESRM20 Vs30 – ESHM20 GMF – Orig. ESRM20
exp."). Despite the incoherencies with respect to the lateral force coefficient and/or design
code level, the calculated damage presents insignificant differences from the calculation
using the simplified exposure ("ESRM20 Vs30 – ESHM20 GMF – ESRM20 exp.").
25. Lines 291, 294: Please clarify in the manuscript that only residential buildings from the
ESRM20 exposure model are being included in the calculation (I have deduced this from
looking at the ESRM20 exposure model for France). Please clarify as well if the BRGM
exposure considers only residential buildings as well, and whether it covers the same spatial
extent (even better if using a map). Please clarify if the damage observations only cover
residential buildings as well.
Review round 1 reply
Yes, for the aggregated exposure models (Section 3.3.4) the BRGM exposure considers only
residential buildings as well and it covers the same spatial extent (Teil administrative borders).
The residential exposure data were extracted from the building census database at the
municipality (and infra-municipality) level, provided freely by the national statistical database
INSEE. Based on structural criteria available, as well as a pilot project in Bouches-du-Rhône
Department (Sedan et al., 2008), which compared field investigation data and INSEE data at
the departmental scale level, we derived a matrix—consisting of a cross between the age of
construction, number of stories, and type of construction—for a simplified description of the
vulnerability based on the INSEE data. Therefore, starting from INSEE statistics, we classified
the buildings into EMS98 taxonomy classes. The EMS98 scale associates vulnerability classes
(A, B, C, D, E, and F) to the most common structural types (masonry, reinforced concrete,
steel, and wood), indicating the most likely, probable, and less probable ranges that a structural
type belongs to a given vulnerability class. Then, the EMS98 taxonomy classes were converted
into RISK-UE vulnerability indices, based on the method developed by (Lagomarsino and
Giovinazzi, 2006; Milutinovic and Trendafiloski, 2003). A national classification was done in
the past by brgm. More details about this procedure can be find in Fayjaloun et al. (2021).
For "building-by-building" exposure model (Sect 3.3.3) we used the AFPS database that
concerns, as well, only the residential buildings.
Associated reference:
Fayjaloun, R., Negulescu, C., Roullé, A., Auclair, S., Gehl, P., & Faravelli, M. (2021). Sensitivity
of earthquake damage estimation to the input data (soil characterization maps and building
exposure): Case study in the Luchon Valley, France. Geosciences, 11(6), 249.
Description of revision
The manuscript specifies that the exposure models and the damage observations concern
residential buildings (lines 423-426, 435).
26. Lines 300-304: By using a weighting scheme for the so-called "ESHM VS30" model but
not for the BRGM model, this comparison becomes not just about the VS30 models but the different ways of assigning values to an aggregated area. It would be useful to highlight this
further in the text.
Review round 1 reply
Thank you for pointing this out. We will add a sentence on this issue in the text:
"It should be noted that these two different ways to collect Vs30 values at the centroids
(weighted mean of Vs30 values across the area versus punctual value at the centroid) may
constitute an additional source of discrepancy, in addition to the initial differences between the
two Vs30 models."
Description of revision
The sentence above has been added to the manuscript (lines 452-454).
27. Line 310, Table 3-5: The table shows 8 locations but the text (line 294) says "9
centroids". Please correct where needed.
Review round 1 reply
We apologize for this mistake, as a line of the table was erased. The table will be corrected so
that it shows 9 locations. This will also be corrected throughout the manuscript in the revised
version.
The new table will also contains new fields, providing values for the slope and geology related
to the 9 locations (since these parameters will be used by the
KothaEtAl2020ESHM20SlopeGeology GMM). The new version of the table is shown in the
answer to Comment 9.
Description of revision
The table has been corrected (Table 3-8, line 466).

**Reviewer 1 - Language Use, Typos**

Please make the following changes.

1. What do the authors mean with "ShakeMap analyses"? It seems to me that, in most cases, the authors simply mean "ShakeMaps". Please revise and re-phrase all instances along the paper. Examples:

    2.a. Line 14: Just "ShakeMaps in order to…".

    2.b. Line 49: Just "to distributions given by ShakeMaps".

2. Line 10: "validated individually, although testing and validating".

3. Line 12: "damage from past earthquakes".

4. Line 15: "components of the 2020 European Seismic Hazard Model" (not "Euro-Mediterranean").

5. Line 16: "the degree of damage" or "the damage grade".

6. Line 22: "insured and uninsured direct economic losses". I assume this was the intention, as only indirect economic losses are mentioned otherwise.

7. Line 23: "(PSHA, PSRA are…" (not "PSHR").

8. Line 53: Please define VS30 in its first appearance (this line).

9. Line 77: "vulnerability classes" (small letters).

10. Line 93: "data in the forms that we used are" (no commas).

11. Line 101, Table 2-1: "Vertical load-bearing" and "Horizontal load-bearing" (not "loads").

12. Line 115: "the ruptures in the ShakeMap as well as".

13. Line 121: "scaling relation".

14. Line 123: "we assume that its geometric centroid is located at the hypocentre".

15. Line 131, Table 3-1: In the caption, "Rupture parameters associated with the five source models".

16. Line 156, Fig. 1, caption: "ground motion intensity measures aggregated from all exposure centroids".

17. Line 164: "to identify the ruptures leading to".

18. Line 168: The equation starts "MCS =" but the subscript of the standard deviation says "MMI". Is this correct? (See line 170 as well).

19. Line 177: "The CA2015 model".

20. Line 201: "(FM2010), and b) the macroseismic intensity".

21. Line 203: "PGA given by and the ground motion-to-intensity".

22. Line 210, caption: "at the exposure centroids of the BRGM exposure in the site
models…" (or appropriate name for the exposure model).

23. Line 350: "closer to the estimation of EMS-98 macroseismic intensity by Schlupp et al.
(2022)". The text before that statement had not yet mentioned macroseismic intensity.

**Reviewer 1 - Issues with References**

1. Line 384: There are numbers at the end of "Munson" and "Stamatakos".

2. Lines 396-397: The citation of Crowley et al. (2021) is incomplete (no initials of first
names, no DOI, mention of EFEHR Technical Report 002 missing). Please cite as (apply
journal formatting style):

Crowley, H., Dabbeek, J., Despotaki, V., Rodrigues, D., Martins, L., Silva, V., Romão, X.,
Pereira, N., Weatherill, G. and Danciu, L., 2021. European Seismic Risk Model (ESRM20),
EFEHR Technical Report 002, V1.0.1, 84 pp, https://doi.org/10.7414/EUC-EFEHR-TR002-
ESRM20

3. Lines 408-411: The citation of Danciu et al. (2021) is not fully correct. Please cite as :

Danciu, L., Nandan, S., Reyes, C., Basili, R., Weatherill, G., Beauval, C., Rovida, A.,
Vilanova, S., Sesetyan, K., Bard, P.-Y., Cotton, F., Wiemer, S., and Giardini, D.: The 2020
update of the European Seismic Hazard Model: Model Overview, EFEHR Technical Report
001, V1.0.0, https://doi.org/10.12686/A15, 2021.

Review round 1 reply
The issues with the References, as well as the typos and the instances of incorrect language
use will be corrected in the revised manuscript. Thank you for pointing them out.

Description of revision
The instances of incorrect language use, the issues with references, and the typos have been
corrected.

**Reviewer 2**

Review of Manuscript egusphere-2023-1740
Testing the 2020 European Seismic Hazard and Risk Models using data from the 2019 Le Teil (France) earthquake

The manuscript is a research study devoted to carry out a testing and validation study of components involved in the seismic hazard and seismic risk estimation. The testing of ground motion and damage to building is done using several models, observations of ground shaking and observed damage from past earthquakes. The authors investigate if the obtained scenarios are consistent with observations and the reason for the obtained differences.

The topic of the paper is very interesting and suitable for the readers of the journal. However, the title and the redaction of the manuscript do not help to get this goal. The focus on European Seismic Hazard and Risk Models distracts from the very interesting part of the manuscript.

The manuscript should be focused as a sensitivity study of the ground motion estimation and damage estimation using different input models and how these are closest or not to the observed data from Le Teil earthquake.

Therefore, each section must be introduced with the models that are going to be compared, why are those comparisons going to be done in that section?.

Additionally, each comparison must be explained more in detail so the reader can see clearly which models are kept constant and which are compared.

Finally, the author must try to rewrite the conclusions according to the comparisons they are doing. My final recommendation is to reconsider the publication of the manuscript after major revisions.

We thank the reviewer for their positive and constructive comments. We agree that the topic of the paper should lean more towards the comparison of various components of the damage estimation (rupture model, ground-motion model, exposure model, fragility model) instead of sticking strictly to the ESHM20 and ESRM20 framework.

We will clarify the nature and objective of the various comparisons by adding more details in the Introduction (addition of a Figure explaining the structure of the paper) and new tables detailing the various models and their assumptions (see our answers to Comment 1).
We will also enrich the Conclusions section with an account of our findings.
The answers to the reviewer's comments are detailed below.

**REVIEWER 2 - MAIN COMMENTS**

The concept ShakeMap analysis is not clear. The authors cite Wald et al. 2022, but they should explain better.

Review round 1 reply
We will add a few lines to explain the concept of ShakeMap (objective, algorithm, observations used, etc.). It should be noted that we have generated the shake-map ourselves, using our specific configuration of the USGS ShakeMap software (version 4). The parameters related to this shake-map are detailed in the last row of the following table that we propose to add (model GM4):

| GM Map ID | Type | GMM | Site model | Rupture model | Observations |
|---|---|---|---|---|---|
| GM1 | ground-motion field | KothaEtAl2020 Site | BRGM soil classes to Vs30 | Ritz et al. | No |
| GM2 | ground-motion field | KothaEtAl2020 ESHM20Slope Geology | Slope & Geology (ESRM20 data) | Ritz et al. | No |
| GM3 | ground-motion field | KothaEtAl2020 Site | ESRM20 Vs30 data | Ritz et al. | No |
| GM4 | shake-map | KothaEtAl2020 Site | BRGM Soil class to Vs30 | Ritz et al. | Seismic stations |

Description of revision
Lines 56-57 have been added explaining briefly what shake-map analyses are used for and why they are used in this paper.

**Line 62.** When describing the earthquake, you have to indicate also the registered magnitude and focal depth. Also, they indicate a estimated near-faults PGAs with a 68% confidence interval of 0.3-1.9g . Is this a range in the rupture area? Which is the size of the rupture? How can you explain such a high attenuation because the at 15 km the recorded PGA was only 0.04 g (that is a reduction of 77% of the PGA in 15 km if compared with 0.3g).

Review round 1 reply
We will modify the sentence in order to specify the magnitude and focal depth (however, keep in mind that several models have proposed different depths and magnitudes):
"The Le Teil earthquake took place on the 11th of November 2019, and its epicentre is located at 44.518° N 4.671° E (Ritz et al., 2020), with a focal depth of 1 km and a magnitude Mw 4.9 (Ritz et al., 2020), in close proximity to the municipality of Le Teil and the town of Montélimar in the Lower Rhône valley in France."

Causse et al. (2021) estimated a PGA with a 68 % confidence interval of 0.3-1.9 g in the fault projection on ground surface.

In the scenario calculations we use ruptures, whose size is equal to the median rupture area given by the Wells and Coppersmith (1994) scaling law. In the case of the rupture model according to the parameters based on Ritz et al. (2020), the area of the rupture model is equal to 6.49 km$^2$. The revised manuscript will include these details.

The observed high attenuation of PGA is probably due to the very shallow rupture: the Le Teil earthquake is a specific event, which generated very high large intensities right next to the epicentre, however the ground motion attenuated very quickly.

Description of revision
The manuscript has been revised (lines 86-88, 100-103, 94, 212-214).

**Line 75.** Do not use number for macroseismic intensity, it is better to say VII-VIII instead 7-8

In line 81, we mention a decimal intensity of 7.5 (this was mentioned as is in the publication by Schlupp et al., 2022). In order to remain faithful to that publication and to be consistent, we propose to keep numbers to express macroseismic intensity. For the sake of consistency, we will also use "intensity 7" instead of "intensity VII" in line 79.

The use of numbers instead of letters for macroseismic intensity has been advocated by Musson et al. (2010).

Associated reference:
Musson, R. M., Grünthal, G., & Stucchi, M. (2010). The comparison of macroseismic intensity scales. Journal of Seismology, 14, 413-428.

**Line 110.** Regarding the test based on the intensity of the seismic ground motion. The authors compare the different scenarios pointing that the lowest PGA and Sa0.3s must be due to differences in the rupture distance but they do not say anything about which scenarios is closest to the observed ground motion. Which models fit beter the observations?

Review round 1 reply
It is very difficult to compare the models with measured observations (i.e., recordings of seismic stations), since such measures are very sparse (the nearest station is around 15km from the epicentre). Therefore, in the absence of measures in the epicentral area, it is difficult to compare the effects of different rupture distances in this area to measured ground-motions (this is where the relative differences in rupture distance are the largest, as they are greatly reduced further away from the epicentre). This is why we use macroseismic intensity (precise estimates obtained from field surveys) for the comparison. We will add a couple of sentences of explanations on this issue in the text.

Description of revision
Lines 360-365 have been added to the manuscript.

**Line 160.** Regarding the test based on the macroseismic intensity. I do not understand what the authors are trying to demonstrate. If you are using correlations from Ground Motion to Intensity the results that you are going to obtain should be similar to the obtained in the previous section. If the idea is to see which is the best GMICEs for the region, then using only those scenarios is not enough, the authors should look for the most recent correlation (using a higher number of observations ground motions and macroseismic intensity) and simply use that relationship with the corresponding standard deviation and probably the observed intensity at Le Tail will be in that range.

Review round 1 reply
Thank you for this comment. The comparisons based on the macroseismic intensity serve the purpose of selecting one rupture to use in subsequent comparison. This will be clarified in the revised manuscript.

Description of revision
This is clarified in lines 359-360.

**Line 209.** Estimation of damage using different risk analsys tools
    Here the authors compare the damage results using Armagedom and OpenQuake but the section should be explained better. As far as I understand the damage obtained with Armagedom is obtained using the ground motion modelled by the deterministic scenarios (all of used in the previous sections?, one of them?) and the semi-emprical macroseismic method, but regarding Openquake the authors indicate the use the ESHM20 ground motion logic tree (is this meaning you are comparing damage using a deterministic scenarios with damage from a probabilistic hazard map? It sound strange to me. Can you clarify?

Review round 1 reply
For the estimation of damages, Armagedom uses a ground motion or a macroseismic intensity map. This map can be modelled either for a deterministic scenario (magnitude, epicentre, ground-motion models), by numerical simulation or by a probabilistic procedure (probabilistic hazard map). The ground motion map can be derived by Armagedom or can be uploaded from the output of other softwares (ShakeMap, OpenQuake hazard module, etc.). The acceleration ground-motion map must then be converted to macroseismic intensity with a GMICE. In addition, an observed macroseimsic intensity map can also directly be used for damage estimation with Armagedom.

As you well understood, the intensity map is used with the RISK-UE semi-empirical macroseismic method for damage calculation (hence the need for intensity map).

The calculation with OpenQuake is not a classical PSHA. It is a scenario calculation, where the rupture is deterministically defined. Although a ground motion logic tree can be used in combination with a deterministically defined rupture, we do not use any ground motion logic trees, we only use a single GMPE.

Description of revision
The calculation with Armagedom is now compared to the damage scenario DS1.

Which is the method used in OPENQUAKE for the damage estimation is also the same used in Armagedom? Is it a different method? You have explained how this is done to be sure that you can compare the results.

Review round 1 reply
OpenQuake and Armagedom use different methods for the damage estimation.

As mentioned previously, Armagedom uses the RISK-UE semi-empirical macroseismic method. This is based on the intensity values and a vulnerability index for the calculation of the mean damage degree for the beta distribution.

OpenQuake uses ground motion intensities and fragility curves.

The two methods are obviously different, but, no matter what their path, the results of both methods have the same aim: asses the damages after an earthquake. Considering this same objective, the results from the two methods can be compared.

Nevertheless, we agree with your comment, and we will add a paragraph to summarise both methods.

A few articles attempt to address the issue (e.g. Lestuzzi et al. 2016).
Lestuzzi, P., Podestà, S., Luchini, C. et al. Seismic vulnerability assessment at urban scale for two typical Swiss cities using Risk-UE methodology. Nat Hazards 84, 249–269 (2016). https://doi.org/10.1007/s11069-016-2420-z

Description of revision
The method used in Armagedom is described in lines 521-532.

**Line 237.** Regarding the Damage based on observations. Again, this is rather difficult to understand. The paragraph starts speaking about test related to vulnerability and risk modelling, but the conclusion of the paragraph is simply a table assigning building taxonomies to the building database. If the author wants to create different taxonomies to their database, they should name the section: Vulnerability estimation or something related to that.

Review round 1 reply
We understand the remark of the reviewer. Yes, the name of the Section is not adequate, and this will be changed in the revised manuscript.

We do not want to create different (new) taxonomies to our database, we just want to assign, based on the structural information in the AFPS forms, the building in the existing taxonomies (both RISKUE and ESRM20 building classes). The names of these two taxonomies are different but there is a real physical correspondence between these two typologies, based on the construction code, construction material, load-bearing resistant system, etc.).

Description of revision
The estimations based on the observations are described in the revised section 2.2 "Post-seismic emergency diagnoses dataset".

**Line 248.** Regarding Estimated damage based on a "building-by-building" Here the authors, compare the building-by-building damage results using OPENQUAKE when using Ritz et al. scenario and Shakemap analysis (try to find a better name for this). Initially those analysis use the same Vs30 model and they also include a new Vs30 model (named ESHM20 Vs30) to the Ritz et al. scenario. Again, this is very messy. If you want to compare the influence of the ground motion scenario, it is clear the comparison between Ritz and Shakemap using the same Vs30 model but if you want to compare the Vs30 influence you should also include the Shakemap scenario with the ESHM20 Vs30 model to be consistent.

Review round 1 reply
We agree that our presentation of the various comparisons in the submitted manuscript is unclear. We will revise the nomenclature and we will clarify the assumptions behind each scenario, using a table like this:

| GM Map ID | Type | GMM | Site model | Rupture model | Observations |
|---|---|---|---|---|---|
| GM1 | ground-motion field | KothaEtAl2020 Site | BRGM soil classes to Vs30 | Ritz et al. | No |
| GM2 | ground-motion field | KothaEtAl2020 ESHM20Slope Geology | Slope & Geology (ESRM20 data) | Ritz et al. | No |
| GM3 | ground-motion field | KothaEtAl2020 Site | ESRM20 Vs30 data | Ritz et al. | No |
| GM4 | shake-map | KothaEtAl2020 Site | BRGM Soil class to Vs30 | Ritz et al. | Seismic stations |

Description of revision
The section on the estimations using the "building-by-building" exposure model has been revised (Section 5.1) and includes the table above (Table 5-1).

**Line 287.** Regarding Estimated damage based on aggregated exposure model. Here the authors carry out many different comparisons. Again, it is very messy, and it is not clear why you are doing it and what are you looking for.

Review round 1 reply
Again, we will take greater care of explaining these various comparisons. We propose to add the following table to summarize the different damage estimation models:

| Damage scenario ID | GM Map ID | Exposure model |
|---|---|---|
| DS1 | GM1 | BRGM exposure |
| DS2 | GM1 | ESRM20 exposure |
| DS3 | GM2 | BRGM exposure |
| DS4 | GM2 | ESRM20 exposure |
| DS5 | GM3 | BRGM exposure |
| DS6 | GM3 | ESRM20 exposure |
| DS7 | GM4 | BRGM exposure |
| DS8 | GM4 | ESRM20 exposure |

These damage scenarios can then be compared to the damage "observations" DD2 and DD3, as introduced in the following table:

| Observed Damage Data ID | Exposure resolution | Exposure data | Damage estimation method | Damage conversion method | |
|---|---|---|---|---|---|
| DD1 | Building-by-building (327 buildings) | AFPS emergency survey | AFPS emergency observations on 327 buildings (Green / Yellow / Red tags) | Conversion to EMS-98 damage grades (Tab. 2.1) | Related to Fig. 4 |
| DD2 | Infra-municipality districts (2778 buildings) | National statistics database (BRGM-CCR) | AFPS emergency observations on 327 buildings (Green / Yellow / Red tags) + "Extrapolation" | Conversion to EMS-98 damage grades with expert judgment (Tab. 3.6) | Related to Fig. 5 |
| DD3 | Infra-municipality districts (2778 buildings) | National statistics database (BRGM-CCR) | AFPS emergency observations on 327 buildings (Green/Yellow/Red tags) + "Extrapolation" | Conversion to EMS-98 damage grades (Tab. 2.1) + Bias adjustment on total number of 2778 buildings (accounting for non-surveyed buildings) | Related to Fig. 5 |

Description of revision
In the revised manuscript, the damage estimations based on the aggregated exposure models are presented in Section 5.2. The aggregated exposure models are described in Section 3.2. The tables above have been added (Tables 2-3, 5-2).

**Conclusions:** The first conclusion is that the FM2010 model is the best to estimate macroseismic intensity since it is closer to Schlupp et al. (2022). Is this the model used in your national seismic hazard maps or shakemaps to convert from ground motion to macroseismic intensity? Is it only appropriate for the Le Teil region?

Review round 1 reply
The national seismic hazard map is not based on the use of GMICE. In mainland France, the "official" shake-map generated by BCSF uses the GMICE by Caprio et al. (2015). We will add a sentence of discussion on this.

Description of revision
This subject is no longer part of the conclusions.

Along the paper you have made multiple comparison, so it would be nice if the conclusions also indicate the main conclusion about those comparisons. At the moment, 11 lines are conclusions regarding the ground motion comparisons (sections 3.1 and 3.2) and 11 lines are conclusions regarding the rest of comparisons (3.3.1 to 3.3.4).

Review round 1 reply
We will add a paragraph of main conclusions in the Conclusions section. This comment is also in line with a remark from Reviewer 1.

Description of revision
A paragraph has been added to the conclusions (lines 568-578).

---

## Referee Report (RR1)

**Review of Manuscript egusphere-2023-1740 R1**

**Comparing components for seismic risk modelling using data from the 2019 Le Teil (France) earthquake**

This manuscript is a revised version of the previously entitled "Testing the 2020 European Seismic Hazard and Risk Models using data from the 2019 Le Teil (France) earthquake" paper draft. The newly-proposed title ("Comparing components for seismic risk modelling using data from the 2019 Le Teil (France) earthquake") is much more appropriate and better reflects the contents of the manuscript. It is now clear that the work is not a "test" on the 2020 European Seismic Hazard and Risk Models.

The authors have done a commendable job reorganising the contents of the paper and labelling the different (combinations of) components being used for the different damage scenario calculations. It is now a lot clearer what comparisons are being made, which makes the work a lot more readable and understandable. The new figures and tables are very useful, as are the modifications introduced to the bar plots of the results.

There are, however, still some points that require clarification before publication. I recommend that the manuscript be considered for publication after the following minor revisions.

**Main Comments**

1.  Lines 22-23: The statement "an exposure and fragility model assembled herein leads to lower probabilities for damage grades 3-5 than the ESRM20 exposure and fragility model" suggests that the difference observed in Figure 6a is much larger than it looks (please see my comment on line 576 of the conclusions). Please re-phrase.

2.  Lines 53-55: The description of section 2 is focusing on just one of the aspects being presented there. I suggest either re-writing it as a more general statement (like "section 2 focuses on the interpretation of post-earthquake assessment damage data acquired for a small sample of buildings in terms of a 3-level-scale…"), without going into more details, or enumerating the several things being done (i.e., add that three different distributions of damage are defined, etc).

3.  Lines 69-71: I suggest to re-arrange the order of the three comparisons as they appear later in the paper (if so, the subsequent sentences need to be changed, e.g., "the last two types" → "the first two…").

4.  Lines 127-129: Is the distribution of green/yellow/red tags across these 174 entries similar or different from the distribution of the other 327 entries (i.e., shown in Table 2-2)? This might give a hint on how the inspections were conducted (e.g., are undamaged buildings under-sampled?), which can inform the other two criteria (DD2 and DD3) defined in the paper, or indicate if any bias is introduced by removing these 174 entries (35%) of the damage dataset. Please comment in the paper.

5.  Line 163 and Table 2-3: Please re-phrase. The word "extrapolation" immediately brings to mind that the same proportions of damage of the 327 buildings were applied to the 2,778 buildings, while a more complex combination of observed values and judgement-based decisions was applied for DD2 and DD3.

6.  Lines 176-178: Please explain the logic behind your judgement. This is relevant to give meaning to DD2, given that a weight of 0.82 is applied to those numbers, which means that DD2 ends up being almost a pure reflection of such judgement (i.e., the red/yellow/green tag-to-EMS98 conversion has little impact on DD2). It is clear that the process is inherently subjective, but explaining the rationale behind the subjectivity would make it more transparent and useful. Something that strikes the eye is the assumption that every single building of the 2,778 set was damaged at least non-structurally. This seems like a strong assumption, especially for buildings for which inspections were not requested. At the same time, non-structural damage is difficult to assess, as it is quite common to encounter non-structural cracks in buildings, and very hard to determine if they were caused by that particular earthquake or not. Please discuss the rationale behind assuming no EMS-98 damage grade zero at all in the whole municipality.

7. Lines 184-188: These lines explain how to use Table 2-8 in combination with the P(tag) column of Table 2-4 to obtain a final damage distribution, but not where the numbers in Table 2-8 are coming from. The caption of Table 2-8 says they were defined by expert judgement. For the same reasons stated in the previous comment (regarding DD2), please explain the rationale behind your expert judgement. Please highlight that this criterion applies to this particular earthquake, to avoid an erroneous interpretation that in any earthquake a red tag would imply only a 5% probability of DG5.

8. Lines 264-265: Given that the date of construction was available, the lateral force coefficients could have been estimated (following Crowley et al., 2021, https://doi.org/10.1007/s10518-021-01083-3, for example). Please rephrase to simply say that the lateral force coefficient was not considered, but avoid saying it *cannot* be considered.

9. Line 276: Please clarify these are the OpenQuake names of these ground motion models and add the corresponding citations:

   - KothaEtAl2020Site: Kotha et al. (2020) (it's already in the references)

   - KothaEtAl2020SlopeGeology: Weatherill et al. (2023) (already in the references too)

   - KothaEtAl2020ESHM20 would be Weatherill et al. (2020) but, as it is not being used in the paper, there is no need to mention it (see comment below).

10. Lines 276-278: The statement "which were developed in the context of the development of the GMM KothaEtAl2020ESHM20" is not accurate (please see the explanation in my previous review). However, it is not necessary to explain all the alternative versions of this GMM, given that not all of them are being used in the paper and it is now clear that the paper is not "testing ESHM20/ESRM20", as it was presented in the previous version of the manuscript. As per my previous comment, please focus on the two GMMs used in the comparisons and provide the corresponding citations (e.g., like done in lines 318-319, "a version of…", which could be moved to section 3.3). Lines 279-281 are fine.

11. Lines 289, 291, 295-296: If the values of Vs30 used result from "averaging over the polygon of the municipality" (line 291), then they are not "the values […] at the coordinates of the exposure centroids" (line 289). Of course, once the Vs30 values are retrieved, they are used in OpenQuake as if located in the centroids, but they are not the values of Vs30 at the centroids. Please correct. Lines 295-296 present the same issue.

12. Lines 289-294: I assume the same procedure was used to extract slope and geology. Please explain in the paper.

13. Line 290: Please add the following citation for the ESRM20 exposure-to-site tool:

    Dabbeek, J., Crowley, H., Silva, V., Weatherill, G., Paul, N., and Nievas, C.I.: Impact of exposure spatial resolution on seismic loss estimates in regional portfolios, Bulletin of Earthquake Engineering, 19(14), 5819-5841, 2021.

14. Line 306, Table 3-3: Are the values in the "ESRM20" column calculated as weighted averages of the district polygons or are those the values for the points themselves (without averaging)? If so, which polygons, given that centroids 0-8 are sub-municipal divisions of Le Teil?

15. Section 3.3: In their reply to point 16a of my previous review, the authors indicated they have used the ESHM20/ESRM20 ground motion model logic tree, keeping only the branches associated with their tectonic setting (active shallow crust), which is correct. The active shallow crust branch of the logic tree still contains 15 sub-branches. I agree with the authors that all these details are likely not so relevant for most readers, but then the paper talks about 1,000 realisations of ground motion when, in reality, it looks like 15,000 realisations (1,000 x 15) were used. I suggest to add a small comment that clarifies that the ESHM20/ESRM20 ground motion model logic tree for active shallow crust area sources was used, that it consists of 15 branches (and their associated weights), and that 1,000 realisations of ground motion were sampled for each of the 15 branches. These 15 branches stem from a 5-branch discrete approximation to

the Gaussian distribution describing the regional variability of the earthquake source, which effectively represent five different levels of stress drop and thus "account for the possibility of having extreme stress parameter values", as written in line 105 of the present manuscript.

Clarification regarding the authors' reply to main comment 4c of my previous review: In my original review I did not explicitly say that I was referring to the ground motion logic tree, not the source model logic tree (e.g., when saying "The ESRM20 logic tree input file and its "cut" version used for shallow crustal areas when comparing against past earthquakes indicate that this is the GMPE used in ESRM20 to calculate losses"), though the links to the files implicitly referred to the ground motion logic tree. The authors' reply referring to the collapsed version of the logic tree for ESRM20 refers to the source model logic tree, not the ground motion logic tree. The ground motion logic tree was not collapsed for use in ESRM20.

16. Lines 318-319: Please specify which site model was used for comparing ground motions and macroseismic intensities.

17. Lines 330-337: Was correlation between spectral periods not considered in OpenQuake? If so, this can be another source of difference in the results obtained using the same rupture as the shake-map. Please comment if that is the case. Moreover, two alternative correlation models are mentioned as being used with the shake-map (BJ2008+JB2009 vs the Nearest Positive Definite Matrix). Please clarify how the two alternatives co-exist (i.e., are they all grouped together and averaged out in the results?).

18. Lines 376-377: The equation says $\sigma_{MCS}$ but the text says $\sigma_{MMI}$. Please correct.

19. Lines 373-394:

- According to the text, FM2010 is equation 1 and CA2015 is equation 2.

- In equation 1, $\sigma_{MCS}$ or $\sigma_{MMI}$ are used.

- In equation 2, $\sigma_{singleline}$ is used.

- The caption of Table 4-2 says it refers to the FM2010 model, which according to equation 1 uses $\sigma_{MCS}$ or $\sigma_{MMI}$, but the table says $\sigma_{singleline}$.

- The caption of Table 4-1 says it refers to the CA2015 model, which according to equation 2 uses $\sigma_{singleline}$, but the table says $\sigma_x$.

Please revise and adjust where needed. Moreover, I suggest keeping only the coefficients for PGA in Tables 4-1 and 4-2, as the ones for PGV are not used in the paper.

20. Line 417: It would increase clarity if section 4.2 concluded by stating which rupture model the rest of the paper is going to be based on, as selecting it was the purpose of section 4.2 (instead of stating it only later in section 5).

21. Lines 425-429, Table 5-1: For GM2, please re-phrase the site model as "ESRM20 site model (slope & geology)". For GM3, please re-phrase it as "ESRM20 Vs30 model". None of the two are data.

22. Lines 456-458: In the sentence "The fact that in these cases a green tag...", I suggest to refer back to Table 2-2, otherwise the reader might not follow where this statement is coming from.

23. Lines 471-473: I understand Table A4 was also used to convert results in section 5.1, but it was never mentioned there. I suggest stating this in section 5.1 as well. Moreover, please comment somewhere in the main text on the decision to assume that ESRM20's D0 (no damage) translates to EMS-98 damage grade 1, i.e. slight non-structural damage (see comment 6 about lines 176-178 above).

24. Line 486-487: Please re-phrase "which utilize the damage observations and expert judgement, respectively", given that DD2 is also heavily influenced by expert judgement (as explained in lines 177-178).

25. Lines 479-500: The discussion on the effect of the exposure models focuses on the results obtained with GM3 (i.e., DS5 vs DS6). Please comment if the same trends are observed when using the other GMs (i.e., DS1 vs DS2, DS3 vs DS4, DS7 vs DS8).

26. Line 510: It is stated that scenario DS5 uses the KothaEtAl2020ESHM20SlopeGeology GMM, but Table 5-2 indicates that DS5 uses GM3, which is based on KothaEtAlSite (Table 5-1) instead. Please revise and adjust.

27. Figures 6 and 7: The large peak of damage grade 3 for DD3 and the associated extreme valley of damage grade 3 (also for DD3) suggest that perhaps it was too optimistic to assume 55% of red tags meaning damage grade 3 and only 5% of red tags meaning damage grade 5 (Table 2-8). It is of course impossible to know for sure, as DD2 is also heavily influenced by expert judgement and all other values are estimates, not observations (i.e., it is not possible to pinpoint one "correct" value). It might be worth including a comment on this.

28. Line 559: The comparison of ground motions and macroseismic intensities was not "based on components of the ESRM20". The GMM used was KothaEtAl2020Site, which was not used in ESHM20 or ESRM20 (please see main comment 4a in my previous review). The rupture models were obtained from the literature, not from the ESHM20 source model. The site model used is not specified (see my comment above regarding lines 318-319). Please re-phrase.

29. Line 576: Assuming this statement comes from analysing Figure 6a, the difference in the proportion of buildings with damage grades 3-5 obtained with DS5 and DS6 looks very small. Please re-phrase the conclusion so that it better reflects the difference observed, and/or provide numbers that justify the statement.

30. Line 577: Assuming this statement comes from analysing Figure 6b, the number of buildings is larger in DS5 than in DS6 for damage grade 3, slightly larger (almost the same) in DS5 than in DS6 for damage grade 4, and slightly smaller (almost the same) in DS5 than in DS6 for damage grade 5. Moreover, lines 494-496 state that "the results of the damage scenarios for damage grades 3-5 [in terms of number of buildings] present minor differences". This contradicts the statement on line 577 of the conclusions. Please revise.

31. Lines 579-580: Please clarify what "the dataset based on the emergency post-seismic diagnosis" refers to. Damage dataset? Exposure dataset?

32. Lines 587-594: This is a very relevant point to make. However, these lines seem to refer only to DD1 and not to DD2 or DD3 (e.g., "the proposed rule"). Line 591 states that "the effect of possible alternative conversion rules" was not studied, which I assume refers to potential alternatives of Table 2-1, but it can be confusing for the reader because DD2 and DD3 *are* alternative conversion rules. Please re-phrase so that it is clear when you are referring only to DD1 and consider adding some comments regarding DD2/DD3.

33. Lines 598-599: This is a very relevant point to make as well. Given the large weight that expert opinion had in the definition of DD2 and DD3, it would be important to remind the reader here of these subjective assumptions and the fact that all comparisons carried out against DD2 and DD3 have this inherent limitation.

34. Lines 599-603: Given that inspections were carried out upon request from the owners, the assumption that undamaged buildings are underrepresented in the sample of 327 buildings seems quite reasonable. This statement appears as misaligned with the assumption in the conversion between ESRM20 and EMS-98 damage grades that ESRM20 damage grade 0 equates to EMS-98 damage grade 1 (Table A4). Regarding completely destroyed buildings not being inspected, this looks like a more complicated assumption. Post-earthquake damage assessments tend to be carried out not only "to inform about the risk associated with the use of impacted buildings" (as stated in line 603), but also to understand future housing needs and for governments to make an estimate of the need for relief funds. I would say it makes sense that completely destroyed buildings may not have been inspected in detail (to classify damage to different

structural components, for example), but they may have been tagged in terms of the global state of the building. This is why it would be important to comment in the paper about the 174 entries of the damage assessment for which only colour tags for the buildings were available (as per my comment earlier about lines 127-129). Are completely destroyed buildings better represented in these 174 entries? Please consider all this to potentially re-phrase this last paragraph of the conclusions.

**Minor Comments/Edits**

1. Line 13: To my knowledge, "emergency post-seismic diagnosis" is not a standard term. Consider replacing it with "emergency post-earthquake assessment" or "emergency post-earthquake inspection" (here and all throughout the paper).

2. Line 14: "shake-map analyses" can be replaced by simply saying "shake-maps", while "scenario damage analyses" can be replaced by "scenario damage calculations", here and all throughout the paper. The word "analysis" does not imply a computation/calculation.

3. Line 18: No need for inverted commas around "building-by-building" (here and all throughout the paper). Many inverted commas throughout the paper could be removed to improve legibility.

4. Line 56: "to investigate as  components".

5. Lines 62: "Shake-maps are employed".

6. Line 71: "comparisons based on a building-by-building exposure model".

7. Line 75: Mentioning the GMM is too much detail at this stage (the other GMM model is not being mentioned, there is no reference or explanation, all this comes later), I suggest to just focus on the site models.

8. Line 93: "and a moment magnitude Mw 4.9".

9. Lines 97-98: "the municipality of Le Teil, that they cannot".

10. Lines 115 & 117: erase "the" in front of "Le Teil".

11. Line 131: "to EMS-98 damage grades".

12. Line 158: "compare results of analyses against three different sets".

13. Line 173: A reference to Table 2-2 at the end of the sentence would improve clarity.

14. Line 180: I believe the intended reference is Table 2-6, not 2-7.

15. Captions of Tables 2-6 and 2-7: "of the EMS-98 damage grades".

16. Lines 190-205: Please consider labelling the final damage probabilities for DD1, DD2 and DD3, so that the reader can easily come back to them once they reach the sections with the comparisons. Please add these final damage probabilities to Table 2-8.

17. Line 244: "similarity of their lateral load-bearing systems".

18. Line 249 and several instances along the paper: There is an error with the references. The text "Erreur! Source du renvoi introuvable" appears.

19. Line 260: Eliminate "building class".

20. Line 284: Please define "EC8" and add the appropriate citation.

21. Line 286: "extracted".

22. Line 322: "was  calculated with the".

23. Line 331: At the end of the line it says "Moreover, the", but then the next line contains a different sentence.

24. Line 332: What do you mean by "updated" parameters? "Updated" with respect to what? Please revise this sentence.

25. Line 421: I suggest adding "which includes 327 buildings with classes defined in Table 3-2", to ease readability.

26. Line 423: It is three different site models being used (BRGM Vs30, ESRM20 Vs30, and ESRM20 slope and geology), not two (as the sentence says). Please be consistent with the use of upper/lower case for BRGM (sometimes it appears in lower case and sometimes in upper case).

27. Line 450: Should the reference be Table 2-2 instead of Table 3-2?

28. Line 455: I suggest adding "the horizontal structural elements have a yellow tag (see Table 2-1)".

29. Line 481: I suggest adding "the same rupture model, GMM, and site model (GM3)", to ease readability.

30. Lines 500-501: I suggest adding to the end of the caption "using ground motion map ID GM3", to ease readability.

31. Lines 531-532: I infer that Armagedom takes the map of macroseismic intensity as a user input. As it is currently phrased, it can be interpreted that this particular map (of Schlupp et al., 2022) is *the only* map of macroseismic intensity that is hard-coded in Armagedom. I suggest re-phrasing.

32. Lines 538-540: I suggest erasing this last sentence. Now that the goals of the paper have been re-phrased and it is clear that the paper is not a test of ESHM20/ESRM20 components, the purpose of this comparison has become more self-explanatory.

33. Line 576: "The scenario damage analyses leads to".

34. Lines 581-582: "The estimation based on the Armagedom tool results in probabilities".

---

## Author Response (AR2)

23 Feb 2024

Editor decision: Publish subject to minor revisions (review by editor)

by Fabrice Cotton

Public justification (visible to the public if the article is accepted and published):

The paper is much improved and the results are better presented and discussed. One reviewer provides a solid list of recommendations/comments that should be considered in the final version.

We would like to thank you very much for handling our manuscript. We appreciate it very much.

I also have some suggestions/recommendations:

- The abstract should be strengthened to better reflect the lessons learned from this analysis. The current last sentence is weak and does not provide solid information to the reader. I suggest adding to the abstract the key "lessons learned" to improve future testing, which are listed in the conclusion.

Thank you for suggesting this improvement. The final lines of the revised abstract state the lessons learned from the comparisons in the paper.

- The authors compare many models, but do not provide enough information on which models are the most "solid/likely". Of course it is difficult to compare models, but the paper would really be more useful if the authors could give an expert opinion on them. For example, the paper states that the ESRM20 and brgm exp exposure models are different, but it is not really clear which of the two models is considered the most reliable (would the authors give similar weights to them in a logic tree approach?)

Thanks for your comment, which is obviously very important. This is quite a delicate issue and difficult to discuss, as the results are still open to debate despite the different models used. However, as we agree with you on the usefulness of the article in providing an expert opinion, we have tried to offer some guidance to the reader. (lines 670-687).

First of all, contrary to what might have been expected, the building-by-building scenario calculations did not provide enough reliable information to be able to calibrate the scenarios. This is due to the need to convert tags into degrees of damage, or to reinterpret inspection sheets. In France (as in Italy), emergency diagnostics (by the AFPS or by the firefighters) tag buildings in three colours (red, yellow and green), which is normal in an emergency context. One recommendation we can make, which will be useful for research and benchmarking work, is to add to the emergency inspection sheets the classification of the building according to the EMS-98 damage grade or to the damage scale in the ESRM20.

As far as the calculations based on the aggregate exposure models are concerned, we can say that, for site effects, the combination of BRGM $V_{S30}$ model and BRGM's (infra-communal) exposure is the best choice at the commune scale. This choice is supported by the values of the $V_{S30}$ in tables 3.3 and 3.4, where the values of this combination are closest to the site effects expected in the area. There are two reasons for this: the resolution of the exposure (nine points instead of one) and the resolution of the site effect zones in the BRGM $V_{S30}$ model is better than that of the ESRM20, which is perfectly normal since the ESRM20 has been developed for application on the European scale.

Now, always in terms of aggregate models, from an exposure perspective, it appears that the simplification of the ESRM20 exposure and fragility model has a minor effect on the results (Figure 6a), at the municipal scale, if regional hazards and site effects have good description and resolution. Hence the complexity of drawing a general conclusion.

This is what emerges from this study and what is important to mention is that the resolution (extent) of the exposure is also important for the description of the effects of the site (as long as we use the centroid or even the mean within) not only for vulnerability calculation; if it is necessary to allocate resources, we would prioritize the detailed description of site effects and the assignment of the right typologies in the smallest exposure zones. This is quite consistent with other studies.

Finally, concerning our results, and considering that those of Armagedom software were the most calibrated with the observations through the AFPS studies, we think that, in our case, the best choice is the DS1 scenario which is the combination between the GM1 and BRGM exposure.

- It is not clear to me how low-cost sensors can contribute to a better explanation of damage levels (last sentence of the conclusion).

The last sentence of the conclusion has been revised and refers to Goulet et al. (2015), who proposed this.

- The conclusion discusses the use of the AS2000 model (Atkinson and Sonley, 2000 ?). This model is not mentioned in the analysis or in the bibliography. The use of many acronyms AS2000, FM2010, Ko2020 makes the paper difficult to read (the link with the reference list becomes difficult). I suggest to stick to the classical way of citing papers (Atkinson and Sonley, 2000 instead of AS2000).

The conclusion no longer discusses the AS2000 model.
The acronyms have been replaced in the main body of the text. Now they only appear in Figure 4.

- Figure captions ("model") to the right of Figure 1 and 2 are not necessary.

The captions ("model") have been removed from Figures 1 and 2.

**Review of Manuscript egusphere-2023-1740 R1**

Comparing components for seismic risk modelling using data from the 2019 Le Teil (France) earthquake

This manuscript is a revised version of the previously entitled "Testing the 2020 European Seismic Hazard and Risk Models using data from the 2019 Le Teil (France) earthquake" paper draft. The newly-proposed title ("Comparing components for seismic risk modelling using data from the 2019 Le Teil (France) earthquake") is much more appropriate and better reflects the contents of the manuscript. It is now clear that the work is not a "test" on the 2020 European Seismic Hazard and Risk Models.

The authors have done a commendable job reorganising the contents of the paper and labelling the different (combinations of) components being used for the different damage scenario calculations. It is now a lot clearer what comparisons are being made, which makes the work a lot more readable and understandable. The new figures and tables are very useful, as are the modifications introduced to the bar plots of the results.

There are, however, still some points that require clarification before publication. I recommend that the manuscript be considered for publication after the following minor revisions.

We would like to thank you very much for all the comments and the guidance, which were of major help in improving the manuscript.

**Main Comments**

1. Lines 22-23: The statement "an exposure and fragility model assembled herein leads to lower probabilities for damage grades 3-5 than the ESRM20 exposure and fragility model" suggests that the difference observed in Figure 6a is much larger than it looks (please see my comment on line 576 of the conclusions). Please re-phrase.

The revised abstract (line 24) states that the differences are small, which is in agreement with the revised conclusions.

2. Lines 53-55: The description of section 2 is focusing on just one of the aspects being presented there. I suggest either re-writing it as a more general statement (like "section 2 focuses on the interpretation of post-earthquake assessment damage data acquired for a small sample of buildings in terms of a 3-levelscale…"), without going into more details, or enumerating the several things being done (i.e., add that three different distributions of damage are defined, etc).

Lines 58-62 have been re-written based on your suggestion without going into more details.

3. Lines 69-71: I suggest to re-arrange the order of the three comparisons as they appear later in the paper (if so, the subsequent sentences need to be changed, e.g., "the last two types" □ "the first two…").

In the revised manuscript (lines 76-80), the comparisons are ordered as they appear in the paper, and the subsequent sentences have been revised.

4. Lines 127-129: Is the distribution of green/yellow/red tags across these 174 entries similar or different from the distribution of the other 327 entries (i.e., shown in Table 2-2)? This might give a hint on how the inspections were conducted (e.g., are undamaged buildings under-sampled?), which can inform the other two criteria (DD2 and DD3) defined in the paper, or indicate if any bias is introduced by removing these 174 entries (35%) of the damage dataset. Please comment in the paper.

We have added Table A5 and added a comment in the manuscript (lines 137-140) explaining that the removal of these entries does not introduce any significant bias.

5. Line 163 and Table 2-3: Please re-phrase. The word "extrapolation" immediately brings to mind that the same proportions of damage of the 327 buildings were applied to the 2,778 buildings, while a more complex combination of observed values and judgement-based decisions was applied for DD2 and DD3.

The word "extrapolation" has been replaced with the word "adjustment" (line 174).

6. Lines 176-178: Please explain the logic behind your judgement. This is relevant to give meaning to DD2, given that a weight of 0.82 is applied to those numbers, which means that DD2 ends up being almost a pure reflection of such judgement (i.e., the red/yellow/green tag-to-EMS98 conversion has little impact on DD2). It is clear that the process is inherently subjective, but explaining the rationale behind the subjectivity would make it more transparent and useful. Something that strikes the eye is the assumption that every single building of the 2,778 set was damaged at least non-structurally. This seems like a strong assumption, especially for buildings for which inspections were not requested. At the same time, nonstructural damage is difficult to assess, as it is quite common to encounter non-structural cracks in buildings, and very hard to determine if they were caused by that particular earthquake or not. Please discuss the rationale behind assuming no EMS-98 damage grade zero at all in the whole municipality.

The revised manuscript discusses the reasoning behind the selection of the probabilities based on our judgement, and the assumption that every building has at least non-structural damage. We claim that this assumption is reasonable in the case of the inspected buildings. Moreover, we acknowledge that this assumption may lead to an overestimation of non-structural damage in the uninspected buildings, but this overestimation should not be excessive due to pre-existing non-seismic damage (lines 190-198; 474-478; 529-533).

7. Lines 184-188: These lines explain how to use Table 2-8 in combination with the P(tag) column of Table 2-4 to obtain a final damage distribution, but not where the numbers in Table 2-8 are coming from. The caption of Table 2-8 says they were defined by expert judgement. For the same reasons stated in the previous comment (regarding DD2), please explain the rationale behind your expert judgement. Please highlight that this criterion applies to this particular earthquake, to avoid an erroneous interpretation that in any earthquake a red tag would imply only a 5% probability of DG5.

The revised manuscript (lines 208-210) explains that the Table 2-8 reflects the judgement of experts who participated in the post-seismic emergency assessments. It also states that they apply only in the case of this earthquake.

8. Lines 264-265: Given that the date of construction was available, the lateral force coefficients could have been estimated (following Crowley et al., 2021, https://doi.org/10.1007/s10518-021-01083-3, for example). Please rephrase to simply say that the lateral force coefficient was not considered, but avoid saying it cannot be considered.

The manuscript has been rephrased as suggested (lines 292-294).

9. Line 276: Please clarify these are the OpenQuake names of these ground motion models and add the corresponding citations:
- KothaEtAl2020Site: Kotha et al. (2020) (it's already in the references)
- KothaEtAl2020SlopeGeology: Weatherill et al. (2023) (already in the references too)
- KothaEtAl2020ESHM20 would be Weatherill et al. (2020) but, as it is not being used in the paper, there is no need to mention it (see comment below).

10. Lines 276-278: The statement "which were developed in the context of the development of the GMM KothaEtAl2020ESHM20" is not accurate (please see the explanation in my previous review). However, it is not necessary to explain all the alternative versions of this GMM, given that not all of them are being used in the paper and it is now clear that the paper is not "testing ESHM20/ESRM20", as it was presented in the previous version of the manuscript. As per my previous comment, please focus on the two GMMs used in the comparisons and provide the corresponding citations (e.g., like done in lines 318-319, "a version of…", which could be moved to section 3.3). Lines 279-281 are fine.

The manuscript has been revised according to comments 9 and 10. It no longer refers to KothaEtAl2020ESHM20 (lines 306-308, 351-353).

11. Lines 289, 291, 295-296: If the values of Vs30 used result from "averaging over the polygon of the municipality" (line 291), then they are not "the values […] at the coordinates of the exposure centroids" (line 289). Of course, once the Vs30 values are retrieved, they are used in OpenQuake as if located in the centroids, but they are not the values of Vs30 at the centroids. Please correct. Lines 295-296 present the same issue.

The revised manuscript (lines 320-322) has been corrected. It states that the values are those returned for the coordinates of the centroids by the *point* workflow of the *exposure to site tool*.

12. Lines 289-294: I assume the same procedure was used to extract slope and geology. Please explain in the paper.

The revised manuscript explains this (lines 328-330).

13. Line 290: Please add the following citation for the ESRM20 exposure-to-site tool:

Dabbeek, J., Crowley, H., Silva, V., Weatherill, G., Paul, N., and Nievas, C.I.: Impact of exposure spatial resolution on seismic loss estimates in regional portfolios, Bulletin of Earthquake Engineering, 19(14), 5819-5841, 2021.

This citation has been added (line 321).

14. Line 306, Table 3-3: Are the values in the "ESRM20" column calculated as weighted averages of the district polygons or are those the values for the points themselves (without averaging)? If so, which polygons, given that centroids 0-8 are sub-municipal divisions of Le Teil?

The values for the ESRM20 column are the values for the points themselves without averaging. This is specified in the revised manuscript (line 321).

15. Section 3.3: In their reply to point 16a of my previous review, the authors indicated they have used the ESHM20/ESRM20 ground motion model logic tree, keeping only the branches associated with their tectonic setting (active shallow crust), which is correct. The active shallow crust branch of the logic tree still contains 15 sub-branches. I agree with the authors that all these details are likely not so relevant for most readers, but then the paper talks about 1,000 realisations of ground motion when, in reality, it looks like 15,000 realisations (1,000 x 15) were used. I suggest to add a small comment that clarifies that the ESHM20/ESRM20 ground motion model logic tree for active shallow crust area sources was used, that it consists of 15 branches (and their associated weights), and that 1,000 realisations of ground motion were sampled for each of the 15 branches. These 15 branches stem from a 5-branch discrete approximation to the Gaussian distribution describing the regional variability of the earthquake source, which effectively represent five different levels of stress drop and thus "account for the possibility of having extreme stress parameter values", as written in line 105 of the present manuscript.

In the original submission, we used the ground motion logic tree in the comparison with Armagedom. In the revised version submitted in the previous round of review in January, Armagedom is compared to the DS1 calculation that does not use a ground motion logic tree. We consider that the paper cannot afford to expand on this subject. We hope that you will approve this change.

Clarification regarding the authors' reply to main comment 4c of my previous review: In my original review I did not explicitly say that I was referring to the ground motion logic tree, not the source model logic tree (e.g., when saying "The ESRM20 logic tree input file and its "cut" version used for shallow crustal areas when comparing against past earthquakes indicate that this is the GMPE used in ESRM20 to calculate losses"), though the links to the files implicitly referred to the ground motion logic tree. The authors' reply referring to the collapsed version of the logic tree for ESRM20 refers to the source model logic tree, not the ground motion logic tree. The ground motion logic tree was not collapsed for use in ESRM20.

Thank you for clarifying this. Our reply in the previous round of review was incorrect. Please accept our apologies.

16. Lines 318-319: Please specify which site model was used for comparing ground motions and macroseismic intensities.

Line 353 of the revised manuscript specifies that the BRGM $V_{S30}$ site model was used.

17. Lines 330-337: Was correlation between spectral periods not considered in OpenQuake? If so, this can be another source of difference in the results obtained using the same rupture as the shake-map. Please comment if that is the case. Moreover, two alternative correlation models are mentioned as being used with the shake-map (BJ2008+JB2009 vs the Nearest Positive Definite Matrix). Please clarify how the two alternatives co-exist (i.e., are they all grouped together and averaged out in the results?).

The correlation between spectral periods was considered using the default correlation model in OpenQuake *BakerJayaram2008* by Baker and Jayaram (2008). For the spatial correlation, the by Jayaram and Baker is used. The correlation models are used to create the correlation matrix. The Nearest Positive Definite Matrix is used during the sampling process. The revised manuscript gives these details (lines 371-378).

18. Lines 376-377: The equation says σMCS but the text says σMMI. Please correct.

This has been corrected (line 416-418).

19. Lines 373-394:
- According to the text, FM2010 is equation 1 and CA2015 is equation 2.
- In equation 1, $\sigma_{MCS}$ or $\sigma_{MMI}$ are used.
- In equation 2, $\sigma_{singleline}$ is used.
- The caption of Table 4-2 says it refers to the FM2010 model, which according to equation 1 uses $\sigma_{MCS}$ or $\sigma_{MMI}$, but the table says σsingleline.
- The caption of Table 4-1 says it refers to the CA2015 model, which according to equation 2 uses $\sigma_{singleline}$, but the table says $\sigma_x$.

Please revise and adjust where needed. Moreover, I suggest keeping only the coefficients for PGA in Tables 4-1 and 4-2, as the ones for PGV are not used in the paper.

The manuscript lines 416-425 have been corrected. The coefficients for PGV have been deleted from Tables 4-1 and 4-2.

20. Line 417: It would increase clarity if section 4.2 concluded by stating which rupture model the rest of the paper is going to be based on, as selecting it was the purpose of section 4.2 (instead of stating it only later in section 5).

The revised manuscript (lines 449-450) states that the Ritz et al. rupture model is used in the rest of the paper.

21. Lines 425-429, Table 5-1: For GM2, please re-phrase the site model as "ESRM20 site model (slope & geology)". For GM3, please re-phrase it as "ESRM20 Vs30 model". None of the two are data.

This has been corrected (line 450, Table 5-1).

22. Lines 456-458: In the sentence "The fact that in these cases a green tag...", I suggest to refer back to Table 2-2, otherwise the reader might not follow where this statement is coming from.

A reference to Table 2-2 has been added (line 507).

23. Lines 471-473: I understand Table A4 was also used to convert results in section 5.1, but it was never mentioned there. I suggest stating this in section 5.1 as well. Moreover, please comment somewhere in the main text on the decision to assume that ESRM20's D0 (no damage) translates to EMS-98 damage grade 1, i.e. slight non-structural damage (see comment 6 about lines 176-178 above).

We hope that you will agree that the revisions with respect to comment 6 have covered this comment too.

24. Line 486-487: Please re-phrase "which utilize the damage observations and expert judgement, respectively", given that DD2 is also heavily influenced by expert judgement (as explained in lines 177-178).

The revised manuscript (lines 543-545) explains that DD2 depends mostly on expert judgement and on the damage observation on a lesser extent, while DD3 is entirely based on expert judgement.

25. Lines 479-500: The discussion on the effect of the exposure models focuses on the results obtained with GM3 (i.e., DS5 vs DS6). Please comment if the same trends are observed when using the other GMs (i.e., DS1 vs DS2, DS3 vs DS4, DS7 vs DS8).

The same trends are observed with the other GMs too (see figure below). This is mentioned in the revised manuscript (lines 551-552).

[Figure]

Figure R1 Effect of the exposure model on the probabilities per EMS-98 damage grade for the analyses calculations with an aggregated exposure including the total number of buildings in Le Teil based on the ground motion map a) GM3 b) GM1 c) GM2 d) GM4

26. Line 510: It is stated that scenario DS5 uses the KothaEtAl2020ESHM20SlopeGeology GMM, but Table 5-2 indicates that DS5 uses GM3, which is based on KothaEtAlSite (Table 5-1) instead. Please revise and adjust.

The manuscript has been corrected (lines 551-552). It states: "The damage grade probabilities in the scenario DS3, which uses the KothaEtAl2020ESHM20SlopeGeology GMM, are between the results for DS1 and DS5 for all damage grades."

27. Figures 6 and 7: The large peak of damage grade 3 for DD3 and the associated extreme valley of damage grade 3 (also for DD3) suggest that perhaps it was too optimistic to assume 55% of red tags meaning damage grade 3 and only 5% of red tags meaning damage grade 5 (Table 2-8). It is of course impossible to know for sure, as DD2 is also heavily influenced by expert judgement and all other values are estimates, not observations (i.e., it is not possible to pinpoint one "correct" value). It might be worth including a comment on this.

Indeed, in DD3, the probabilities for damage grades 3-5 depend heavily on the probabilities of these damage grades conditioned on a red tag, which were assigned based on expert judgement. Indeed, it was too optimistic to assign a 55 % probability of damage grade 3 in case of a red tag. Alternative assignments of the probabilities for a red tag may smooth out in DD3 the peak for damage grade 3. The Table below is an example (may be added to the manuscript if you deem it necessary). These comments have added to the manuscript (lines 545-548).

| tag | P(DG1|tag) | P(DG2|tag) | P(DG3|tag) | P(DG4|tag) | P(DG5|tag) |
|---|---|---|---|---|---|
| Green | 0.80 | 0.20 | 0 | 0 | 0 |
| Yellow | 0 | 0.40 | 0.60 | 0 | 0 |
| Red | 0 | 0 | 0.05 | 0.70 | 0.25 |

| tag | P(DG1|tag)·P(tag) | P(DG2|tag)·P(tag) | P(DG3|tag)·P(tag) | P(DG4|tag)·P(tag) | P(DG5|tag)·P(tag) |
|---|---|---|---|---|---|
| Green | 0.380 | 0.095 | 0.000 | 0.000 | 0.000 |
| Yellow | 0.000 | 0.125 | 0.188 | 0.000 | 0.000 |
| Red | 0.000 | 0.000 | 0.011 | 0.148 | 0.053 |
| Sum: | 0.380 | 0.220 | 0.199 | 0.148 | 0.053 |

28. Line 559: The comparison of ground motions and macroseismic intensities was not "based on components of the ESRM20". The GMM used was KothaEtAl2020Site, which was not used in ESHM20 or ESRM20 (please see main comment 4a in my previous review). The rupture models were obtained from the literature, not from the ESHM20 source model. The site model used is not specified (see my comment above regarding lines 318-319). Please re-phrase.

This sentence has been rephrased (lines 625-626) and it mentions neither the ESHM20 nor the ESRM20.

29. Line 576: Assuming this statement comes from analysing Figure 6a, the difference in the proportion of buildings with damage grades 3-5 obtained with DS5 and DS6 looks very small. Please re-phrase the conclusion so that it better reflects the difference observed, and/or provide numbers that justify the statement.

Indeed the differences are very small. Thank you for this comment. The revised manuscript has been corrected (line 644).

30. Line 577: Assuming this statement comes from analysing Figure 6b, the number of buildings is larger in DS5 than in DS6 for damage grade 3, slightly larger (almost the same) in DS5 than in DS6 for damage grade 4, and slightly smaller (almost the same) in DS5 than in DS6 for damage grade 5. Moreover, lines 494-496 state that "the results of the damage scenarios for damage grades 3-5 [in terms

of number of buildings] present minor differences". This contradicts the statement on line 577 of the conclusions. Please revise.

The sentence in line 577 has been deleted.

31. Lines 579-580: Please clarify what "the dataset based on the emergency post-seismic diagnosis" refers to. Damage dataset? Exposure dataset?

This sentence has been replaced with the following "…with the calculations DD2 and DD3 (see Section 2.2 for the details of the calculations)." (lines 647-648).

32. Lines 587-594: This is a very relevant point to make. However, these lines seem to refer only to DD1 and not to DD2 or DD3 (e.g., "the proposed rule"). Line 591 states that "the effect of possible alternative conversion rules" was not studied, which I assume refers to potential alternatives of Table 2-1, but it can be confusing for the reader because DD2 and DD3 are alternative conversion rules. Please re-phrase so that it is clear when you are referring only to DD1 and consider adding some comments regarding DD2/DD3.

The revised manuscript has been rephrased to make clear that it refers to Table 2-1 and DD1 (lines 659-662). Comments with respect to DD2/DD3 have been added (lines 662-665).

33. Lines 598-599: This is a very relevant point to make as well. Given the large weight that expert opinion had in the definition of DD2 and DD3, it would be important to remind the reader here of these subjective assumptions and the fact that all comparisons carried out against DD2 and DD3 have this inherent limitation.

We hope that you will agree that the revision in response to comment 32 has covered this comment too.

34. Lines 599-603: Given that inspections were carried out upon request from the owners, the assumption that undamaged buildings are underrepresented in the sample of 327 buildings seems quite reasonable. This statement appears as misaligned with the assumption in the conversion between ESRM20 and EMS- 98 damage grades that ESRM20 damage grade 0 equates to EMS-98 damage grade 1 (Table A4). Regarding completely destroyed buildings not being inspected, this looks like a more complicated assumption. Post-earthquake damage assessments tend to be carried out not only "to inform about the risk associated with the use of impacted buildings" (as stated in line 603), but also to understand future housing needs and for governments to make an estimate of the need for relief funds. I would say it makes sense that completely destroyed buildings may not have been inspected in detail (to classify damage to different structural components, for example), but they may have been tagged in terms of the global state of the building. This is why it would be important to comment in the paper about the 174 entries of the damage assessment for which only colour tags for the buildings were available (as per my comment earlier about lines 127-129). Are completely destroyed buildings better represented in these 174 entries? Please consider all this to potentially re-phrase this last paragraph of the conclusions.

We hope that you will agree that the revisions in reply to comment 6 have discussed the conversion between the ESRM20 and the EMS-98 damage scales.

First, as a side note, we would like to note that, to the best of our knowledge, the post-seismic emergency inspections by the AFPS have as exclusive purpose to warn with respect to the risk presented to the users of the buildings. Therefore, please allow us to claim that it is not unreasonable to assume that completely destroyed buildings may have not been inspected by the AFPS. However, we suppose that completely destroyed buildings may have been included in surveys by others such as the firefighters and researchers working on macroseismic intensity.

Moreover, the entries that were filtered out of the dataset have a similar distribution of colour tags (Table A5) to the remaining entries. We assume that they are incomplete for reasons unrelated to the level of damage.

**Minor Comments/Edits**

Unless otherwise indicated, all minor comments/edits have been made.

1. Line 13: To my knowledge, "emergency post-seismic diagnosis" is not a standard term. Consider replacing it with "emergency post-earthquake assessment" or "emergency post-earthquake inspection" (here and all throughout the paper).

2. Line 14: "shake-map analyses" can be replaced by simply saying "shake-maps", while "scenario damage analyses" can be replaced by "scenario damage calculations", here and all throughout the paper. The word "analysis" does not imply a computation/calculation.

3. Line 18: No need for inverted commas around "building-by-building" (here and all throughout the paper). Many inverted commas throughout the paper could be removed to improve legibility.

4. Line 56: "to investigate as of components".

5. Lines 62: "Shake-maps are employed".

6. Line 71: "comparisons based on a building-by-building exposure model".

This has been replaced with "comparisons based on probabilities of EMS-98 damage grades".

7. Line 75: Mentioning the GMM is too much detail at this stage (the other GMM model is not being mentioned, there is no reference or explanation, all this comes later), I suggest to just focus on the site models.

8. Line 93: "and a moment magnitude Mw 4.9".

9. Lines 97-98: "the municipality of Le Teil, that they cannot".

10. Lines 115 & 117: erase "the" in front of "Le Teil".

11. Line 131: "to EMS-98 damage grades".

12. Line 158: "compare results of analyses against three different sets".

13. Line 173: A reference to Table 2-2 at the end of the sentence would improve clarity.

14. Line 180: I believe the intended reference is Table 2-6, not 2-7.

15. Captions of Tables 2-6 and 2-7: "of the EMS-98 damage grades".

16. Lines 190-205: Please consider labelling the final damage probabilities for DD1, DD2 and DD3, so that the reader can easily come back to them once they reach the sections with the comparisons. Please add these final damage probabilities to Table 2-8.

The damage probabilities for DD1, DD2 and DD3 have been labelled in Tables 2-5, 2-7, and 2-8. The final damage probabilities have been added to Table 2-8.

17. Line 244: "similarity of their lateral load-bearing systems".

18. Line 249 and several instances along the paper: There is an error with the references. The text "Erreur! Source du renvoi introuvable" appears.

19. Line 260: Eliminate "building class".

20. Line 284: Please define "EC8" and add the appropriate citation.

21. Line 286: "extracted".

22. Line 322: "was re-calculated with the".

23. Line 331: At the end of the line it says "Moreover, the", but then the next line contains a different sentence.

24. Line 332: What do you mean by "updated" parameters? "Updated" with respect to what? Please revise this sentence.

The word "updated" has been deleted (line 333).

25. Line 421: I suggest adding "which includes 327 buildings with classes defined in Table 3-2", to ease readability.

26. Line 423: It is three different site models being used (BRGM Vs30, ESRM20 Vs30, and ESRM20 slope and geology), not two (as the sentence says). Please be consistent with the use of upper/lower case for BRGM (sometimes it appears in lower case and sometimes in upper case).

27. Line 450: Should the reference be Table 2-2 instead of Table 3-2?

We intended to add a reference to the table that gives more information on DD1. This table is Table 2-3 and it has been placed immediately after DD1 (line 457) in the revised manuscript.

28. Line 455: I suggest adding "the horizontal structural elements have a yellow tag (see Table 2-1)".

29. Line 481: I suggest adding "the same rupture model, GMM, and site model (GM3)", to ease readability.

30. Lines 500-501: I suggest adding to the end of the caption "using ground motion map ID GM3", to ease readability.

31. Lines 531-532: I infer that Armagedom takes the map of macroseismic intensity as a user input. As it is currently phrased, it can be interpreted that this particular map (of Schlupp et al., 2022) is the only map of macroseismic intensity that is hard-coded in Armagedom. I suggest re-phrasing.

Lines 540-542 have been revised.

32. Lines 538-540: I suggest erasing this last sentence. Now that the goals of the paper have been re-phrased and it is clear that the paper is not a test of ESHM20/ESRM20 components, the purpose of this comparison has become more self-explanatory.

33. Line 576: "The scenario damage analyses leads to".

34. Lines 581-582: "The estimation based on the Armagedom tool results in probabilities".